# *EPB41L4A-AS1* long noncoding RNA acts in both *cis*- and *trans*-acting transcriptional regulation and controls nucleolar biology

Alan Monziani[1,2], Juan Pablo Unfried[1,2], Todor Cvetanovic[1,2†], Igor Ulitsky[1,2]*

[1]Department of Immunology and Regenerative Biology, Weizmann Institute of Science, Rehovot, Israel; [2]Department of Molecular Neuroscience, Weizmann Institute of Science, Rehovot, Israel

## eLife Assessment

This paper provides **important** findings towards understanding the role of the lncRNA EPB41L4A-AS1 in a human cell line. The data is generally **convincing**, supported by extensive and clever integrative analysis. The work provides insights into how this lncRNA regulates gene expression via complex mechanisms; however the biological relevance awaits validation in other models.

## Abstract
Mammalian genomes are pervasively transcribed into long noncoding RNAs (lncRNAs), whose functions and modes of action remain poorly understood. *EPB41L4A-AS1* is an evolutionarily conserved, broadly and highly expressed lncRNA that produces the H/ACA snoRNA *SNORA13* from one of its introns. We studied the consequences of *EPB41L4A-AS1* perturbation in breast cancer cells and found that it acts both in *cis*, to enhance transcription of the proximal *EPB41L4A* gene and additional genes in its two flanking topologically associated domains, and in *trans* by broadly regulating gene expression, including expression of snoRNAs, transcription of genes involved in nucleolar biology and the distribution of nucleolar proteins. These effects are phenocopied by the loss of SUB1, an interactor of *EPB41L4A-AS1*, and are observed following transient perturbations of *EPB41L4A-AS1* that do not affect steady-state *SNORA13* levels or the rRNA modification it helps install. Exogenous expression of the full-length *EPB41L4A-AS1* locus but not *SNORA13* expression can rescue the *trans*-acting transcriptional effects of its perturbation. The *EPB41L4A-AS1* gene is thus a versatile locus producing RNA molecules acting on multiple levels for key cellular functions.

## Introduction

Long noncoding RNAs (lncRNAs) are a highly heterogeneous group of RNA molecules, defined as being >500 nt long and not displaying a significant protein-coding potential (*Statello et al., 2021*; *Wu et al., 2017*; *Ransohoff et al., 2018*; *Mattick et al., 2023*). Given this broad definition and the pervasive transcription of the human genome (*Hangauer et al., 2013*; *Carninci et al., 2005*; *The ENCODE Project Consortium, 2007*; *Kapranov et al., 2007*), lncRNAs are now the RNA class with the largest number of annotated genes (*Mudge et al., 2025*). However, only a limited number of lncRNAs have well-defined mechanisms of action. Long noncoding RNAs exhibit a wide range of expression levels, driven by varying transcription rates and half-lives. Similarly to protein-coding mRNAs, lncRNAs are transcribed by RNA polymerase II (Pol II), 5' capped with 7-methyl guanosine (m[7]G), polyadenylated at their 3' ends, and have a canonical exon–intron structure. Interestingly, small peptides arising from

*For correspondence:
igor.ulitsky@weizmann.ac.il

Present address: †Cold Spring Harbor Laboratory, Cold Spring Harbor, NY, United States

Competing interest: The authors declare that no competing interests exist.

short and poorly conserved translated open reading frames (ORFs) have been reported in several lncRNAs (*Ruiz-Orera et al., 2014*; *Chen et al., 2020*; *Barczak et al., 2023*; *Kesner et al., 2023*), and studies using ultra-deep sequencing of ribosome profiling data and deep proteomic analysis found evidence of translation of ORFs in hundreds of lncRNAs (*Chothani et al., 2022*). Yet it remains unclear how many of these are encoding functional peptides, mainly because these ORFs are typically poorly conserved and the protein products do not appear to be abundant (*Housman and Ulitsky, 2016*).

One way to functionally classify lncRNAs is by the nature of the functional elements harbored within their loci. One class comprises genes in which the DNA element has a function unrelated to the transcription of the locus (*Paralkar et al., 2016*). The other includes cases in which the act of transcription is functionally relevant, while the resulting RNAs are 'byproducts' of Pol II activity, which, for example, can tune the expression of another locus in 3D proximity (*Statello et al., 2021*; *Ali and Grote, 2020*; *Beucher et al., 2022*; *De Santa et al., 2010*). This class is potentially broad because it can explain the prevalence of lncRNA loci in which the lncRNAs are expressed at very few copies and/ or have sequences that are evolving without any apparent selection. For the cases where the RNA product is functional, a further distinction can be made between *cis*-acting lncRNAs, the function of which is restricted to the locus from which they are transcribed, and *trans*-acting lncRNAs, which are transcribed, processed, and then move away by diffusion or active transport to exert their function elsewhere. The latter RNAs require a relatively stable RNA molecule because they have to interact with one or more other RNA species and/or proteins, whereas, for *cis*-acting lncRNAs, such an RNA molecule may or may not be necessary, with the act of transcription and concomitant chromatin remodeling reported to be sufficient in several cases (*Gil and Ulitsky, 2020*).

Small nucleolar RNAs (snoRNAs) are intermediate-sized RNAs whose main function is to guide the installation of RNA modifications on other RNAs, mainly ribosomal RNAs (rRNAs). In mammalian cells, snoRNAs are produced from introns of polyadenylated transcripts, some of which code for proteins, and others are instead lncRNAs. Noncoding snoRNA host genes (or ncSHNG) are characterized by several characteristic features, including high and broad expression levels, relatively inefficient splicing, and sensitivity to nonsense-mediated decay (*Monziani and Ulitsky, 2023*). While lncRNA functions have been shown for some ncSNHGs, most remain poorly understood. Here, we describe a bioinformatic screen that converged on *EPB41L4A-AS1*, a very abundant and conserved ncSHNG acting as a regulator in *cis*, and our observations that this lncRNA also works in *trans* to regulate nucleolar biology.

## Results

### A computational screen for putative *cis*-acting lncRNAs

We hypothesized that lncRNAs likely to act in *cis* will be occasionally co-regulated with their target genes and also found in spatial proximity to the target promoters. We sought such lncRNA–target pairs in the GeneHancer database (*Fishilevich et al., 2017*), that connects regulatory elements and their targets by combining tissue co-expression data, enhancer-enriched histone modifications, transcription factor binding, Hi-C data, and eQTLs. We complemented this resource with RNA-seq datasets from the breast cancer cell line MCF-7 exposed to different treatments (*Sun et al., 2015*; *Nagarajan et al., 2020*; *Janky et al., 2014*; *Figure 1A* and *Supplementary file 1*). This analysis (see Methods) led us to focus on four lncRNA–target pairs connected in GeneHancer and in which both genes were significantly either upregulated or downregulated in the treatment conditions, compared to controls (*Figure 1—figure supplement 1A–D*). We knocked down (KD) each candidate for 72 hr with two distinct GapmeRs—antisense oligonucleotides with a DNA core that induces the degradation of target nascent RNA via RNAse H activity. We then tested the expression of the target genes using RT-qPCR. Among the four lncRNAs, KD of *EPB41L4A-AS1* with both GapmeRs significantly repressed *EPB41L4A* (*Figure 1B*), whereas the other lncRNAs had no substantial effects on the putative target genes.

### *EPB41L4A-AS1* is a complex lncRNA gene

The *EPB41L4A-AS1* locus is ~2 kb (kilobases) long, and the main transcribed isoform terminates ~300 nt from the 3' end of the main isoform of the convergently transcribed *EPB41L4A* gene, and >250 kb away from the *EPB41L4A* transcription start site (TSS, *Figure 1C*). An H/ACA snoRNA, *SNORA13*, a

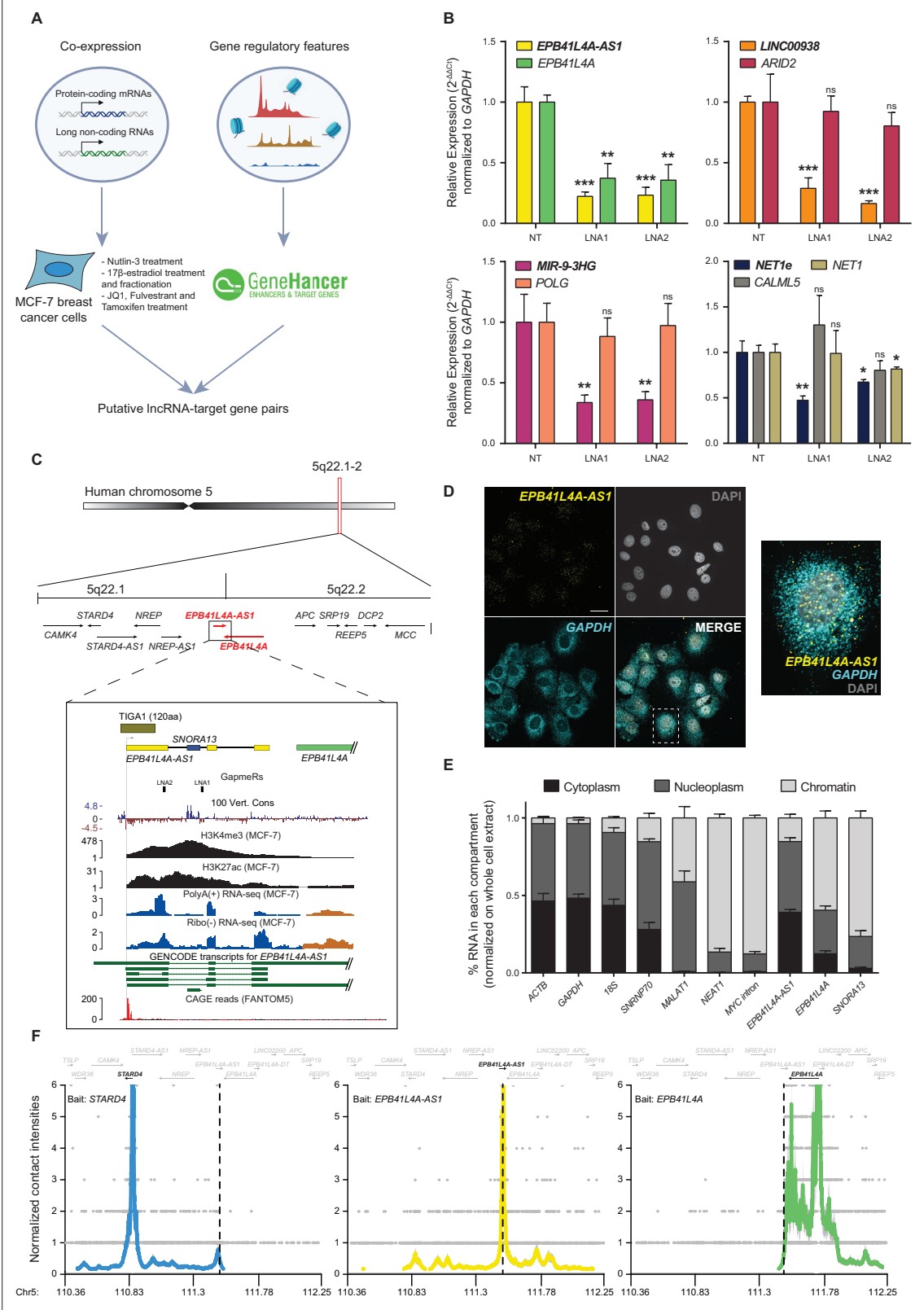

**Figure 1.** *EPB41L4A-AS1* is a highly expressed lncRNA with a widespread cellular distribution. (**A**) Graphical overview of the screen to identify candidate *cis*-acting lncRNAs. (**B**) RT-qPCR to assess the expression of each candidate *cis*-acting lncRNA and their predicted *cis*-regulated target genes, following KD with two unique GapmeRs. (**C**) (Top) Overview of the 5q22.1–2 locus, with the bars highlighting the two TADs. (Bottom) Zoom in on the *EPB41L4A-AS1* locus, with tracks for the PhyloP conservation score (UCSC genome browser), H3K4me3 and H3K27ac histone modifications (ENCODE), PolyA+

*Figure 1 continued*

and Ribo(−) RNA-seq coverage (this study), GENCODE transcripts and CAGE reads (FANTOM5). The location of the GapmeRs used in this study is also reported. (**D**) A representative single-molecule RNA FISH (smFISH) image shows *EPB41L4A-AS1* intracellular localization. *GAPDH* and DAPI were used to label the cytoplasm and the nucleus, respectively (scale bar = 20 µm, ×100 magnification). (**E**) RT-qPCR analysis of subcellular fractionation experiments. The percentages of RNA in each compartment were obtained by normalizing the expression in the different fractions to that in whole cells. (**F**) UMI-4C contact profiles using baits targeting the TSSes of *STARD4* (left), *EPB41L4A-AS1* (center), or *EPB41L4A* (right). The dotted line represents the center of the *EPB41L4A-AS1* locus. All experiments were performed in n = 3 biological replicates, except UMI-4C with n = 2, with the error bars in the bar plots representing the standard deviation. ns = p > 0.05; *p < 0.05; **p < 0.01; ***p < 0.001 (two-sided Student's *t*-test).

The online version of this article includes the following figure supplement(s) for figure 1:

**Figure supplement 1.** UCSC Genome Browser view of the loci containing the selected lncRNAs and target genes for validation.

**Figure supplement 2.** EPB41L4A-AS1 is dysregulated in several cancer types and correlates with survival.

**Figure supplement 3.** EPB41L4A-AS1 expression is altered upon multiple stimuli.

homolog of yeast *snR35*, is embedded within the first intron of *EPB41L4A-AS1*. This snoRNA guides pseudouridylation on position 1248 of the small subunit 18S rRNA (*Babaian et al., 2020*). This modification is the first step in the installation of one of the most complex RNA modifications known to date—*N1*-methyl-*N3*-aminocarboxypropyl-pseudouridine (m(1)acp(3)$\Psi$)—that is located next to the tRNA P-site and is often lost in human cancers (*Babaian et al., 2020*). Recently, *SNORA13* has been reported to negatively regulate ribosome biogenesis by decreasing the incorporation of RPL23 into maturing 60S subunits, eventually triggering p53-mediated cellular senescence (*Cheng et al., 2024*).

Several groups reported different functions of *EPB41L4A-AS1*. It was found to be differentially expressed across the cell cycle, and its KD affected cell cycle progression and reduced proliferation in multiple cancer cell lines (*Hegre et al., 2021*). Another study implicated *EPB41L4A-AS1* in glucose metabolism through interactions with HDAC2 and NPM1 (*Liao et al., 2019*). Silencing of *EPB41L4A-AS1* led to HDAC2 translocation into the nucleoplasm, where it binds the *VDAC1* and *VHL* promoter regions, among others (*Liao et al., 2019*), and effects on the expression of additional genes were reported in other systems as well (*Zhu et al., 2019*; *Liao et al., 2022*; *Bin et al., 2021*; *Wang et al., 2021*). In human cancers, *EPB41L4A-AS1* is also downregulated and correlates with a poor prognosis in a variety of cancers, including breast cancer (BRCA), which we also observed by analyzing the TCGA data (*Figure 1—figure supplement 2*), aligning well with the widespread loss of the m(1)acp(3)$\Psi$ modification. *EPB41L4A*, instead, is a poorly studied gene, reported to be involved in predisposition to papillary thyroid carcinoma by modulating the WNT/β-catenin pathway (*Ishiguro et al., 2000*; *Comiskey et al., 2020*), possibly playing a role in connecting the cytoskeleton to the plasma membrane, as well as playing unspecified roles throughout development (*Guo et al., 2011*).

## *EPB41L4A-AS1* is highly expressed in both the nucleus and cytoplasm and is regulated in multiple conditions

Using single-molecule fluorescence in situ hybridization (smFISH) and subcellular fractionation, we found that *EPB41L4A-AS1* is expressed at an average of 33.37 ± 3.95 molecule per cell and displays both nuclear and cytoplasmic localization in MCF-7 cells (*Figure 1D*), with a minor fraction associated with chromatin as well (*Figure 1E*). In contrast, both *EPB41L4A* and *SNORA13* were mostly found in the chromatin fraction (*Figure 1E*), the former possibly due to the length of its pre-mRNA (>250 kb), which would require substantial time to transcribe (*Didiot et al., 2018*; *Ly et al., 2022*; *Bahar Halpern et al., 2015*; *Lubelsky and Ulitsky, 2018*). Coding-Potential Assessment (CPAT) (*Wang et al., 2013*) analysis showed the sequence of *EPB41L4A-AS1* has little coding potential. A translated ORF in the first exon of *EPB41L4A-AS1* has been reported to produce a small peptide (TIGA1), which localizes to the mitochondria and induces growth arrest (*Yabuta et al., 2006*; *Figure 1C*). Applying CPAT to each exon separately, we found evidence for some coding potential only in the first exon that harbors the TIGA1 peptide (*Figure 1—figure supplement 3A*). Importantly, analysis of the RNA-seq signal coverage shows that the common *EPB41L4A-AS1* isoform in MCF-7 cells begins downstream of the AUG codon at the start of TIGA1, and TIGA1 does not have another in-frame AUG in its sequence. Consistently, the vast majority of CAGE tags at the *EPB41L4A-AS1* are also downstream of the AUG of TIGA1, consistent with the start sites of most isoforms of *EPB41L4A-AS1* annotated in GENCODE (*Figure 1C*). We therefore conclude that while some fraction of *EPB41L4A-AS1* might

produce TIGA1 peptides, most RNA products of this gene in MCF-7 do not produce TIGA1 or other peptides predicted to be functional.

Consistent with a previous study (*Yabuta et al., 2006*), we found a strong upregulation of *EPB41L4A-AS1*, *EPB41L4A*, and *SNORA13* in MCF-7 cells upon serum starvation (*Figure 1—figure supplement 3B*). In contrast with a previous report (*Hegre et al., 2021*), we did not detect *EPB41L4A-AS1* to be differentially expressed across the cell cycle in MCF-7 cells (*Figure 1—figure supplement 3C*). We further explored the effect of different stresses and environmental conditions on *EPB41L4A-AS1* by analyzing its expression in a published dataset comprising five different cell types exposed to a panel of 50 different substances (*Moyerbrailean et al., 2016*; *Figure 1—figure supplement 3D*), as well as exposing MCF-7 cells to lipopolysaccharide (LPS; bacterial infection), hydrogen peroxide ($H_2O_2$; oxidative stress), thapsigargin (ER stress and unfolded protein response), and etoposide (DNA damage) (*Figure 1—figure supplement 3E*). Strikingly, *EPB41L4A-AS1* expression was induced in nearly all conditions, often together with *EPB41L4A* but not *SNORA13*, suggesting the levels of the snoRNA are typically detached from that of the lncRNA (*Fafard-Couture et al., 2021*), likely because the latter is substantially more stable (see below).

## *EPB41L4A-AS1* regulates *EPB41L4A* transcription

We assessed the 3D chromatin architecture around the *EPB41L4A-AS1* locus by performing UMI-4C-seq (*Schwartzman et al., 2016*) with baits targeting the lncRNA, *EPB41L4A* and the promoter of *STARD4*—a flanking gene found ~650 kb upstream of the *EPB41L4A-AS1* promoter. *EPB41L4A-AS1* formed contacts with both genes (*Figure 1F*, middle), with prominent peaks corresponding to the promoter of *EPB41L4A*. Interestingly, the promoter of *EPB41L4A* displays several contacts with its own gene body (*Figure 1F*, right panel), consistent with a 'stripe' in HiC data where a single anchor interacts with the whole topologically associated domain (TAD) at high frequency (*Vian et al., 2018*), with those frequencies reduced to below background levels upstream of the *EPB41L4A-AS1* locus. Conversely, contacts from the *STARD4* promoter also show a peak near *EPB41L4A-AS1* and then reduced below the background (*Figure 1F*, left panel). These data indicate that *EPB41L4A-AS1* is located at a TAD boundary region, forming strong spatial contacts with sites upstream and downstream, including the promoter of *EPB41L4A*. In agreement with this, *EPB41L4A-AS1* is located between the 5q22.1 and 5q22.2 chromosomal bands, delimiting two TADs as evident in both our 4C-seq and Micro-C data from H1 cells (*Figures 1 and 2*). Both *EPB41L4A* and *EPB41L4A-AS1* are characterized by CTCF binding near their promoters, and the lncRNA locus is extensively demarcated with H3K27ac (*Figure 2A*). A recent study identified the CTCF cluster near *EPB41L4A-AS1* as one of the most insulated in human embryonic stem cells (*Salnikov et al., 2024*).

We then focused on a possible reciprocal regulation between *EPB41L4A-AS1* and *EPB41L4A*. *EPB41L4A-AS1* KD with antisense GapmerRs reduced the expression of *EPB41L4A* mRNA and its promoter-associated divergent lncRNA *EPB41L4A-DT* (*Figure 2B*), as well as the EPB41L4A protein levels (*Figure 2C*). CUT&RUN-qPCR of the H3K27ac mark—which is associated with active promoters and enhancers—showed a reduction of H3K27ac at the *EPB41L4A* promoter (*Figure 2D*). In contrast to the effects of GapmeRs, which mostly act in the nucleus and on the nascent RNA (*Liang et al., 2011*; *Hagedorn et al., 2018*), KD of *EPB41L4A-AS1* with siRNAs, which instead mainly act on the mature RNA in the cytoplasm, did not affect *EPB41L4A* levels (*Figure 2—figure supplement 1A*). Levels of *EPB41L4A* were not affected by increased expression of *EPB41L4A-AS1* from the endogenous locus by CRISPR activation (CRISPRa), nor by its exogenous expression from a plasmid (*Figure 2—figure supplement 1B, C*). The former suggests that endogenous levels of *EPB41L4A-AS1*—that are far greater than those of *EPB41L4A*—are sufficient to sustain the maximal expression of this target gene in MCF-7 cells. Moreover, downregulation (*Figure 2—figure supplement 1D, E*) or upregulation (*Figure 2—figure supplement 1F, G*) of *EPB41L4A* did not have any consistent effects on *EPB41L4A-AS1* expression, suggesting a unidirectional effect of *EPB41L4A-AS1* transcription of its nascent RNA product on *EPB41L4A*. Consistent with a *cis*-acting regulation, the reduced *EPB41L4A* mRNA levels after *EPB41L4A-AS1* knockdown were rescued only after CRISPRa of the endogenous *EPB41L4A-AS1* locus (*Figure 2—figure supplement 1H, I*). To study the potential mode of action of this regulation, we inspected UMI-4C data anchored at the *EPB41L4A-AS1* promoter and found a reduction in the contact frequency with the *EPB41L4A* promoter upon treatment with *EPB41L4A-AS1*-targeting GapmeRs, that was statistically significant for LNA1 (*Figure 2E* and *Figure 2—figure*

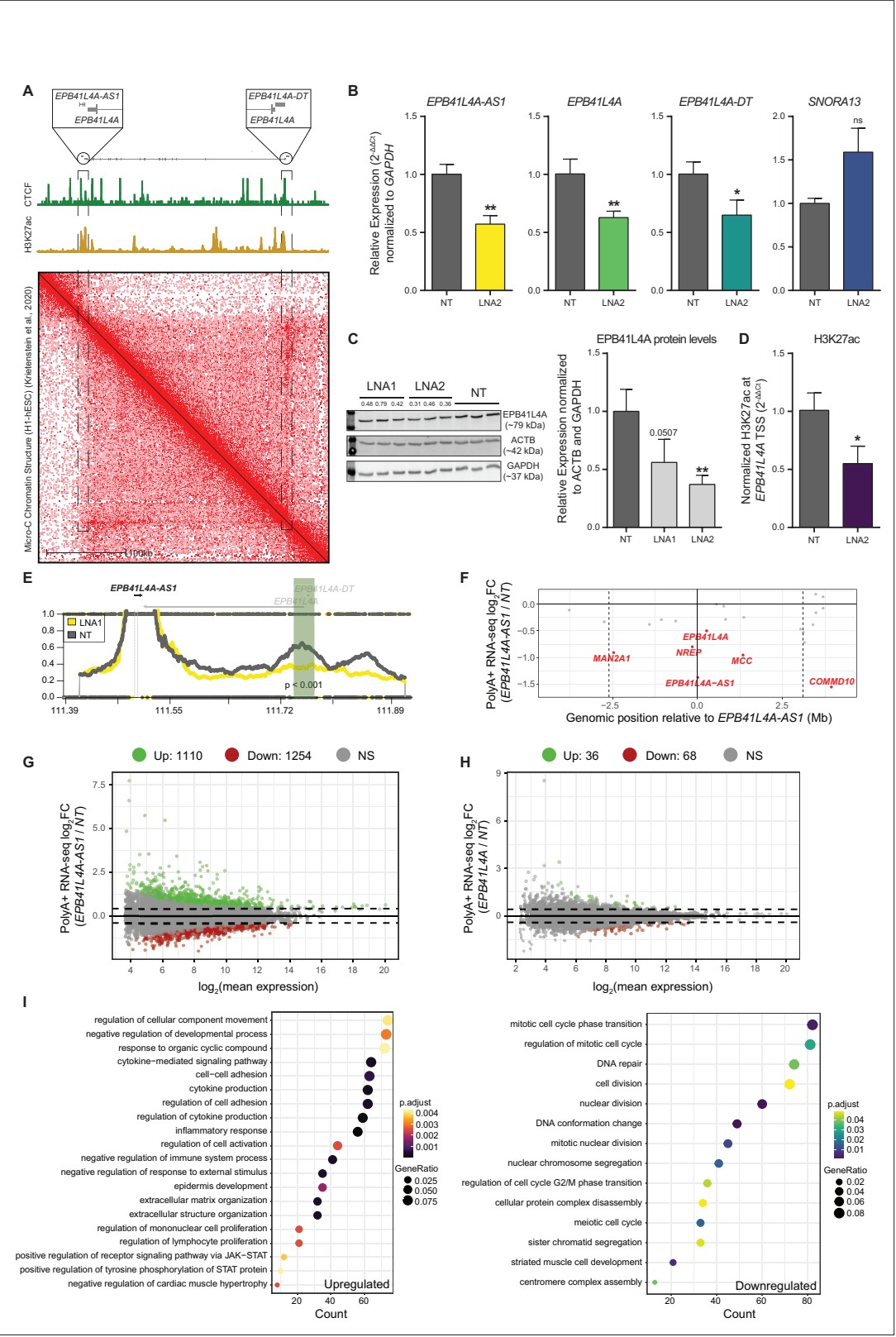

**Figure 2.** *EPB41L4A-AS1* is a *cis*-acting lncRNA affecting genome-wide gene expression. (**A**) (Top) In scale view of the *EPB41L4A-AS1-EPB41L4A* locus, with zoomed areas corresponding to the two TSSes. (Middle) CTCF and H3K27ac coverage across this region. (Bottom) Micro-C data in H1-hESCs (*Krietenstein et al., 2020*) show continuous contacts throughout the *EPB41L4A* gene body. (**B**) RT-qPCR and (**C**) Western blot for the indicated

*Figure 2 continued on next page*

*Figure 2 continued*

genes and proteins following *EPB41L4A-AS1* KD. (**D**) H3K27ac CUT&RUN-qPCR following *EPB41L4A-AS1* KD with GapmeRs using primers targeting the promoter of *EPB41L4A*. (**E**) UMI-4C contact profiles in control and LNA1-transfected cells using baits targeting the TSS of *EPB41L4A-AS1*. The green area represents the quantified genomic interval, and the p-value was calculated using a Chi-squared test. (**F**) Changes in gene expression for the genes in the two flanking TADs of the lncRNA in cells transfected with GapmeRs targeting *EPB41L4A-AS1*. The vertical dotted lines represent the TAD boundaries (as assessed by TADmap; *Singh and Berger, 2021*), the continuous vertical line the lncRNA locus and inter-TAD boundary, and the horizontal continuous line a $\log_2$Fold-change equal to 0. The dots represent individual genes, with the significant ones highlighted. (**G**) MA plot showing the changes in genome-wide gene expression in cells transfected with GapmeRs targeting *EPB41L4A-AS1*. (**H**) Same as (**G**), but with GapmeRs targeting *EPB41L4A*. (**I**) GO enrichment analysis for the upregulated (left) and downregulated (right) genes after *EPB41L4A-AS1* KD. All experiments were performed in $n = 3$ biological replicates, except UMI-4C with $n = 2$, with the error bars in the bar plots representing the standard deviation. ns = $p > 0.05$; *$p < 0.05$; **$p < 0.01$; ***$p < 0.001$ (two-sided Student's *t*-test). A gene was considered to be differentially expressed if both adjusted $p < 0.05$ and $|\log_2$Fold-change$| > 0.41$ (corresponding to a change of 33%).

The online version of this article includes the following source data and figure supplement(s) for figure 2:

**Source data 1.** Full Western blot gels of EPB41L4A, ACTB and GAPDH after *EPB41L4A-AS1* KD with LNA1 and LNA2.

**Source data 2.** Full Western blot gels of EPB41L4A, ACTB and GAPDH after *EPB41L4A-AS1* KD with LNA1 and LNA2.

**Figure supplement 1.** EPB41L4A-AS1 unidirectionally facilitates EPB41L4A expression in *cis*.

**Figure supplement 2.** Most of the transcriptomic changes following EPB41L4A-AS1 downregulation are not explained by EPB41L4A.

**Figure supplement 2—source data 1.** Full Western blot gels of CDKN1A, ACTB and GAPDH after transfection with LNA1 and LNA2.

**Figure supplement 2—source data 2.** Full Western blot gels of CDKN1A, ACTB and GAPDH after transfection with LNA1 and LNA2.

**Figure supplement 3.** LNA1 and LNA2 induce a similar transcriptional response.

*supplement 1J*). We conclude that the transcription of the RNA product of *EPB41L4A-AS1* acts unidirectionally and in *cis* to promote the expression of *EPB41L4A*, possibly by ensuring spatial proximity between the TAD boundary and the *EPB41L4A* promoter.

## Genome-wide transcriptional alterations upon depletion of *EPB41L4A-AS1*

To evaluate the global transcriptional response to *EPB41L4A-AS1* levels, we sequenced polyadenylated RNA from cells transfected with *EPB41L4A-AS1*-targeting, *EPB41L4A*-targeting, or control GapmeRs by RNA-seq (*Supplementary file 2*). Because LNA1 induced the expression of the stress-responsive gene *CDKN1A* (p21)—a known offtarget effect of some GapmeR (*Maranon and Wilusz, 2020*; *Lee and Mendell, 2020*; *Lai et al., 2020*; *Figure 2—figure supplement 2A–C*) —we have decided to initially sequence only the cells transfected with LNA2. Consistent with the RT-qPCR data, KD of *EPB41L4A-AS1* reduced *EPB41L4A* expression and also reduced expression of several, but not all other genes in the TADs flanking the lncRNA (*Figure 2F*). Based on these data, *EPB41L4A-AS1* is a significant *cis*-acting activator according to TransCistor (*Dhaka et al., 2024*) (p = 0.005 using the digital mode). The *cis*-regulated genes reduced by *EPB41L4A-AS1* KD included *NREP*, a gene important for brain development, whose homolog was downregulated by genetic manipulations of regions homologous to the lncRNA locus in mice (*Salnikov et al., 2024*). Depletion of *EPB41L4A-AS1* thus affects several genes in its vicinity.

Downregulation of *EPB41L4A-AS1* led to broad changes in gene expression beyond the proximal TADs (*Figure 2—figure supplement 2D*). A total of 2364 genes were differentially expressed (1110 up, 1254 down, adjusted p < 0.05 and $|\log_2$Fold-change$| > 0.41$) (*Figure 2G*), in contrast to only 104 (36 up, 68 down) when *EPB41L4A* was targeted (*Figure 2H*). A total of 25 (69%) of the upregulated and 22 (32%) of the downregulated genes after *EPB41L4A* KD were also regulated in the same direction by the KD of *EPB41L4A-AS1* (*Figure 2—figure supplement 2E*, p < $10^{-15}$ for both). In agreement with studies on *EPB41L4A* function (*Ishiguro et al., 2000*; *Comiskey et al., 2020*), these genes were

enriched with those functioning in cell-cell adhesion and cell surface receptor signaling, although with a borderline significance (*Figure 2—figure supplement 2F*). Most of the consequences of the depletion of *EPB41L4A-AS1* are thus not directly explained by changes in *EPB41L4A* levels. An additional *trans*-acting function for *EPB41L4A-AS1* would be consistent with its high expression levels compared to most lncRNAs detected in MCF-7 (*Figure 2—figure supplement 2G*). To strengthen these findings, we have transfected MCF-7 cells with LNA1 and a second control GapmeR (NT2), as well as the previous one (NT1) and LNA2, and sequenced the polyadenylated RNA fraction as before. Notably, the expression levels (in FPKMs) of the replicates of both control samples are highly correlated with each other (*Figure 2—figure supplement 3A*), and the global transcriptomic changes triggered by the two *EPB41L4A-AS1*-targeting LNAs are largely concordant (*Figure 2—figure supplement 3B, C*). Because of this concordance and the cleaner (i.e., no CDKN1A upregulation) readout in LNA2-transfected cells, we focused mainly on these cells for subsequent analyses.

Several GO terms were found to be enriched among both up- and downregulated genes following *EPB41L4A-AS1* KD. Upregulated genes were linked to cell–cell adhesion, inflammatory response, and JAK/STAT/ERK pathway, and the downregulated ones to cell cycle, division, and DNA repair, among others (*Figure 2I*). When considering cellular components, the downregulated genes were associated with ribonucleoprotein and spliceosomal complexes, as well as to ribosome subunit and, interestingly, the nucleolus (*Figure 3—figure supplement 1A, B*). Thus, since *EPB41L4A-AS1* is a snoRNA-host gene (SNHG) and snoRNAs typically localize to and act in the nucleolus, we aimed to investigate whether this lncRNA might interact with some factors that can explain the observed enrichment in nucleolar-associated genes.

## *EPB41L4A-AS1* binds SUB1 and affects the expression of its target genes

In order to identify potential factors that might be associated with *EPB41L4A-AS1*, we inspected protein–RNA-binding data from the ENCODE eCLIP dataset (*van Nostrand et al., 2020*). The exons of the *EPB41L4A-AS1* lncRNA were densely and strongly bound by SUB1 (also known as PC4) in both HepG2 and K562 cells (*Figure 3A*). SUB1 interacts with all three RNA polymerases and was reported to be involved in transcription initiation and elongation, response to DNA damage, chromatin condensation (*Garavís and Calvo, 2017*; *Conesa and Acker, 2010*; *Das et al., 2006*; *Hou et al., 2022*), telomere maintenance (*Dubois et al., 2025*; *Salgado et al., 2024*) and rDNA transcription (*Kaypee et al., 2025*). SUB1 normally localizes throughout the nucleus in various cell lines, yet staining experiments show a moderate enrichment for the nucleolus (source: Human Protein Atlas; here) (*Kaypee et al., 2025*). Another nucleolus-related protein, NPM1, which binds C/D-box snoRNAs and directs 2'-*O*-methylation of rRNAs (*Nachmani et al., 2019*), had a single moderately confident binding site within *EPB41L4A-AS1* intron overlapping *SNORA13* (*Figure 3A*). Interestingly, both *SUB1* and *NPM1* mRNAs were significantly downregulated following *EPB41L4A-AS1* KD (*Figure 3B*).

We hypothesized that loss of *EPB41L4A-AS1* might affect SUB1, either via the reduction in its expression or by affecting its functions. We stratified SUB1 eCLIP targets into confidence intervals, based on the number, strength, and confidence of the reported binding sites. Indeed, eCLIP targets of SUB1 (from HepG2 cells profiled by ENCODE) were significantly downregulated following *EPB41L4A-AS1* KD in MCF-7, with more confident targets experiencing stronger downregulation (*Figure 3C*). Importantly, this still holds true when controlling for gene expression levels (*Figure 3—figure supplement 1C*), suggesting that this negative trend is not due to differences in baseline expression. To obtain SUB1-associated transcripts in MCF-7 cells, we performed a native RNA immunoprecipitation followed by sequencing of polyA+ RNAs (RIP-seq) (*Figure 3D*, *Figure 3—figure supplement 1D, E*). As expected, *EPB41L4A-AS1* precipitated with SUB1, as did other top hits from the eCLIP data (*Figure 3—figure supplement 1D*). A total of 1470 genes were enriched in the IP fraction (adjusted p < 0.05 and |log$_2$Fold-change| > 0.41) (*Figure 3—figure supplement 1E*), and our MCF-7 RIP data was in agreement with that of the eCLIP in HepG2 cells (*Figure 3—figure supplement 1F*). Similarly to what was done with the eCLIP data, we divided the RIP-defined targets into confidence intervals based on the significance and overall enrichment in the IP fraction. Strikingly, SUB1-associated transcripts were downregulated upon *EPB41L4A-AS1* KD (*Figure 3E*), similar to what was observed for the eCLIP-based targets. Since SUB1 is also a transcriptional regulator associated with chromatin, we performed CUT&RUN to uncover the genome-wide chromatin occupancy of SUB1 in

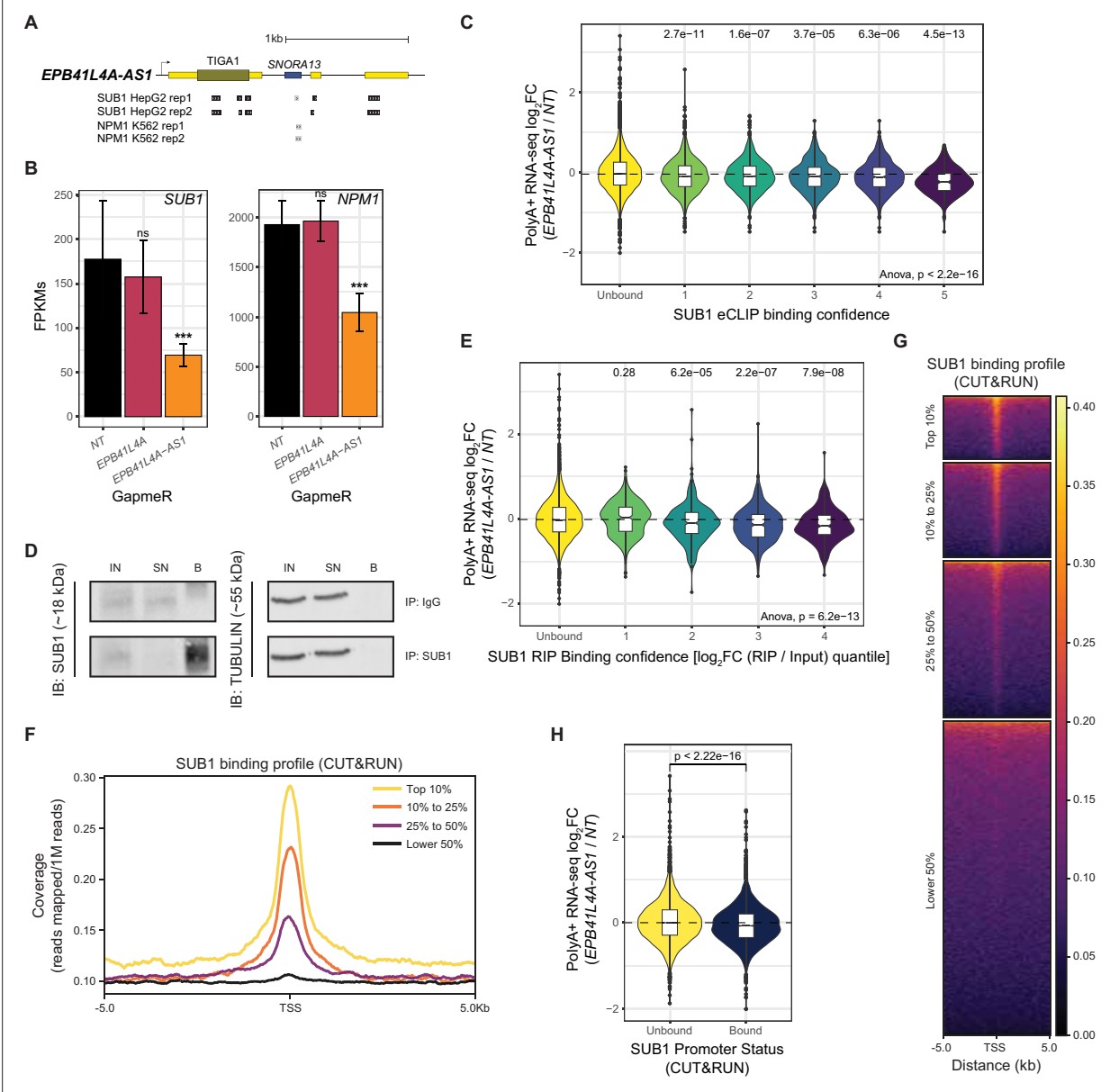

**Figure 3.** SUB1 interacts with *EPB41L4A-AS1* and affects gene expression at the DNA and RNA levels. (**A**) Schematics of the *EPB41L4A-AS1* locus with tracks depicting the eCLIP peaks for both SUB1 and NPM1 (source: ENCODE). (**B**) Average expression levels (in FPKMs) of *SUB1* and *NPM1* in cells transfected with GapmeRs targeting either *EPB41L4A-AS1* or *EPB41L4A*, with the error bars representing the standard deviation across the *n* = 3 replicates. DESeq2 adjusted p-values compared to control GapmeR are also reported. (**C**) Changes in gene expression upon *EPB41L4A-AS1* KD of the genes ranked by the SUB1 eCLIP-binding confidence. (**D**) Western blot following RIP using either a SUB1 or IgG antibody. TUBULIN was used as a negative control, IN—input, SN— supernatant/unbound, B—bound. (**E**) Same as (**C**), but with genes ranked by their enrichment in the RIP data (log2FC RIP/Input). (**F**) Metagene profile around TSS of the normalized SUB1 CUT&RUN signal, stratified by gene expression levels in MCF-7 cells. (**G**) Heatmap of SUB1 reads coverage around the TSSs of all human genes, stratified by gene expression levels in MCF-7 cells. (**H**) Changes in gene expression upon *EPB41L4A-AS1* KD with GapmeRs for genes with and without a high-confidence SUB1 peak in their TSS. All experiments were performed in *n* = 3 biological replicates. In the boxplots, the thick line, edges of the box, and whiskers represent the median, first and third quartiles, and the upper and lower 1.5 interquartile ranges (IQRs), respectively. Outliers (observations outside the 1.5 IQRs) are drawn as single points, the significance of the different comparisons was computed by a Mann–Whitney test, and a global ANOVA p-value is also reported. In all cases, ns = p > 0.05; *p < 0.05; **p < 0.01; ***p < 0.001.

The online version of this article includes the following source data and figure supplement(s) for figure 3:

**Source data 1.** Full Western blot gels for SUB1 and TUBULIN after SUB1 or IgG pulldown.

**Source data 2.** Full Western blot gels for SUB1 and TUBULIN after SUB1 or IgG pulldown.

**Figure supplement 1.** SUB1 is both a chromatin-associated and an RNA-binding protein.

MCF-7 cells (*Supplementary file 3*). SUB1 binding was found to be enriched around TSSs of active genes (*Figure 3F, G*), and among its 835 high-confidence genomic peaks, 418 (~50%) overlapped with the promoters of 411 genes. Of these genes, transcripts from only 56 were also bound by SUB1 at the RNA level, suggesting that SUB1 largely targets distinct gene sets at the DNA and RNA levels. Genes with promoter occupancy of SUB1 were enriched for biological processes related to genome maintenance, stress response, and nuclear transport (*Figure 3—figure supplement 1G*), which is in line with previous studies (*Garavís and Calvo, 2017*; *Conesa and Acker, 2010*; *Das et al., 2006*; *Hou et al., 2022*). These genes were also slightly, yet highly significantly downregulated following *EPB41L4A-AS1* KD (*Figure 3H*). Taken together, *EPB41L4A-AS1* interacts with SUB1, and its KD led to both a decrease in SUB1 mRNA level and the concomitant downregulation of SUB1 targets on chromatin and at the RNA level, positioning this lncRNA as a modulator of SUB1 activity.

## Global de-regulation of snoRNAs and histone mRNA expression upon loss of *EPB41L4A-AS1*

Since many classes of noncoding RNAs—including those localizing to the nucleolus, such as snoRNAs—are not polyadenylated, we next sequenced total rRNA-depleted RNAs from *EPB41L4A-AS1* KD and control MCF-7 cells. As expected, using this approach, we could quantify various RNA species lacking polyA tails, such as small nuclear RNAs (snRNAs), snoRNAs, and histone mRNAs (*Figure 4A*), and could also better quantify intronic expression. Consistent with a *cis*-acting regulation at the transcriptional level, both the exonic and intronic reads of *EPB41L4A* were significantly reduced after *EPB41L4A-AS1* downregulation (*Figure 4—figure supplement 1A*). When considering different transcript classes, we observed a marked upregulation of most snRNAs and snoRNAs and a significant downregulation of histone mRNAs (*Figure 4A*). To quantify SNHG de-regulation at higher resolution, we created a custom annotation in which we divided each SNHG into exonic regions, the snoRNA-encoding region, regions in the snoRNA-containing introns that are found upstream (pre-) and downstream (post-) of the snoRNA, and other snoRNA-less introns (Other) of the SNHG transcripts, while considering the various isoforms annotated for each SNHG (see Methods) (*Figure 4B*). We then quantified the levels of each region in control and KD conditions. Intriguingly, while SNHG exons and unrelated introns were not affected by *EPB41L4A-AS1* KD, read coverage was increased over both the snoRNAs and the flanking intronic regions (*Figure 4C*). These changes were shared among all the snoRNA classes (*Figure 4—figure supplement 1*) and were concordant within individual introns when both (pre- and post-) had enough coverage to be quantified (*Figure 4—figure supplement 1C*). Notably, the effect sizes of these changes were small, and they were not evident when plotting metagenes of these regions—that show the median coverage level—which is close to zero even in rRNA-depleted data (*Figure 4—figure supplement 1D*), therefore suggesting that snoRNA processing remains an efficient process even in cells depleted of *EPB41L4A-AS1*. Therefore, most of the effect we observe is likely in the increased accumulation of the mature and stable snoRNAs, eventually hinting that *EPB41L4A-AS1* perturbations specifically affect the snoRNA production and/or stability rather than SNHG transcription.

To test if the effects on snoRNA biogenesis are related to the changes in SUB1 expression and/or activity, we sequenced the rRNA-depleted RNA from MCF-7 cells in which *SUB1* was depleted by siRNAs (*Figure 4—figure supplement 1E, F*). We identified a total of 243 upregulated and 154 downregulated genes (*Figure 4—figure supplement 1G*), of which 41 (~17%) and 74 (~48%) were concordantly affected by *EPB41L4A-AS1* depletion, and the magnitudes of the genome-wide responses were moderately, but significantly correlated (Spearman's $R = 0.138$, $p < 10^{-15}$). Strikingly, *SUB1* KD led to a similar increase in snoRNA levels and a decrease in histone mRNAs (*Figure 4D, E* and *Figure 4—figure supplement 1H*). Interestingly, the changes in snoRNA expression were highly concordant with those driven by the KD of *EPB41L4A-AS1* (*Figure 4F*, $R = 0.383$) and intronic regions flanking the snoRNAs were also increased, suggesting that *EPB41L4A-AS1* and SUB1 affect the abundance of the same pool of snoRNAs in MCF-7 cells through likely a shared mechanism.

Lastly, we tested whether the effects of *EPB41L4A-AS1* perturbation could be a generic outcome of a noncoding SNHG (ncSNHG) perturbation. To this end, we used GapmeRs to deplete *GAS5*, another abundant SNHG which encodes 11 snoRNAs within its introns (*Figure 4—figure supplement 2A, B*). As expected from the various functions associated with this ncSNHG (*Mourtada-Maarabouni et al., 2008*; *Schneider et al., 1988*; *Mourtada-Maarabouni et al., 2009*; *Yacqub-Usman et al., 2015*; *Kino*

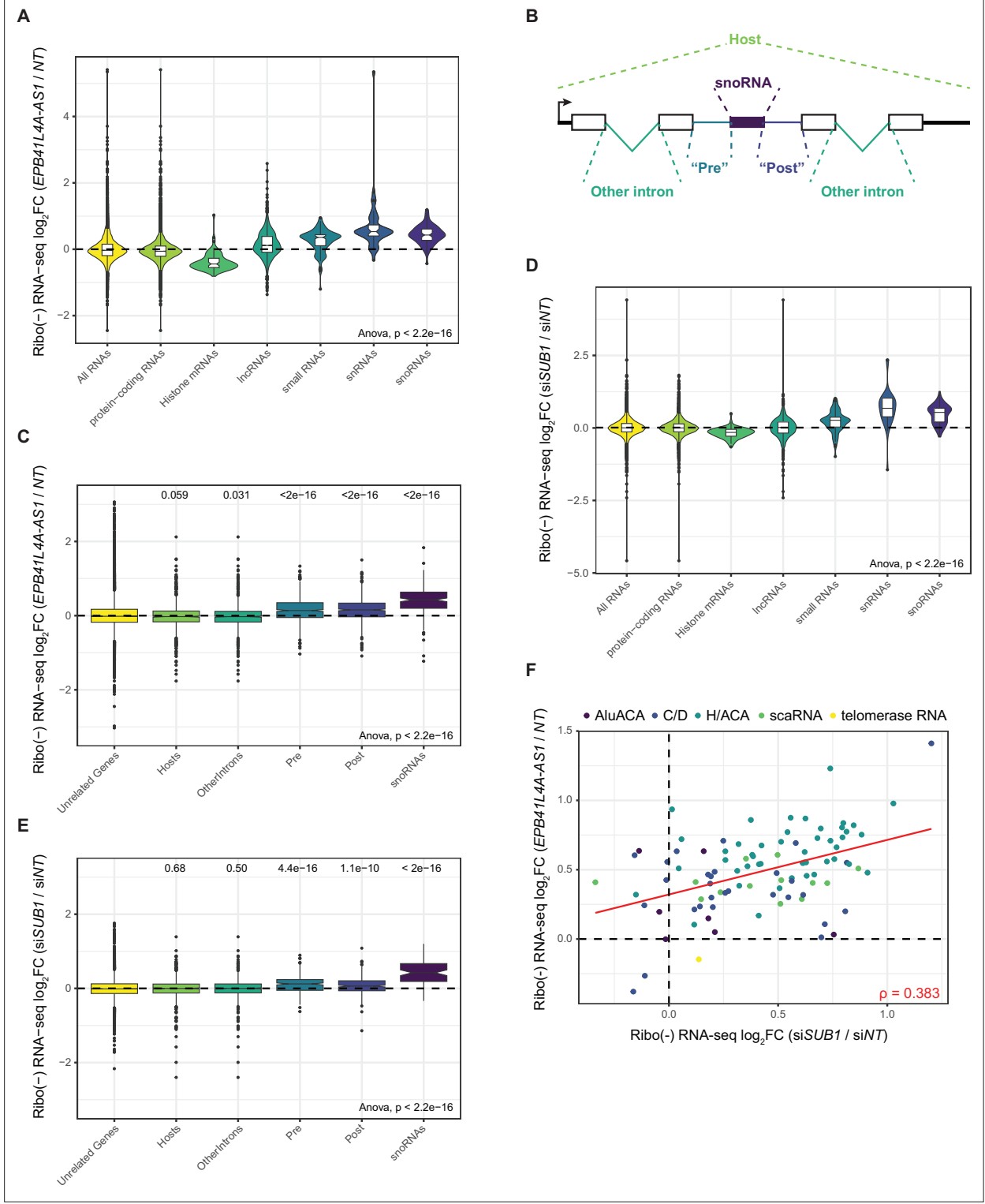

**Figure 4.** *EPB41L4A-AS1* and SUB1 depletion results in a widespread accumulation of mature snoRNAs. (**A**) Changes in gene expression upon *EPB41L4A-AS1* KD with GapmeRs of the indicated RNA classes. (**B**) Schematics depicting the different regions which were separately quantified in each SNHG. (**C**) Changes in RNA-seq read coverage in different regions upon *EPB41L4A-AS1* KD with GapmeRs. (**D**) As in (**B**) for *SUB1* KD with siRNAs of the indicated RNA classes. (**E**) As in (**C**) for *SUB1* KD. (**F**) Correspondence between changes in snoRNA expression after *EPB41L4A-AS1* and *SUB1* KD. The color indicates the different snoRNA classes, and Spearman's correlation coefficient is shown. All experiments were performed in *n* = 3 biological replicates. In the boxplots, the thick line, edges of the box, and whiskers represent the median, first and third quartiles, and the upper and lower

*Figure 4 continued on next page*

*Figure 4 continued*

1.5 interquartile ranges (IQRs), respectively. Outliers (observations outside the 1.5 IQRs) are drawn as single points, the significance of the different comparisons was computed by a Mann–Whitney test, and a global ANOVA p-value is also reported.

The online version of this article includes the following source data and figure supplement(s) for figure 4:

**Figure supplement 1—source data 1.** Full Western blot gels of SUB1 and TUBULIN after *SUB1* KD with siRNAs.

**Figure supplement 1—source data 2.** Full Western blot gels of SUB1 and TUBULIN after *SUB1* KD with siRNAs.

**Figure supplement 1.** EPB41L4A-AS1 and SUB1 depletion affect the expression of different classes of snoRNAs.

**Figure supplement 2.** GAS5 depletion does not affect snoRNAs expression.

*et al., 2010*; *Zhang et al., 2013*; *Hu et al., 2014*; *Sun et al., 2017*; *He et al., 2015*), *GAS5* KD led to substantial changes in gene expression (*Figure 4—figure supplement 2C*), including a marked suppression of histone mRNA transcription, in line with its anti-proliferative function (*Figure 4—figure supplement 2D*). However, snoRNAs or their host introns were largely unaffected (*Figure 4—figure supplement 2D–F*), suggesting that the effect of *EPB41L4A-AS1* on snoRNA expression is not a common feature of ncSNHGs.

## Depletion of *EPB41L4A-AS1* affects SUB1 distribution in the nucleus

The results described so far point to *EPB41L4A-AS1* being important for SUB1 function. To test whether *EPB41L4A-AS1* is important for its proper intracellular distribution, we immunostained for SUB1 and NPM1 in MCF-7 cells transfected with either control or *EPB41L4A-AS1*-targeting GapmeRs. A substantial change in subcellular localization could be observed for both proteins (*Figure 5A, B*). Specifically, SUB1 normally displays a pan-nuclear localization, sometimes slightly enriching into DAPI-low areas corresponding to the nucleoli (*Figure 5A, C*). Upon *EPB41L4A-AS1* depletion, SUB1 shifts toward a stronger nucleolar localization while retaining a presence throughout the nucleus. NPM1 instead regularly shows enrichment inside the nucleoli; however, in GapmeR-transfected cells, this is accompanied by the appearance of globular aggregates within the nucleus (*Figure 5B and D*), reminiscent of the previously reported nucleolar damage/stress response (*Potapova et al., 2023*; *Yang et al., 2016*).

To test if the observed patterns of changes in SUB1 and NPM1 distribution are triggered by nucleolar stress, we exposed MCF-7 cells to CX-5461, an RNA polymerase I inhibitor that is currently being evaluated as a therapeutic agent for solid tumors with DNA repair deficiencies (*Haddach et al., 2012*; *Xu et al., 2017*; *Mars et al., 2020*; *Hilton et al., 2022*). This led to a similar NPM1 pattern as that triggered by *EPB41L4A-AS1* depletion as early as 1 hr after treatment with the higher dose, and the change was more pronounced in later time points (*Figure 5—figure supplement 1A, B*). SUB1 localization also changed, but only at the last time point (24 hr) (*Figure 5—figure supplement 1C, D*). Lastly, we tested whether SUB1 depletion (*Figure 5E*) could affect NPM1 distribution, and indeed *SUB1* KD via siRNAs made NPM1 assume a similar pattern to that caused by *EPB41L4A-AS1* loss (*Figure 5F, G*), suggesting that SUB1 acts upstream of NPM1 and *EPB41L4A-AS1* might modulate this regulation.

## *EPB41L4A-AS1* perturbations do not affect steady-state levels of *SNORA13*

A recent study reported an independent function for *SNORA13*, the snoRNA hosted by *EPB41L4A-AS1* in ribosomal assembly and prevention of nucleolar stress in BJ fibroblasts and mice (*Cheng et al., 2024*). We therefore wondered if the effects we observe in our experimental system result from changes in expression levels of this snoRNA. *SNORA13* levels did not significantly change in RT-qPCR analysis of perturbed cells (*Figure 2B*), or in rRNA-depleted RNA-seq data (*Figure 6A*). We also performed a Northern blot for *SNORA13*, which also did not indicate changes in expression level or changes in the RNA size of *SNORA13* upon GapmeR-mediated depletion of *EPB41L4A-AS1* (*Figure 6B*). To complement these data, we assessed the levels of the 18S:1248m$^1$acp$^3\Psi$ modification, which includes a pseudouridylation guided by *SNORA13*. We used an aminocarboxyl propyl reverse transcription (aRT)-PCR assay as previously described (*Babaian et al., 2020*) and found no difference in HinfI cleavage between control cells and those transfected with either *EPB41L4A-AS1*- or *EPB41L4A*-targeting GapmeRs (*Figure 6C*), suggesting that the MACP modification is not significantly affected

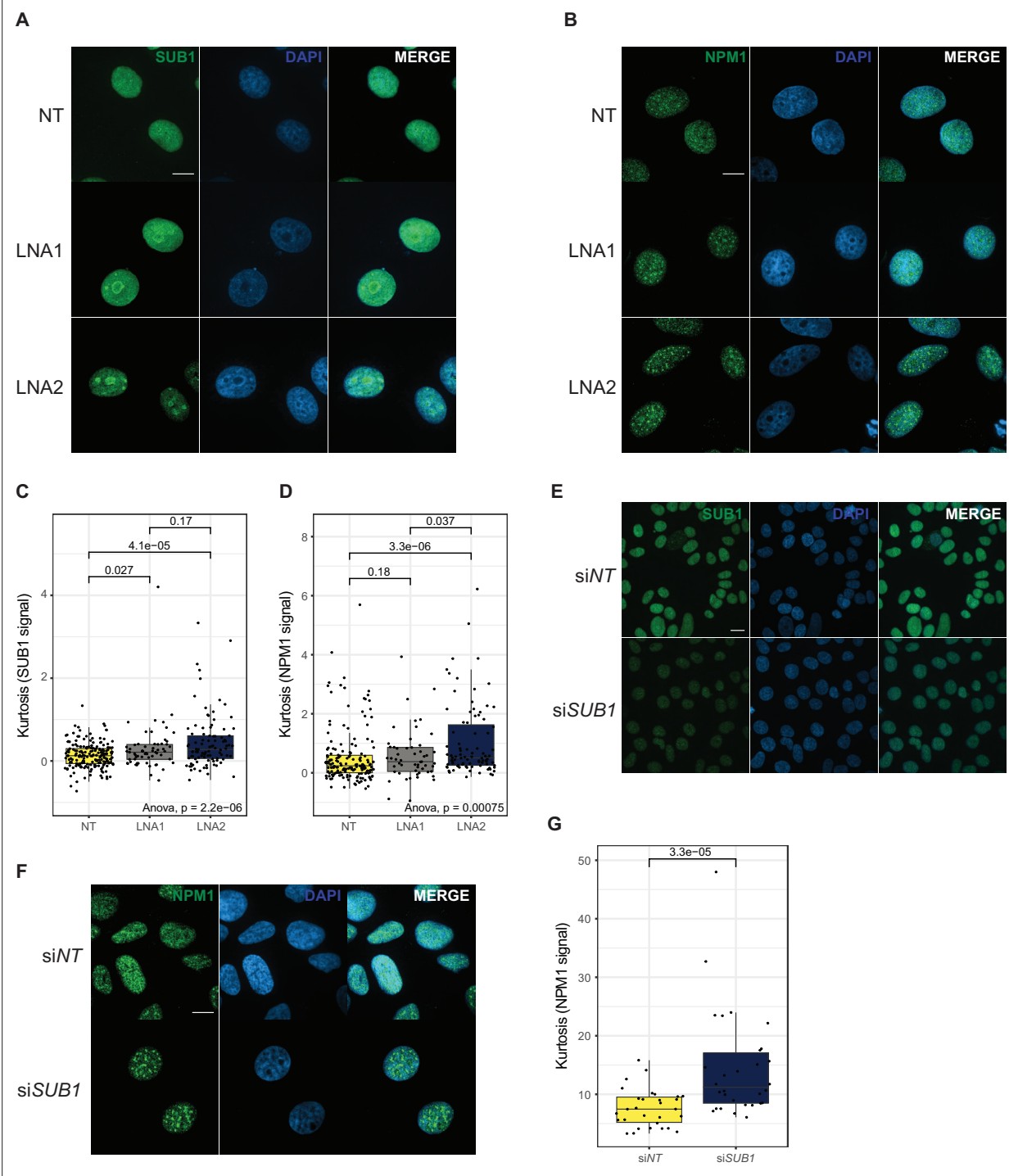

**Figure 5.** Loss of *EPB41L4A-AS1* results in an altered nuclear patterning of SUB1 and NPM1. (**A**) Representative immunofluorescence images for SUB1 after *EPB41L4A-AS1* depletion with two distinct GapmeRs (scale bar = 20 μm, ×100 magnification). (**B**) Same as in (**A**), but for NPM1. (**C**) Quantification of the kurtosis of SUB1 nuclear signal in the indicated conditions. (**D**) Same as in (**C**), but for NPM1. (**E**) Representative immunofluorescence images for SUB1 after *SUB1* KD with siRNAs (scale bar = 20 μm, ×60 magnification). (**F**) Same as in (**E**), but for NPM1. (**G**) Same as in (**D**), but after *SUB1* depletion. All experiments were performed in *n* = 3 biological replicates. In the boxplots, the thick line, edges of the box, and whiskers represent the median, first and third quartiles, and the upper and lower 1.5 interquartile ranges (IQRs), respectively. Outliers (observations outside the 1.5 IQRs) are drawn as single points, the significance of the different comparisons was computed by a Mann–Whitney test, and a global ANOVA p-value is also reported. In each boxplot, points represent individual measurements (cell nuclei).

The online version of this article includes the following figure supplement(s) for figure 5:

**Figure supplement 1.** Nucleolar stress induced by CX-5461 treatment affects SUB1 and NPM1 nuclear patterns.

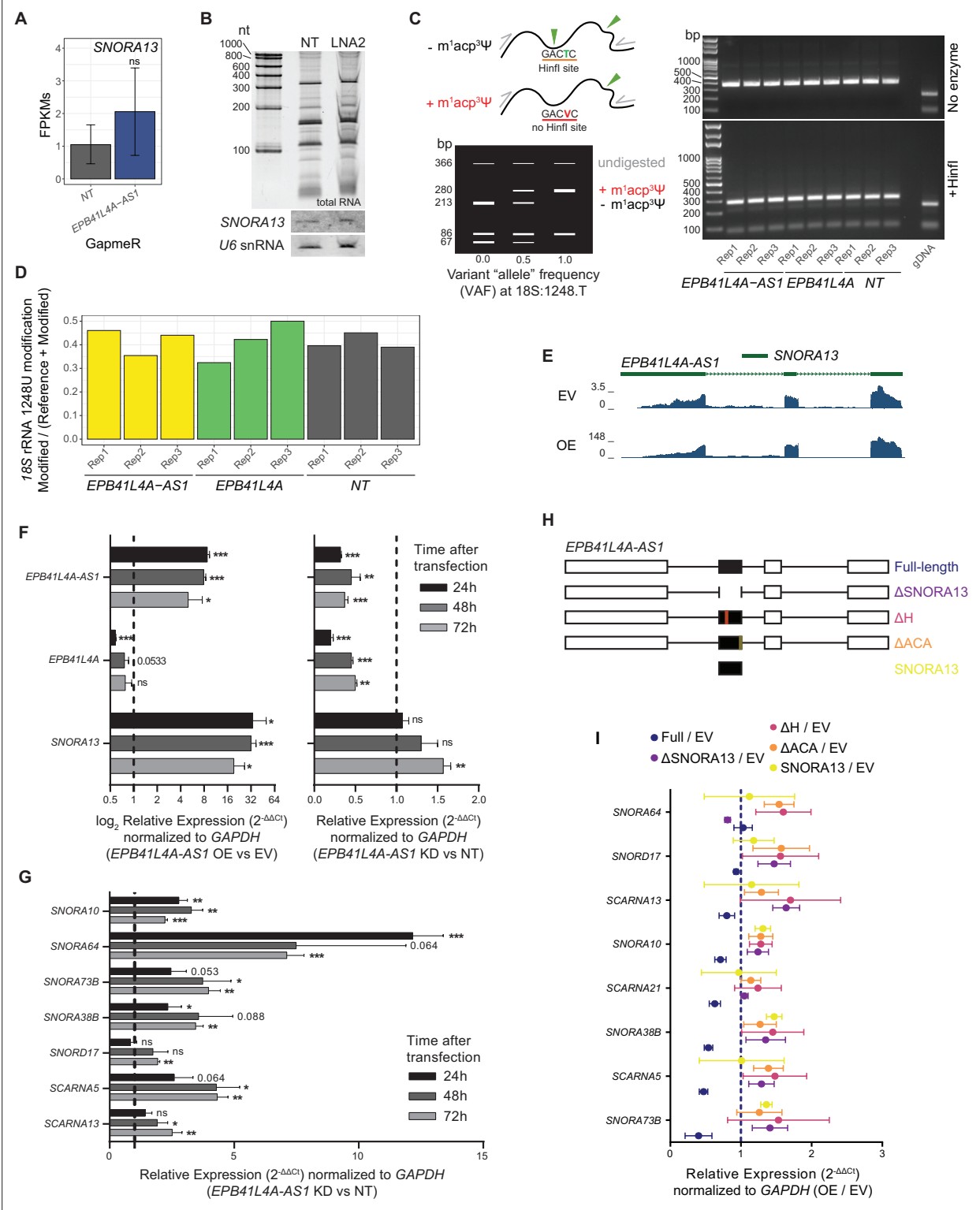

**Figure 6.** *EPB41L4A-AS1*, but not the expression of *SNORA13*, is required for proper snoRNAs expression. (**A**) Average expression levels (in FPKM) of *SNORA13* in cells transfected with GapmeRs targeting *EPB41L4A-AS1* (rRNA-depleted RNA-seq), with the error bars representing the standard deviation across the three replicates. DESeq2 adjusted p-values compared to control GapmeR are also reported. (**B**) Northern blot for *SNORA13* in cells transfected with GapmeRs targeting *EPB41L4A-AS1*. *U6* was used as a loading control, and total RNA stain is also shown on top. (**C**) Schematics of the expected fragment sizes following HinfI digestion in the aRT-PCR assay (left), and agarose gel following HinfI digestion of the PCR-amplified 18S rRNA in cells transfected with GapmeRs targeting either *EPB41L4A-AS1* or *EPB41L4A* (right). No enzyme and genomic DNA (gDNA) were used as negative

*Figure 6 continued on next page*

*Figure 6 continued*

controls. (**D**) Allele frequency at 18S:1248 in the polyA+ RNA-seq dataset, using rRNA reads. (**E**) UCSC Genome Browser view of the *EPB41L4A-AS1* locus, with tracks showing the rRNA-depleted RNA-seq coverage of *EPB41L4A-AS1* OE and control cells. (**F**) RT-qPCR for the indicated genes upon *EPB41L4A-AS1* KD with GapmeRs (right) or overexpression with a full-length unspliced vector (left) over the course of 3 days post-transfection. (**G**) RT-qPCR for the indicated snoRNAs upon *EPB41L4A-AS1* KD with GapmeRs over the course of 3 days post-transfection. (**H**) Schematics of the different *EPB41L4A-AS1* and *SNORA13* overexpressing vectors used in the rescue experiments. (**I**) The ratio between changes in gene expression detected by RT-qPCR in cells transfected with an *EPB41L4A-AS1* or *SNORA13* overexpressing vector vs control cells. All experiments were performed in $n = 3$ biological replicates, with the error bars in the bar plots and forest plots representing the standard deviation. In all cases, ns = $p > 0.05$; *$p < 0.05$; **$p < 0.01$; ***$p < 0.001$ (two-sided Student's *t*-test).

The online version of this article includes the following source data for figure 6:

**Source data 1.** Full agarose gels of the MACP quantification and full northern blot gels for total RNA, *SNORA13* and *U6* after *EPB41L4A* KD with LNA2.

**Source data 2.** Full agarose gels of the MACP quantification and full northern blot gels for total RNA, *SNORA13* and *U6* after *EPB41L4A* KD with LNA2.

as well. In addition, variant calling at 18S:1248 exploiting the residual rRNA reads in our polyA+ RNA-seq dataset revealed no distinction in allele frequency between control and transfected cells (*Figure 6D*), concordant with the aRT-PCR results. In conclusion, we show that in our experimental system—MCF-7 cells transiently transfected with GapmeRs targeting its host—we do not observe changes in steady-state *SNORA13* levels nor abundance of the only known RNA modification driven by this snoRNA. Therefore, we postulate that the observed endophenotypes are likely due to changes in the *EPB41L4A-AS1* lncRNA levels and its cooperation with SUB1.

## A full-length, intron- and *SNORA13*-containing *EPB41L4A-AS1* is required to partially rescue snoRNA expression

We reasoned that if *EPB41L4A-AS1* indeed regulates snoRNA expression, this function likely occurs in *trans*, because none of the genes in the vicinity of the *EPB41L4A-AS1* locus are known to be involved in snoRNA biogenesis or stability. To test this, we performed rescue experiments in which we co-transfected lncRNA-targeting GapmeRs with different GapmeR-resistant *EPB41L4A-AS1* variants expressed from a plasmid driven by a constitutive CMV promoter. We confirmed that the transfected vector is well spliced (*Figure 6E*) and results in an increase in both *EPB41L4A-AS1* and *SNORA13* expression (*Figure 6F*, left).

We then assessed both KD dynamics and the abundance of snoRNAs selected based on their sensitivity to *EPB41L4A-AS1* KD in the RNA-seq data at 24, 48, and 72 hr after transfection. *EPB41L4A* expression decreased concomitantly with that of *EPB41L4A-AS1*, in line with the transcription of the lncRNA regulating this gene in *cis* (*Figure 6F*, right). The expression of most *EPB41L4A-AS1*-sensitive snoRNAs increased as early as 24 hr after transfection, and they generally remained upregulated throughout the next 2 days (*Figure 6G*). We then designed four different *EPB41L4A-AS1* expression vectors to attempt a rescue: (1) full length, including introns; (2) full length, including introns but without *SNORA13* (Δ*SNORA13*); (3) full length, including introns but with either the H or ACA *SNORA13* boxes deleted (ΔH and ΔACA); and (4) *SNORA13* alone (*Figure 6H*). We then compared gene expression between cells transfected with the overexpressing vectors (OE) or an empty control (EV). We were able to obtain a partial rescue of snoRNA expression only when co-transfecting the full-length *EPB41L4A-AS1*, and not any other vector not expressing any or just a mutated *SNORA13* (*Figure 6I*), indicating that the *SNORA13* sequence is involved in the process. However, the expression of a stand-alone *SNORA13,* using the expression vector used and validated previously (*Cheng et al., 2024*), was incapable of driving any rescue of the snoRNAs abundance (*Figure 6I*). This suggests that the entire *EPB41L4A-AS1* transcript is required to counteract *EPB41L4A-AS1* knockdown, most likely because the intact *SNORA13* sequence is necessary to undergo the characteristic processing steps of a SNHG.

## *EPB41L4A-AS1* affects RNA metabolism of both the genes found in *cis* and of snoRNAs

To distinguish between the effects of *EPB41L4A-AS1* depletion on RNA synthesis and RNA decay, we performed SLAM-seq (*Herzog et al., 2017*) using ribo-depleted total RNA in *EPB41L4A-AS1* KD cells (*Supplementary file 4*). Following 48 hr after control or *EPB41L4A-AS1*–targeting GapmeRs

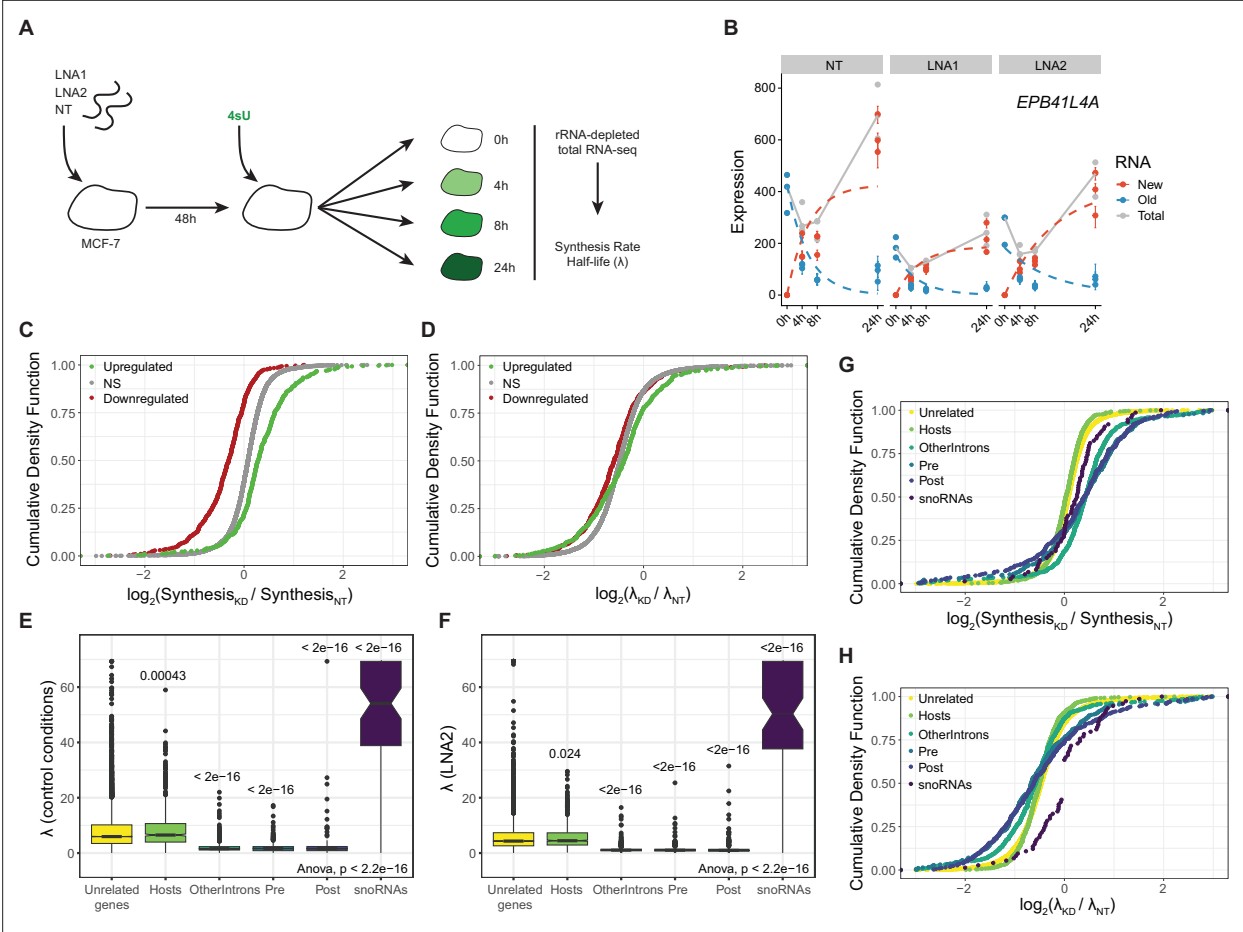

**Figure 7.** The increased abundance of snoRNAs is primarily due to their hosts' increased transcription and stability. (**A**) Workflow of the SLAM-seq experiment. MCF-7 cells were transfected with the indicated GapmeRs for 48 hr, after which the media was replaced with media containing 4sU and the cells were harvested at different time points. (**B**) Fitted model depicting the synthesis and decay rates of the *EPB41L4A* mRNA upon *EPB41L4A-AS1* KD with GapmeRs. (**C**) Changes in synthesis rate upon *EPB41L4A-AS1* KD with GapmeRs for the indicated group of genes. (**D**) Same as in (**C**), but for changes in half-lives. (**E**) Half-lives in control conditions of the indicated SNHG regions as described in *Figure 4B*. p-values refer to the comparison with the unrelated genes. (**F**) Same as in (**E**), but after *EPB41L4A-AS1* KD with LNA2. (**G**) Changes in synthesis rate upon *EPB41L4A-AS1* KD with GapmeRs for the indicated SNHG regions. (**H**) Same as in (**G**), but for changes in half-lives. All experiments were performed in *n* = 3 biological replicates. In the boxplots, the thick line, edges of the box, and whiskers represent the median, first and third quartiles, and the upper and lower 1.5 interquartile ranges (IQRs), respectively. Outliers (observations outside the 1.5 IQRs) are drawn as single points, the significance of the different comparisons was computed by a Mann–Whitney test, and a global ANOVA p-value is also reported. In all cases, ns = p > 0.05; *p < 0.05; **p < 0.01; ***p < 0.001.

The online version of this article includes the following figure supplement(s) for figure 7:

**Figure supplement 1.** EPB41L4A-AS1 affects local RNA metabolism.

transfection, we collected RNA at 0, 4, 8, and 24 hr post-addition of 4sU to the media, while replenishing fresh 4sU at each time point (see Methods) (*Figure 7A*). *EPB41L4A* showed decreased synthesis rate and unchanged half-life (*Figure 7B*), as expected from a gene regulated at the level of transcription. As a group, genes upregulated after *EPB41L4A-AS1* KD in the regular steady-state RNA-seq data had substantially higher synthesis rates, and those downregulated had lower synthesis rates, when compared to unchanged ones (p = 3.25 × 10⁻¹⁴ for the upregulated genes and p = 0.002 for the downregulated genes, Mann–Whitney test, *Figure 7C*), whereas only a minor difference was observed in terms of half-lives (p = 0.011 for the upregulated genes and p = 1.82 × 10⁻⁸ for the downregulated genes, Mann–Whitney test, *Figure 7D*). This suggests that the *trans*-regulated genes are also mostly regulated at the level of their transcription. Additionally, when comparing the effects across genes in the two neighboring TADs (Cis), other TADs on the same chromosome (Chr5), or other chromosomes (Trans), the genes proximal to *EPB41L4A-AS1* specifically exhibit a reduced synthesis rate compared

to the others (*Figure 7—figure supplement 1A*), but no changes in their half-lives (*Figure 7—figure supplement 1B*).

Compared to their hosts and other genes, snoRNAs were, as expected, relatively stable in both KD and control conditions (*Figure 7E, F*), while the intronic regions exhibited high turnover. When comparing *EPB41L4A-AS1* KD to the control GapmeRs, a slight increase in the synthesis rates of different parts of SNHGs was observed. The snoRNA half-life is difficult to measure accurately due to their relative stability and their short length, which means that the number of incorporated 4sU that can be detected is much lower than that in longer transcripts. With these caveats in mind, there appeared to be a relative increase in snoRNA stability compared to a reduction observed in other genes and other parts of the SNHGs (*Figure 7G, H*). Altogether, this indicates that the increase of snoRNA levels driven by *EPB41L4A-AS1* knockdown is primarily due to the increased transcription of their hosts, with a possible additional component of relative stabilization.

## EPB41L4A-AS1 loss affects cell proliferation and invasion

Lastly, we aimed to assess the phenotypic consequences of *EPB41L4A-AS1* loss. As mentioned above, the genes affected by the KD were found to be enriched for GO terms related to cell adhesion, movement, and division (*Figure 2I*). Thus, we measured cell migration and proliferation by performing wound healing and cell counting assays in cells with reduced *EPB41L4A-AS1* levels. Cells transfected with *EPB41L4A-AS1* targeting GapmeRs exhibited significantly higher migration rates, compared to controls (*Figure 8A, B*), with cells bordering the wounded region displaying a flattened and translucent morphology (*Figure 8A*). This higher migration is not due to higher cellular proliferation/division, as cells with reduced lncRNA levels proliferate less (*Figure 8C*), and this deficiency could be rescued by co-transfecting a vector encoding the GapmeR-resistant, full-length *EPB41L4A-AS1* (*Figure 8C*), suggesting that this modulation takes place in *trans*.

These findings suggest that proper *EPB41L4A-AS1* expression is required for cellular proliferation, whereas its deficiency results in the onset of more aggressive and migratory behavior, likely linked to the increase of the gene signature of epithelial-to-mesenchymal transition (EMT) (*Figure 8D*). Because EMT is not characterized by a unique gene expression program and rather involves distinct and partially overlapping gene signatures (*Youssef et al., 2024*), we checked the expression level of marker genes linked to different types of EMTs (*Figure 8E*). The most upregulated gene in *Figure 8D* is *TIMP3*, a matrix metallopeptidase inhibitor associated with a particular EMT signature that is less invasive and more profibrotic (EMT-T2) (*Youssef et al., 2024*). Interestingly, we observed a stark upregulation of other genes linked to EMT-T2, such as *TIMP1*, *FOSB*, *SOX9*, *JUNB*, *JUN*, and *KLF4*, whereas *MPP* genes (linked to EMT-T1, which is highly proteolytic and invasive) are generally downregulated or not expressed. This suggests that the downregulation of *EPB41L4A-AS1* is primarily linked to a specific EMT program (EMT-T2), and future studies aimed at uncovering the exact mechanisms and relevance will shed light upon a possible therapeutic potential of this lncRNA.

## Discussion

We show here that the *EPB41L4A-AS1* gene locus harbors a pleiotropic gene that, within a single biological system—MCF-7 breast cancer cells—performs diverse functions (*Figure 9*). At the site of its own transcription, which overlaps a strong TAD boundary, *EPB41L4A-AS1* is required to maintain expression of several adjacent genes, which are regulated at the level of transcription. Strikingly, the promoter of *EPB41L4A-AS1* ranks in the 99.8th percentile of the strongest TAD boundaries in human H1 embryonic stem cells (*Salnikov et al., 2024*; *Abdennur et al., 2024*). It features several CTCF-binding sites (*Figure 2A*), and in MCF-7 cells, we demonstrate that it blocks the propagation of the 4C signal between the two flanking TADSs (*Figure 1F*). Future studies will help elucidate how *EPB41L4A-AS1* transcription and/or the RNA product regulate this boundary. So far, we found that *EPB41L4A-AS1* did not affect CTCF binding to the boundary, and while some peaks in the vicinity of *EPB41L4A-AS1* were significantly affected by its loss, they did not appear to be found near genes that were dysregulated by its KD (*Figure 7—figure supplement 1C*). We also found that the KD of *EPB41L4A-AS1* for 72 hr—which depletes the RNA product, but may also affect the nascent RNA transcription (*Lee and Mendell, 2020*; *Lai et al., 2020*)—reduces the spatial contacts between the TAD boundary and the *EPB41L4A* promoter (*Figure 2E*, *Figure 2—figure supplement 1J*). Further

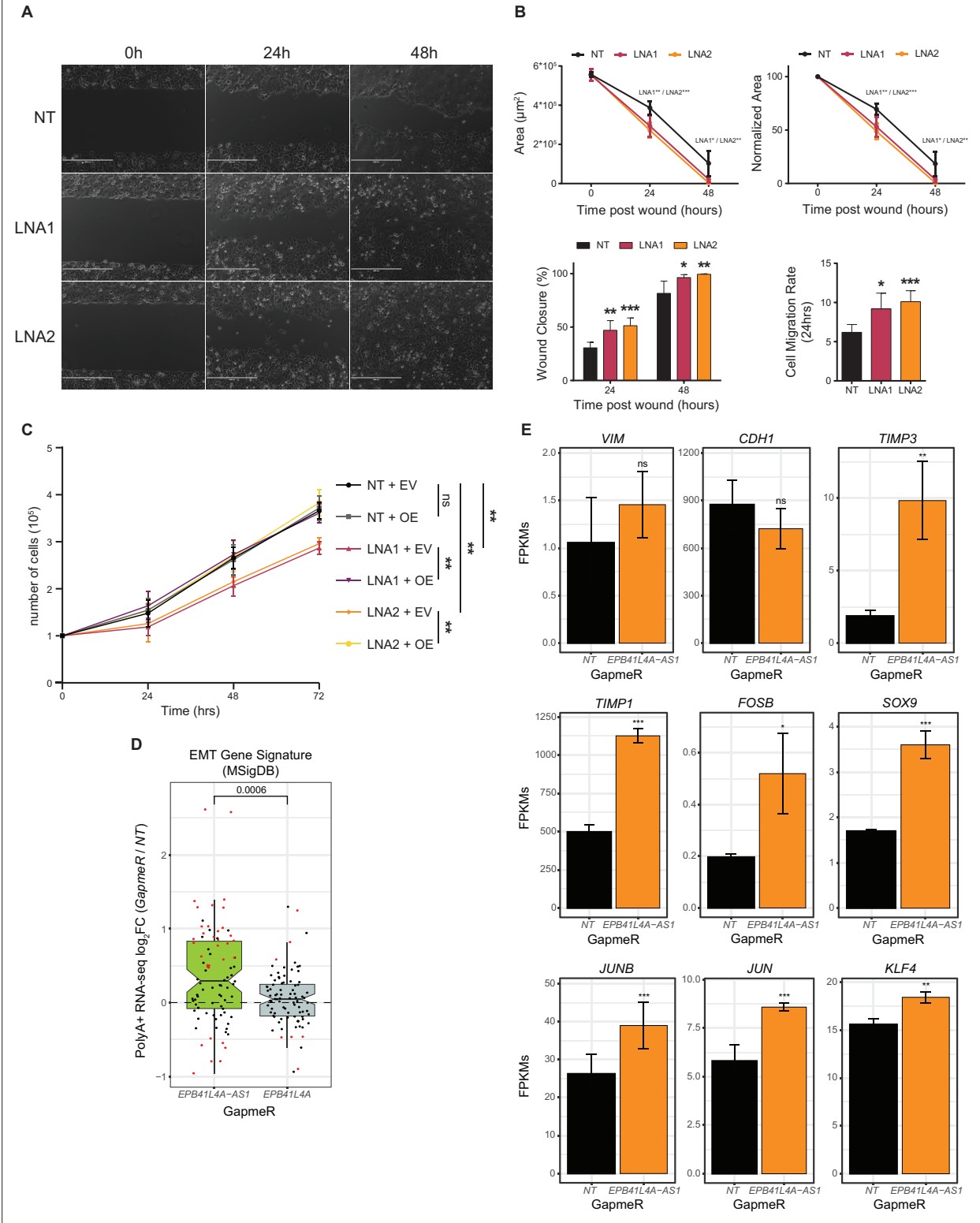

**Figure 8.** Cells with reduced *EPB41L4A-AS1* expression display reduced proliferation and increased invasion capacity. (**A**) Representative brightfield images of the wound at the indicated time points and conditions. (**B**) Wound area (top-left), normalized wound area (top-right), closure percentage (bottom-left) and migration rate (bottom-right) at the indicated time points and conditions. The significance was calculated by a two-sided Student's *t*-test. (**C**) Growth curve of the cells in the indicated conditions over the course of 3 days. The significance was calculated by a two-sided Student's *t*-test. (**D**) Changes in gene expression of the epithelial-to-mesenchymal transition (EMT) signature genes (from MSigDB) after KD with GapmeRs targeting

*Figure 8 continued on next page*

*Figure 8 continued*

either *EPB41L4A-AS1* or *EPB41L4A* (polyA+ RNA-seq data). The significance of the comparison was computed by a Mann–Whitney test. (**E**) Average expression levels (in FPKMs) following *EPB41L4A-AS1* KD with GapmeRs of selected genes that have been previously linked to EMT, with the error bars representing the standard deviation across the *n* = 3 replicates. DESeq2 adjusted p-values compared to control GapmeR are also reported. All experiments were performed in *n* = 3 biological replicates. The error bars in the bar plots represent the standard deviation. In the boxplot, the thick line, edges of the box, and whiskers represent the median, first and third quartiles, and the upper and lower 1.5 interquartile ranges (IQRs), respectively. Outliers (observations outside the 1.5 IQRs) are drawn as single points. The points in the boxplot represent individual genes, and their color indicates whether they were found to be significantly (adjusted p < 0.05 and |log$_2$Fold-change| > 0.41) dysregulated (red) or not (black). In all cases, ns = p > 0.05; *p < 0.05; **p < 0.01; ***p < 0.001 (two-sided Student's *t*-test).

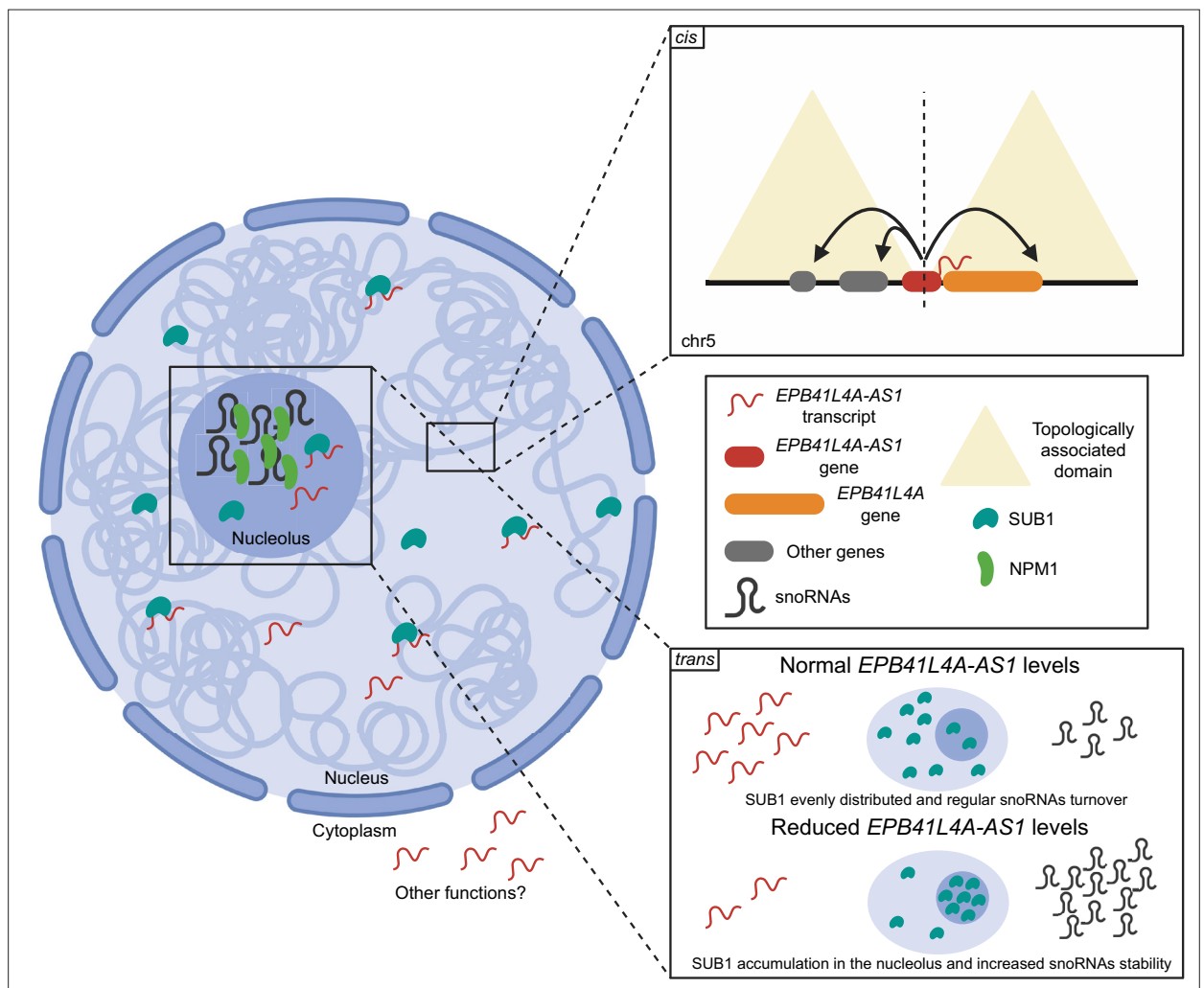

**Figure 9.** Proposed model of the *cis* and *trans* functions of the *EPB41L4A-AS1* lncRNA. Under normal conditions, the *EPB41L4A-AS1* locus is transcribed and ensures the proper expression of several genes located in *cis*. In *trans*, it interacts with SUB1, which eventually is evenly distributed throughout the nucleus. When there are reduced levels of *EPB41L4A-AS1*, the expression of the genes in *cis* is reduced, and SUB1 accumulates in the nucleolus. As a consequence, snoRNA abundance increases, likely due to their increased stability, and MCF-7 cells acquire a more invasive in vitro phenotype. Created with Biorender.com.

The online version of this article includes the following source data and figure supplement(s) for figure 9:

**Figure supplement 1—source data 1.** Full Western blot gels of MTREX and GAODH after *EPB41L4A-AS1* KD with LNA1 and LNA2.

**Figure supplement 1.** EPB41L4A-AS1 depletion consistently represses MTREX.

elucidation of the exact functional entity needed for the *cis*-acting regulation will require detailed genetic perturbations of the locus that are difficult to carry out in the polypoid MCF-7 cells, without affecting other functional elements of this locus or cell survival, as we were unable to generate deletion clones despite several attempts.

A recent study found that *SNORA13* negatively regulates ribosome biogenesis in TERT-immortalized human fibroblasts (BJ-HRAS^G12V) by decreasing the incorporation of RPL23 into the maturing 60S ribosomal subunits, eventually triggering p53-mediated cellular senescence (*Cheng et al., 2024*). In their experimental setting, *SNORA13* was constitutively reduced by either CRISPRi of the *EPB41L4A-AS1* locus for 7 days or its KO leaving the lncRNA sequence and expression unaltered, whereas our approach relies solely on transient knockdown for 48–72 hr that does not appreciably affect *SNORA13* (*Figures 1B, 2B, 6A, B*), or the levels of the RNA modification that it guides (*Figure 6C, D*). Therefore, while both studies implicate *EPB41L4A-AS1* in nucleolar biology and its perturbation in nucleolar stress, there is an apparent discrepancy between our results on the role that *SNORA13* plays. In our hands, the differential expression of snoRNAs and cell proliferation can be rescued by the expression of *EPB41L4A-AS1* with the snoRNA, and not by separate expression of *SNORA13*, suggesting the snoRNA alone is not playing a central role in the observed phenotypes. Also, the study by Cheng et al. showed that activity of the p53 pathway was less active in *SNORA13* knockout cells, with or without induction of oncogenic stress, whereas we see an opposite trend of some upregulation of the p53 pathway upon transient *EPB41L4A-AS1* KD (*Figure 7—figure supplement 1D*). We cannot, however, completely rule out potential contributions of this snoRNA to the phenotypes observed in this study. For instance, the phenotype of snoRNA expression is partially rescued by *EPB41L4A-AS1* overexpression in *trans* only when a full-length, unspliced, and *SNORA13*-containing vector is used, and not when the snoRNA is deleted, mutated, or expressed alone (*Figure 6I*). This would suggest that the processing of *EPB41L4A-AS1* is involved in its mode of action, although further experiments will be needed to address whether *SNORA13* or its DNA sequence is indeed required to regulate the expression of the snoRNAs. We note that compensatory effects are possible in cells that grow for extended periods of time without *SNORA13*.

Beyond its site of transcription, *EPB41L4A-AS1* associates with SUB1, an abundant protein linked to various functions, and these two players are required for the proper distribution of at least two nuclear proteins. Their dysregulation results in large-scale changes in gene expression, including upregulation of snoRNA expression, mostly through increased transcription of their hosts, and possibly through a somewhat impaired snoRNA processing and/or stability. The exact molecular pathways involved in snoRNA biogenesis, maturation, and decay are still not completely understood. One of the genes that most consistently was affected by *EPB41L4A-AS1* KD with GapmeRs is *MTREX* (also known as *MTR4*), that becomes downregulated at both the RNA and protein levels (*Figure 9—figure supplement 1A–C*). Interestingly, MTREX is part of the NEXT and PAXT complexes (*Contreras et al., 2023*) that target several short-lived RNAs for degradation, and the depletion of either *MTREX* or other complex members leads to the upregulation of such RNAs, that include PROMPTs, uaRNAs, and eRNAs, among others. It is therefore tempting to hypothesize a role for *MTREX*-containing complexes in trimming and degrading snoRNA host introns and releasing the mature snoRNAs. Future studies specifically aimed at uncovering novel players in mammalian snoRNA biology will both conclusively elucidate whether *MTREX* is indeed involved in these processes.

Several features of the response to *EPB41L4A-AS1* resemble nucleolar stress, including altered distribution of NPM1 (*Potapova et al., 2023*; *Yang et al., 2016*). SUB1 was shown to be involved in many nuclear processes, including transcription (*Conesa and Acker, 2010*), DNA damage response (*Yu et al., 2016*; *Mortusewicz et al., 2008*), telomere maintenance (*Dubois et al., 2025*), and nucleolar processes including rRNA biogenesis (*Kaypee et al., 2025*; *Tafforeau et al., 2013*). Our results suggest a complex and multi-faceted relationship between *EPB41L4A-AS1* and SUB1, as *SUB1* mRNA levels are reduced by the transient (72 hr) KD of the lncRNA (*Figure 3B*), the distribution of the protein in the nucleus is altered (*Figure 5A, C*), while the protein itself is the most prominent binder of the mature *EPB41L4A-AS1* in ENCODE eCLIP data (*Figure 3A*). The most striking connection between *EPB41L4A-AS1* and SUB1 is the similar phenotype triggered by their loss (*Figure 4*). We note that a recent study has shown that SUB1 is required for Pol I-mediated rDNA transcription in the nucleolus (*Kaypee et al., 2025*). In the presence of nucleolar SUB1, rDNA transcription proceeds as expected, but when SUB1 is depleted or its nucleolar localization is affected—by either sodium

butyrate treatment or inhibition of KAT5-mediated phosphorylation at its lysine 35 (K35)—the levels of the 47S pre-rRNA are significantly reduced. In our settings, SUB1 enriches into the nucleolus following *EPB41L4A-AS1* KD; thus, we might expect to see a slightly increased rDNA transcription or no effect at all, given that SUB1 localizes in the nucleolus in baseline conditions as well. It is, however, difficult to determine which of the connections between these two genes is the most functionally relevant and which may be indirect and/or feedback interactions. For example, it is possible that *EPB41L4A-AS1* primarily acts as a transcriptional regulator of *SUB1* mRNA, or that its RNA product is required for proper stability and/or localization of the SUB1 protein, or that *EPB41L4A-AS1* acts as a scaffold for the formation of protein-protein interactions in which SUB1 is involved.

In the context of cancer, a systematic study of TCGA data, which is consistent with our analysis (*Figure 1—figure supplement 2*), found that *EPB41L4A-AS1* is downregulated in tumors compared to tumor-adjacent normal tissues (*Liao et al., 2019*), in accordance with both the downregulation of the TIGA1 protein (*Liao et al., 2019*) and the reduction in the *SNORA13*-guided rRNA modification (*Babaian et al., 2020*). The association with poor prognosis (*Liao et al., 2019*) is consistent with the observed increased invasiveness that we find in MCF-7 cells. Effects of *EPB41L4A-AS1* on migration were also reported in colorectal cancer cells (*Bin et al., 2021*). The notion that cells with reduced *EPB41L4A-AS1* levels exhibit lower proliferation (*Figure 8C*), yet increased invasion (*Figure 8A, B*), is compatible with a function as an oncogene by promoting EMT (*Figure 8D, E*). Cells undergoing this process may reduce or even completely halt proliferation/cell division, until they revert back to an epithelial state (*Dongre and Weinberg, 2019*; *Brabletz et al., 2018*). Notably, downregulated genes following *EPB41L4A-AS1* KD are enriched in GO terms related to cell proliferation and cell cycle progression (*Figure 2I*), whereas those upregulated for terms linked to EMT processes.

We note that this is not the first instance in which this lncRNA has been investigated. Several recent studies have described the functions of *EPB41L4A-AS1* in various biological systems, highlighting its dysregulation in cancer (*Liao et al., 2019*; *Bin et al., 2021*; *Wang et al., 2020*), diabetes (*Liao et al., 2022*), aging (*Yang et al., 2021*; *Wang et al., 2024*), and Alzheimer's disease (*Wang et al., 2024*). Both p53 and PGC-1α were shown to act upstream of *EPB41L4A-AS1* and regulate its expression (*Liao et al., 2019*). In these different systems, different pathways were described as regulated by this lncRNA, including glycolysis, glutamine metabolism (*Liao et al., 2019*), and autophagy (*Wang et al., 2024*). Mechanistically, it was reported that this lncRNA regulates gene expression via its association with and upregulation of KAT2A (GCN5/GNC5L2) (*Liao et al., 2022*; *Wang et al., 2024*). These prior studies generally relied on candidate-gene or co-expression approaches to identify the genes regulated by *EPB41L4A-AS1* and did not systematically examine its *cis*- and *trans*-acting effects, with the exception of a *cis*-acting regulation that was briefly hinted in a preprint posted 4 years ago (*Samdal et al., 2021*), but not published so far to the best of our knowledge. These studies also did not systematically investigate the *trans*-regulated targets of *EPB41L4A-AS1*, did not separately deplete the nascent and the mature forms of the lncRNA, and did not attempt to decouple its functions from those of *SNORA13*.

## Methods
### Cell culture, transfection, and treatments

MCF-7 cells were cultured in DMEM supplemented with 4.5 g/l of glucose, 4 mM L-glutamine, 10% FBS, 1% PenStrep solution, grown in incubators maintained at 37°C with 5% $CO_2$, and were detached for passaging using a Trypsin (0.05%)–EDTA (0.02%) solution. DNA plasmid, siRNAs (final concentration 25 nM), and GapmeRs (50 nM) transfections were performed by using Lipofectamine 3000 (Thermo Fisher Scientific), according to the manufacturer's protocol. In all cases, cells were plated the day before the transfection in an antibiotic-free medium which was subsequently changed prior to adding the transfection mix, and again after 24 hr to maximize cell viability. All transfections were carried out for 72 hr, unless specified otherwise. The complete list with the sequences of both siRNAs and LNA/GapmeRs is reported in *Supplementary file 5*.

Cell synchronization was performed using a standard double-thymidine block. Briefly, thymidine (Sigma) was added to the culture medium at a final concentration of 2 mM, and the cells were incubated for 18 hr. The medium was then replaced with standard complete medium for 9 hr, followed by the addition of fresh thymidine-containing medium. After an additional 15 hr, the medium was

replaced with standard complete medium to release the cells from the G1/S boundary into the S phase.

LPS was purchased from Sigma, diluted in DNase/RNase-free $H_2O$, and added to the cell culture media at the indicated final concentrations. $H_2O_2$ was purchased from Sigma and added directly to the cell culture media at the indicated final concentrations. Thapsigargin and Etoposide (Sigma) were diluted in tissue culture-grade DMSO and added to the cell culture media at the indicated final concentrations. Cells were cultured with these stressors for a total of 3 days, and cells supplemented with an equal volume of the respective solvent were used as controls. CX-5461 (Sigma) was diluted in sterile-filtered 50 mM $NaH_2PO_4$ at a stock concentration of 10 mM and added to the cell culture media at a final concentration of 0.5, 1, or 5 µM. As a control, cells were supplemented with an equal volume of 50 mM $NaH_2PO_4$.

## Plasmid construction

The sequence corresponding to the cDNA of human *EPB41L4A-AS1* (ENST00000688370.2_3) was synthesized and cloned by Hylabs inside a pcDNA3.1(+) vector. The vector to express the *EPB41L4A* ORF (NM_022140.5) was purchased from Genecopoeia (EX-A8889-Lv201). The vector expressing a full length and unspliced *EPB41L4A-AS1* (pLV[ncRNA]-Puro-CMV) was purchased from Vectorbuilder, and the target site of LNA2 was mutated by introducing four mismatches (AACTTAAAAGCAGCGT → A<u>C</u>CTTA<u>G</u>AA<u>G</u>T<u>AGA</u>GT) via restriction-free cloning. To generate the Δ*SNORA13*, ΔH, and ΔACA vectors, we ordered *EPB41L4A-AS1* gene fragments with the desired mutations from TWIST. These fragments corresponded to the sequence between the unique SalI and MluI restriction sites and were inserted via restriction digestion. The pLKO.1 *SNORA13* plasmid (*Cheng et al., 2024*) was a gift from Prof. Mendell (UT Southwestern). All gRNAs were ordered as single-stranded oligos (Sigma) and cloned inside pKLV-Puro vectors as previously described (*Konermann et al., 2015*). The complete list of gRNAs and their sequences is available in Table S5. All plasmids were sent for Sanger and whole plasmid sequencing (Plasmidsaurus) prior to use.

## Wound healing assay and analysis

Wound healing experiments were carried out as previously described (*Monziani et al., 2025*). To perform the wound healing assay, we employed the µ-Plate 24 Well system (Ibidi). Specifically, we transfected the targeting and control GapmeRs via reverse transfection, diluting the transfection mixes directly into the cell mixtures while seeding the cells. After reaching confluence overnight (O/N), the culture inserts were gently removed with sterile forceps and the wells were washed with 1X PBS. Brightfield images were taken using an EVOS Cell Imaging System (Thermo Fisher Scientific) at different time points. Data analysis was then performed using the Wound_healing_size_tool (*Suarez-Arnedo et al., 2020*) plugin for ImageJ, which calculates the total (µm²) and percentage (%) of the wounded area. From these data, we calculated the rate of cell migration ($R_M$) as:

$$R_M = \frac{W_i - W_t}{t},$$

where $W_i$ represents the initial average wound width, and $W_t$ is the average wound width at a given time $t$. Additionally, we derived the percentage of wound closure defined as:

$$Wound\, Closure\, (\%) = \frac{A_{t=0} - A_{t=\Delta t}}{A_{t=0}},$$

with $A_{t=0}$ and $A_{t=\Delta t}$ being the wounded area (in µm²) at the beginning and after time $t$, respectively.

## Gene expression analysis

RNA was extracted by using TRI-Reagent as previously described (*Monziani et al., 2025*), according to the manufacturer's instructions. The isolated RNA was then resuspended in DNase/RNase-free $H_2O$ and DNase-treated using the Baseline-ZERO DNase (Biosearch Technologies), following the manufacturer's recommendations and incubating the RNAs for 1 hr at 37°C. RNA was then retrotranscribed using the qScript Flex cDNA Synthesis Kit (Quanta Bio), following standard conditions as suggested by the manufacturer. The resulting cDNA was finally diluted 1:5-1:10 with DNase/RNase-free $H_2O$ and used as input for Quantitative Real-Time PCR using the Fast SYBR Green Master Mix (Applied

Biosystems). Each reaction was amplified using the following PCR program: 20" at 95°C, 40 cycles by denaturing for 1" at 95°C followed by annealing/extension for 20" at 60°C. Results were analyzed using the ΔΔCt method. The complete list of primers and their sequences is available in *Supplementary file 5*.

## Aminocarboxyl propyl reverse transcription (aRT)-PCR assay to estimate 18S:1248m$^1$acp$^3$Ψ

The aminocarboxylpropyl reverse transcription (aRT)-PCR assay to detect and quantify the 1248. m1acp3 Ψ modification was performed as previously described (*Babaian et al., 2020*), with minor modifications. Briefly, total RNA was extracted as described above, and 1 μg per sample was DNase-treated using TURBO DNase (Invitrogen), according to the manufacturer's protocol. Retrotranscription was then carried out using SuperScript III (Invitrogen), following the manual's recommendations and by using only random hexamers. The cDNA was then diluted 1:5 with DNase/RNase-free H$_2$O and used for end-point PCR using the Q5 High-Fidelity DNA Polymerase (NEB), according to the standard protocol and using the primers listed in the original protocol (*Babaian et al., 2020*) and available in *Supplementary file 5*. From each PCR reaction, 5 μl were digested in duplicate with HinFI (NEB) in rCutSmart buffer (NEB) and an additional 5 μl were incubated without the enzyme as a negative control. Lastly, the resulting fragments were then run on a 2% agarose gel.

## Northern blot

Northern blot to detect *SNORA13* was performed as previously described (*Cheng et al., 2024*), with minor modifications. A total of 20 μg of RNA per sample was loaded onto an 8% TBE-UREA polyacrylamide gel, which had been pre-equilibrated by running at low voltage in 1X TBE. After electrophoresis, the gel was stained for 15 min in 0.5X TBE with a few drops of ethidium bromide while rocking, to assess the quality of the run. The RNA was then transferred to a HyBond-NX membrane (GE Healthcare) in 0.5X TBE at room temperature (RT) using a constant current of 0.25 A for 2 hr. Following the transfer, the membrane was rinsed in 2X SSC, crosslinked at 1200 J/cm² in a UV oven, and pre-hybridized for 30 min at 42°C in 10 ml of pre-warmed ULTRAhyb Ultrasensitive Hybridization Buffer (Thermo Fisher Scientific) while rotating. IR fluorescent probes targeting *U6* and *SNORA13* were then added to a final concentration of 10 nM, and the membrane was incubated O/N at 42°C while rotating. The next day, the membrane was washed twice in 2X SSC, 0.1% SDS at 42°C for 5 min while rotating, followed by two additional washes in 0.1X SSC, 0.1% SDS, and then finally imaged. The complete list of probes is available in *Supplementary file 5*.

## Protein extraction and Western blot

Proteins were extracted using RIPA as previously described (*Monziani et al., 2025*). Briefly, cells were detached using a 0.05% Trypsin–0.02% EDTA solution, pelleted by centrifugation, resuspended in ice-cold RIPA buffer, and incubated on ice for 20 min with intermittent vortexing. The lysates were then centrifuged at 15,000 × *g* for 15 min at 4°C, after which the supernatants were collected and quantified. For Western blots, equal amounts of protein were loaded onto a polyacrylamide gel and run in an SDS running buffer. Proteins were then transferred onto a pre-activated PVDF membrane in a cold Tris-glycine buffer supplemented with 20% methanol, with a constant current of 0.30 A for 2 hr. Membranes were then blocked in 5% milk in 1X PBS-0.1% Tween 20 for 1 hr, followed by O/N incubation at 4°C with primary antibodies diluted in the same blocking solution. The next day, membranes were washed three times with 1X PBS-0.1% Tween 20, incubated for 2 hr at room temperature with the appropriate secondary antibodies, and washed three more times before image acquisition. The complete list of primary and secondary antibodies and relative dilutions is available in *Supplementary file 5*.

## Immunostaining

Immunofluorescence was performed as previously described (*Monziani et al., 2025*). Cells were plated and transfected on either sterile glass coverslips or 8-well chambers (Ibidi). They were first washed once with 1X PBS, then fixed with cold 4% paraformaldehyde (PFA) in 1X PBS for 15 min at room temperature. After two 5-min washes with 1X PBS, cells were permeabilized by incubating them in a permeabilization/blocking solution containing 5% horse serum, 1 mg/ml BSA, and 0.1% Triton

X-100 in 1X PBS for 30 min at room temperature. Following three washes with 1X PBS, cells were incubated O/N at 4°C with primary antibodies diluted in blocking solution (5% horse serum, 1 mg/ml BSA in 1X PBS). The next day, cells were washed three times with 1X PBS and incubated for 2 hr at room temperature with secondary antibodies in blocking solution, protected from light. After three additional washes with 1X PBS, cells were stained with DAPI and mounted using ProLong Glass Anti-fade Mountant (Invitrogen). The slides were left to cure O/N in the dark at room temperature before imaging with a Zeiss Spinning Disk confocal microscope. The complete list of primary and secondary antibodies and relative dilutions is available in *Supplementary file 5*.

## Subcellular fractionation

To fractionate cells and isolate the cytoplasmic, nucleoplasmic, and chromatin RNA fractions, we employed a published protocol (*Gagnon et al., 2014*) with minor modifications. Confluent 10 cm plates were harvested with a Trypsin (0.05%)–EDTA (0.02%) solution, pelleted by centrifugation, washed once with ice-cold 1X PBS, and resuspended in 380 µl of ice-cold HLB buffer (10 mM Tris-HCl, pH 7.5, 10 mM NaCl, 3 mM $MgCl_2$, 0.3% NP-40, 10% Glycerol) freshly supplemented with RNase Inhibitors (EURX). The samples were then incubated on ice for 20' with occasional vortexing, spun down for 3' at 4°C/1000 × $g$, the supernatant was then transferred to a new tube, centrifuged again for 10' at 4°C/1000 × $g$, after which the supernatant (cytoplasmic fraction) was moved to a new tube, supplemented with 1 ml of RNA precipitation solution (RPS, 150 mM Sodium Acetate in 100% ethanol) and stored at –20°C for at least 1 hr. The semi-pure nuclei were then washed three times with ice-cold HLB and centrifugations for 2' at 4°C/500 × $g$, after which the pellets were resuspended in 380 µl of ice-cold MWS (10 mM Tris-HCl, pH 7.0, 4 mM EDTA, pH 8.0, 0.3 M NaCl, 1 M Urea, 1% NP-40) freshly supplemented with RNase Inhibitors, quickly vortexed and incubated on ice for 5 min, vortexed again and incubated for an additional 10 min. The tubes were then centrifuged for 3' at 4°C/1000 × $g$, the supernatant (nucleoplasmic/nuclear soluble fraction) was saved to a new tube, supplemented with 1 ml of RPS and stored at –20°C for at least 1 hr. The chromatin pellets were finally washed three times with ice-cold MWS and centrifugations for 2' at 4°C/500 × $g$, after which 1 ml of TRI-Reagent was added and samples were stored at –20°C. After 1+ hr at –20°C, the tubes with the cytoplasmic and nucleoplasmic fractions in RPS were briefly vortexed, centrifuged for 15' at 4°C/15,000 × $g$, the pellets were washed with 70% ethanol by vortexing and centrifuged again for 15' at 4°C/15,000 × $g$, after which they were air-dried and resuspended in 1 ml of TRI-Reagent. To each sample in 1 ml of TRI-Reagent, 10 µl of 0.5 M EDTA, pH 8.0 were added followed by an incubation with vortexing for ~10 min at 65°C, or until the RNA pellets fully dissolved. The RNA was then extracted following the standard TRI-Reagent protocol.

## Native RNA immunoprecipitation

To perform Native RNA Immunoprecipitation, we have employed a slightly modified version of the ENCODE IP protocol (*Sundararaman et al., 2016*). Briefly, $5 \times 10^6$ MCF-7 cells were harvested by trypsinization, washed once with 1X PBS, and flash frozen as pellets in 50 µl of 1X PBS. The pellets were then thawed on ice, resuspended in 450 µl of ice-cold lysis buffer (50 mM Tris-HCl, pH 7.4; 100 mM NaCl; 1% NP-40; 0.1% SDS; 0.5% sodium deoxycholate) supplemented with EDTA-free Protease Inhibitor Cocktail (APExBIO) and RNase Inhibitor (EURX), and incubated on ice for 20' with occasional vortexing to allow complete lysis. The lysates were then centrifuged at max speed for 15' at 4°C and the supernatants were collected into a new tube. A total of 20 µl/sample of Protein A/G beads was then washed twice with ice-cold lysis buffer and resuspended in 250 µl of lysis buffer, of which 50 µl were used to pre-clear the lysate by incubation at RT for 30' with gentle rotation, and the remaining 200 µl were instead incubated for 1 hr/RT with 3 µg of antibody. After 30' of preclearing, the clear lysate was moved to a new tube in ice, and 25 µl (5%) of it was saved as INPUT sample. After 1 hr, the antibody/beads were washed three times with ice-cold lysis buffer and incubated with the pre-cleared lysate O/N at 4°C while gently rotating. The following day, 25 µl (5%) of the supernatant was saved as UNBOUND sample, and the beads (BOUND sample) were gently washed twice with ice-cold wash buffer (20 mM Tris-HCl pH 7.4, 10 mM MgCl2, 0.2% Tween-20) freshly supplemented with RNase Inhibitors (EUR$_X$). Finally, 500 µl of TRI-Reagent was added to all three fractions and RNA isolation was carried out as previously described. The complete list of antibodies and relative dilutions is available in *Supplementary file 5*.

## Single-molecule RNA FISH with HCR technology (smFISH HCR)

To visualize the subcellular localization of EPB41L4A-AS1 in vivo, we performed single-molecule fluorescence in situ hybridization (smFISH) using HCR amplifiers. Probe sets (*n* = 30 unique probes) targeting *EPB41L4A-AS1* and *GAPDH* (positive control) were designed and ordered from Molecular Instruments. We followed the Multiplexed HCR v3.0 protocol with minor modifications. MCF-7 cells were plated in 8-well chambers (Ibidi) and cultured O/N as described above. The next day, cells were fixed with cold 4% PFA in 1X PBS for 10 min at RT and then permeabilized O/N in 70% ethanol at –20°C. Following permeabilization, cells were washed twice with 2X SSC buffer and incubated at 37°C for 30 min in hybridization buffer (HB). The HB was then replaced with a probe solution containing 1.2 pmol of *EPB41L4A-AS1* probes and 0.6 pmol of *GAPDH* probes in HB. The slides were incubated O/N at 37°C. To remove excess probes, the slides were washed four times with probe wash buffer at 37°C for 5 min each, followed by two washes with 5X SSCT at RT for 5 min. The samples were then pre-amplified in amplification buffer for 30 min at RT and subsequently incubated O/N in the dark at RT in amplification buffer supplemented with 18 pmol of the appropriate hairpins. Finally, excess hairpins were removed by washing the slides five times in 5X SSCT at RT. The slides were mounted with ProLong Glass Antifade Mountant (Invitrogen), cured O/N in the dark at RT, and imaged using a Nikon CSU-W1 spinning disk confocal microscope. In order to estimate the RNA copy number, we imaged multiple distinct fields, extracted the number of *EPB41L4A-AS1*/*GAPDH* molecules in each field using the 'Find Maxima' tool in ImageJ/Fiji, and divided them by the number of cells (as assessed by DAPI staining).

## Cleavage Under Targets and Release Using Nuclease (CUT&RUN)

CUT&RUN reactions were performed following the V3 of the protocol (*Meers et al., 2019*), with minor modifications which were described previously (*Monziani et al., 2025*). An equal number of cells was harvested and washed once with 1X PBS at room temperature. After three washes with Wash Buffer (20 mM HEPES-NaOH pH 7.5, 150 mM NaCl, 0.5 mM Spermidine supplemented with EDTA-free Protease Inhibitor Cocktail (APExBIO)) at RT, cells were resuspended in 1 ml Wash Buffer and incubated for 10 min while gently rotating with 20 µl of pre-activated Concanavalin A-coated magnetic beads (EpiCypher). The buffer was then removed, and the beads were resuspended in 150 µl of Antibody Buffer (Wash Buffer with 0.1% Digitonin (Sigma) and 2 mM EDTA, pH 8) containing the antibody of interest and left gently rotating O/N with a 180° angle at 4°C. On the next day, cells were washed two times with ice-cold Dig-Wash Buffer (Wash Buffer with 0.1% Digitonin), resuspended in 150 µl of Dig-Wash Buffer supplemented with 1 µl of custom-made pA/G-MNase every mL and left rotating for 1 hr at 4°C. After that, cells were washed again two times with Dig-Wash Buffer, resuspended in 100 µl of the same buffer, and chilled into a thermoblock sitting on ice. To initiate the cleavage reaction, 2 µl of 100 mM $CaCl_2$ was added to each tube, and the tubes were left at 0°C for 30 min. To halt the reaction, 100 µl of a 2X STOP Buffer (340 mM NaCl, 20 mM EDTA pH 8, 4 mM EGTA pH 8, 0.05% Digitonin, 100 µg/ml RNAse A, 50 µg/ml Glycogen) were added to each tube, and cleaved DNA fragments were released by incubating the samples for 30' at 37°C. The tubes were then centrifuged for 5 min at 4°C/16,000 × *g* and the supernatant was collected. Finally, the DNA was purified by standard Phenol/Chloroform extraction using the 5PRIME Phase Lock Gel Heavy tubes (QuantaBio), and the success of the CUT&RUN reaction was assessed by running the positive control (H3K27me3 or H3K4me3) on a Tapestation (Agilent Technologies) using a High Sensitivity D1000 ScreenTape (Agilent Technologies). The complete list of primary and secondary antibodies and relative dilutions is available in *Supplementary file 5*.

## RNA-seq and CUT&RUN Library Preparation

RNA-seq libraries were prepared using the CORALL mRNA-Seq Kit V2 (Lexogen), following the manufacturer's instructions. Prior to library generation, 1 µg of total RNA was either PolyA-enriched using the Poly(A) RNA Selection Kit V1.5 (Lexogen), or depleted of rRNAs using the RiboCop rRNA depletion kit for HMR V2 (Lexogen). To prepare the CUT&RUN libraries, we followed the original protocol (*Meers et al., 2019*), with slight modifications which were described previously (*Monziani et al., 2025*). The libraries were then quality-checked by both dsDNA Qubit (Thermo Fisher Scientific) and Tapestation (Agilent Technologies). All libraries were sequenced on an Illumina NovaSeq 6000 or

Novaseq X instrument, aiming for either 10 million (CUT&RUN) or 15 million (RNA-seq) reads per sample.

## RNA-seq data analysis

Analysis of the RNA-seq libraries was performed as previously described (*Monziani et al., 2025*). Raw FASTQ files were processed by a customized Lexogen CORALL analysis pipeline script (https:// github.com/Lexogen-Tools/corall_analysis; *GrecVict and TomasDrozd, 2025*). Briefly, adaptors were trimmed by using Cutadapt (*Martin, 2011*), and the resulting trimmed FASTQ files were used as input for mapping to the hg19 human genome with STAR aligner (*Dobin et al., 2013*), using the '--quantMode GeneCounts' option in order to count the number of reads mapping to each feature provided by an appropriate GTF file. To call for differential expression, we used DESeq2 (*Love et al., 2014*), considering a gene to be differentially expressed when adjusted p < 0.05 and absolute log$_2$FC >0.41 (~33% increase or decrease). Prior to any subsequent analysis, we filtered out poorly expressed genes, pseudogenes, and genes with an exonic length <200 nt. To perform GO enrichment analysis and GSEA, we used the ClusterProfiler R package (*Wu et al., 2021*) by considering a term to be enriched when the adjusted p < 0.05. Whenever several redundant terms were found, we employed the *simplify* function with a cutoff =0.7 to reduce the redundancy.

## CUT&RUN data analysis

Raw FASTQ files were aligned to the hg19 reference genome with Bowtie2 (*Langmead and Salzberg, 2012*), using the -p 8, -X 2000 and --no-unal options. Peaks were called using the MACS2 call-peak function (*Zhang et al., 2008*). Peak quantification and differential expression were performed using the HOMER tool suite (*Heinz et al., 2010*), using the *annotatePeaks.pl* and *getDiffExpression. pl* functions, respectively. Lastly, to plot the metagene profiles and coverage heatmaps, we employed deepTools (*Ramírez et al., 2016*).

## Computational screen to identify putative *cis*-acting lncRNAs

We reasoned that lncRNAs that are likely to act in *cis* will be occasionally co-regulated with their target genes and also found in spatial proximity to the target promoters. We therefore sought such lncRNA–target pairs using the GeneHancer database (*Fishilevich et al., 2017*) that connects between regulatory elements and their targets by combining tissue co-expression data, ChIP-seq of enhancer-enriched histone modifications and transcription factor binding, Hi-C data, and eQTLs within putative enhancer regions. We complemented this by analyzing several RNA-seq datasets of MCF-7 breast cancer cells exposed to different conditions (*Sun et al., 2015*; *Nagarajan et al., 2020*; *Janky et al., 2014*), to infer genome-wide transcriptional alterations across a variety of different experimental settings. We then focused on those cases in which a lncRNA (1) overlaps a high-confidence Gene-Hancer (GH) element, (2) is differentially expressed in at least one of the conditions analyzed, and (3) the targets of the aforementioned GH element are also concomitantly differentially expressed. Because this approach identified over 600 candidate lncRNA-target pairs (*Supplementary file 1*), we first discarded lncRNAs that have been already reported to act in *cis* and prioritized those cases in which the lncRNA is well expressed and/or the target genes are biologically interesting. This led us to focus on four lncRNA-target pairs connected in GeneHancer and in which both genes were significantly either upregulated or downregulated in the treatment condition, compared to controls (*Figure 1—figure supplement 1A-D*).

## Chromosome Conformation Capture (3C)

Between 5 and 10 × 10$^6$ cells for each condition/replicate were collected and washed once with 1X PBS at RT. Cells were then resuspended in 10 ml of ice-cold 1X PBS with 10% FBS, supplemented with another 10 ml of 1X PBS with 10% FBS and 4% Formaldehyde, and left gently rotating for 5' at RT. Quenching was achieved by adding 1020 µl 2.5 M Glycine to each tube and placing them on ice to cool down, after that the cells were centrifuged for 8' at 4°C/300 × *g* and washed once with ice-cold 1X PBS. The pellet was then resuspended in 10 ml of Nuclear Permeabilization Buffer (10 mM Tris-HCl, pH 8.0, 10 mM NaCl, 0.5% NP-40) supplemented with EDTA-free Protease Inhibitor Cocktail (APExBIO), gently rotated for 1 hr in a cold room and centrifuged for 5' at 4°C/600 × *g*. After discarding the supernatant, the nuclei were resuspended in ~1 ml of ice-cold 1X PBS, moved to a

1.5-ml centrifuge tube, centrifuged and washed again with 1 ml ice-cold 1X PBS, and eventually frozen as a pellet at –80°C.

After 1+ days at –80°C, each pellet was thawed on ice and gently resuspended in 175.5 µl of nuclease-free H$_2$O, after which 24.5 µl of 10X DpnII Reaction Buffer was added and the tubes were moved to a thermomixer set at 37°C. Each reaction was then supplemented with 3 µl of pre-warmed 20% SDS, incubated for 1 hr at 37°C/900 × $g$, supplemented again with 20 µl of pre-warmed 20% Triton X-100 and incubated for an additional 1 hr at 37°C/900 × $g$, after which 15 µl of DpnII (NEB, 50.000 U/ml) were added and the reaction was finally incubated O/N at 37°C/900 × $g$. The following day, the restriction reaction was removed by centrifugation for 5' at 4°C/600 × $g$, and the pellets were washed once with ice-cold 1X PBS, eventually leaving a small volume (~30 µl) on top of the pellet. The tubes were then incubated for 20' at 65°C to inactivate DpnII activity, followed by thorough resuspension in 445 µl of nuclease-free H$_2$O and were transferred to ice. After chilling, 50 µl of 10X Ligation Buffer (NEB), 0.5 µg BSA, and 5 µl of Quick Ligase (NEB) were added to each tube and left incubating O/N at 16°C. After the O/N incubation, the ligation reaction was removed by centrifugation for 5' at 4°C/600 × $g$, the pellets were washed once with ice-cold 1X PBS and eventually resuspended in 250 µl 10 mM Tris-HCl, pH 7.5. A total of 30 µg of RNase A was then added to each sample, which was subsequently incubated for 45' at 37°C/750 $g$, after which 15 µl of PCR-grade Proteinase K was added and incubated O/N at 65°C/750 × $g$. The following day, the de-crosslinked reactions (3C template) were cleaned with 0.5X Sera-Mag Size Selection Beads (Cytiva Life Sciences), according to the manufacturer's instructions, eluted in 200 µl Elution Buffer (QIAGEN) and quantified by dsDNA Qubit.

## UMI Circular Chromosome Conformation Capture (UMI-4C)

Quantitative UMI-4C was performed as previously described (*Schwartzman et al., 2016*), with minor modifications. An equal amount of 3C template (5–10 µg) for each sample/replicate was diluted to a final volume of 200 µl with Elution Buffer and sonicated to an extent the fragments size ranges between 250 and 1000 bp. To the sonicated fragments, 20 µl of 10X End-Repair Buffer (NEB) and 10 µl of End Repair Mix (NEB) were added and the samples were incubated for 30' at 20°C, after which they were cleaned with 2.2X Sera-Mag Selection Beads and eluted in 78 µl 10 mM Tris-HCl, pH 8.0. 76 µl of the cleaned DNA, 10 µl 10X NEBuffer 2, 10 µl 10 mM dATP, and 4 µl Klenow (3'–5' exo, NEB) were added, and the reaction was incubated for 30' at 37°C. After this, the enzymes were inactivated by incubating the samples for 20' at 75°C, then 2 µl of FastCIP (NEB) were added and the reactions were further incubated for 1 hr at 50°C, followed by purification with 2X Sera-Mag Size Selection Beads and elution in 67 µl of 10 mM Tris-HCl, pH 8.0. Illumina-compatible adapters were ligated by adding 80 µl Quick Ligase Buffer (NEB), 10 µl Quick Ligase (NEB), and 4 µl of 15 µM Y-Shaped Adaptors (IDT) to 66 µl of the eluted DNA from the previous step, followed by an incubation at 25°C for 15'. After that, the samples were denatured for 2' at 95°C, placed on ice, brought to volume by adding 160 µl of nuclease-free H$_2$O, cleaned with 1X Sera-Mag Size Selection Beads, and eluted in 40 µl 10 mM Tris-HCl, pH 8.0. To quantify the resulting 4C template, 1 µl was diluted in 4 µl of 10 mM Tris-HCl, pH 8.0, denatured for 5' at 95°C, placed on ice, and analyzed by ssDNA Qubit.

Libraries were then prepared using a nested PCR approach, preparing five reactions for each sample and employing three bait primers for each target (EPB41L4A-AS1 TSS, EPB41L4A TSS, STARD4 TSS, see next for design criteria) to increase complexity of the 4C profiles. For the first PCR (PCR1), upstream bait primers were designed close (<80–85 bp) to a DpnII restriction site (hg19 genome build); for the second PCR (PCR2), the downstream bait primers were designed to follow—but not overlap—the upstream ones, ending ideally <20 bp from the DpnII site ('pad' sequence), and containing a sequence consisting of the Universal Illumina P5 on their 5' end. Both primers were designed to have a standard annealing temperature of ~60°C, and their sequences are available in *Supplementary file 5*. PCR1 consisted of 200 ng of 4C template, 10 µl of 10X Phusion Buffer (NEB), 1 µl of 10 mM dNTPs, 2.5 µl of pooled upstream bait primers with a final concentration of 10 µM, 2.5 µl of 10 µM Universal Illumina P7 primer, 0.5 µL Phusion Polymerase (NEB), and nuclease-free H$_2$O up to a final volume of 50 µl. The reactions were incubated for 30" at 98°C, amplified with 20 cycles consisting of 10" at 98°C, 30" at 56°C and 60" at 72°C, with a final extension of 10' at 72°C, individually purified with 0.8X Sera-Mag Size Selection Beads and eluted in 20 µl of 10 mM Tris-HCl, pH 8.0. PCR2 consisted of 15 µl of purified template from PCR1, 10 µl of 10X Phusion Buffer (NEB), 1 µl of 10 mM dNTPs, 2.5 µl of pooled downstream bait primers with a final concentration of 10 µM,

2.5 µl of 10 µM Universal Illumina P7 primer, 0.5 µl Phusion Polymerase (NEB), and nuclease-free H$_2$O up to a final volume of 50 µl. The reactions were incubated with the same program of PCR1 but with 18 cycles of amplification instead of 20, all five replicates of each sample were then pooled together, purified with 0.8X Sera-Mag Size Selection Beads and eluted in 40 µl of Elution Buffer. The libraries were then quality-checked by both dsDNA Qubit and Tapestation and eventually pooled according to the relative number of reads needed. All libraries were sequenced on an Illumina NovaSeq 6000 instrument, aiming at ~5 M reads per sample.

### UMI-4C data analysis

To analyze the UMI-4C data, we have employed the umi4cpackage R package (https://github.com/tanaylab/umi4cpackage, *Lifshitz et al., 2024*). After setting up the configuration file as described, they were loaded with the *p4cLoadConfFiles* function, and the raw fastq files were analyzed using the *p4cCreate4CseqTrack* function. The 4C profiles were then built and plotted using the *p4cNewProfile* and plot functions, respectively, specifying the *cis* window and overlaying two profiles when needed. The p-values and log$_2$FC between conditions and across a genomic region of interest were finally calculated using the *p4cIntervalsMean* function.

### Metabolic labeling, SLAM-seq, and data analysis

For half-life and synthesis rate estimation upon the KD of *EPB41L4A-AS1*, cells were transfected with control and targeting GapmeRs as described above. Metabolic labeling and SLAM-seq were then performed as previously described (*Zuckerman et al., 2020*), with minor modifications. All steps were performed in the dark until the chemical conversion with iodoacetamide (IAA). After 48 hr, the growth medium was replaced with fresh media supplemented with 500 µM 4-Thiouridine (4sU, Sigma), and replaced again with fresh 4sU after every time point. In total, we collected RNA from cells at time 0 (unlabeled), and after 4, 8, and 24 hr post-4sU addition. RNA extraction was carried out as described above with the addition of DTT (0.1 mM final concentration) to keep the samples under reducing conditions, and IAA conversion was performed as described previously (*Herzog et al., 2017*; *Zuckerman et al., 2020*). Ribosomal RNAs were then depleted using the NEBNext rRNA Depletion Kit V2 (NEB) and libraries were generated with the NEBNext Ultra II Directional Library Kit (NEB), both according the manufacturer's instructions. Estimation of half-lives and synthesis rates was performed using GRAND-SLAM/grandR (*Jürges et al., 2018*), and are listed in *Supplementary file 4*. Notably, the estimation assumes steady-state expression in control samples, but not in the KD ones.

## Acknowledgements

We thank members of the Ulitsky lab for insightful discussions and comments on the manuscript, and in particular Rotem Nevo for his help with the smFISH analysis. We thank Dana Hirsch and Liat Alyagor for assistance with smFISH experiments, Ehud Sivan for imaging data analysis, Noa Gil for discussions about UMI-4C, Florian Erhard for assistance with SLAM-seq analysis, and Josh Mendell's lab for the SNORA13 over-expression plasmid. This work was supported by the ERC Consolidator Grant lncIM-PACT to IU.

## Additional information

### Funding

| Funder | Grant reference number | Author |
| --- | --- | --- |
| European Research Council | lncIMPACT | Alan Monziani<br>Juan Pablo Unfried<br>Igor Ulitsky |

The funders had no role in study design, data collection, and interpretation, or the decision to submit the work for publication.

## Author contributions
Alan Monziani, Conceptualization, Data curation, Software, Formal analysis, Validation, Investigation, Visualization, Methodology, Writing – original draft, Writing – review and editing; Juan Pablo Unfried, Data curation, Formal analysis, Investigation, Methodology; Todor Cvetanovic, Investigation, Methodology; Igor Ulitsky, Conceptualization, Data curation, Formal analysis, Supervision, Funding acquisition, Investigation, Visualization, Methodology, Writing – original draft, Project administration, Writing – review and editing

## Author ORCIDs
Alan Monziani ⓘ https://orcid.org/0000-0001-7505-1986
Todor Cvetanovic ⓘ https://orcid.org/0000-0003-4728-5536
Igor Ulitsky ⓘ https://orcid.org/0000-0003-0555-6561

Reviewer #1 (Public review): https://doi.org/10.7554/eLife.106846.3.sa1
Reviewer #2 (Public review): https://doi.org/10.7554/eLife.106846.3.sa2
Reviewer #3 (Public review): https://doi.org/10.7554/eLife.106846.3.sa3
Author response https://doi.org/10.7554/eLife.106846.3.sa4

---

# Additional files

## Supplementary files
MDAR checklist

Supplementary file 1. Table containing the results of the *in silico* screening combining transcriptomics and GeneHancer data.

Supplementary file 2. Table containing DESeq2 results of the different RNA-seq in this study.

Supplementary file 3. MACS output for the peaks of SUB1 and CTCF obtained by CUT&RUN.

Supplementary file 4. GRAND-SLAM/grandR kinetic results of the SLAM-seq experiment.

Supplementary file 5. List of reagents (GapmeRs, siRNAs, CRISPR guide RNAs, RT-qPCR primers, northern blot probes, antibodies, UMI-4C baits) used in this study.

## Data availability
All the RNA-seq, RIP-seq, UMI-4C, and CUT&RUN datasets were deposited to the GEO database under the accession GSE292272.

The following dataset was generated:

| Author(s) | Year | Dataset title | Dataset URL | Database and Identifier |
|---|---|---|---|---|
| Monziani A, Ulitsky I | 2026 | EPB41L4A-AS1 long noncoding RNA acts in both cis- and trans-acting transcriptional regulation and controls nucleolar biology | https://www.ncbi.nlm.nih.gov/geo/query/acc.cgi?acc=GSE292272 | NCBI Gene Expression Omnibus, GSE292272 |

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
