## [Editor Report · eLife Assessment]

This paper provides **important** findings towards understanding the role of the lncRNA EPB41L4A-AS1 in a human cell line. The data is generally **convincing**, supported by extensive and clever integrative analysis. The work provides insights into how this lncRNA regulates gene expression via complex mechanisms; however the biological relevance awaits validation in other models.

---

## [Referee Report · Reviewer #1 (Public review)]

Monziani and Ulitsky present a large and exhaustive study on the lncRNA EPB41L4A-AS1 using a variety of genomic methods. They uncover a rather complex picture of a RNA transcript that appears to act via diverse pathways to regulate large numbers of genes' expression, including many snoRNAs. The activity of EPB41L4A-AS1 seems to be intimately linked with the protein SUB1, via both direct physical interactions and direct/indirect of SUB1 mRNA expression.

The study is characterised by thoughtful, innovative, integrative genomic analysis. It is shown that EPB41L4A-AS1 interacts with SUB1 protein and that this may lead to extensive changes in SUB1's other RNA partners. Disruption of EPB41L4A-AS1 leads to widespread changes in non-polyA RNA expression, as well as local cis changes. At the clinical level, it is possible that EPB41L4A-AS1 plays disease relevant roles, although these seem to be somewhat contradictory with evidence supporting both oncogenic and tumour suppressive activities.

A couple of issues could be better addressed here. Firstly, the copy number of EPB41L4A-AS1 is an important missing piece of the puzzle. It is apparently highly expressed from the FISH experiments. To get an understanding of how EPB41L4A-AS1 regulates SUB1, an abundant protein, we need to know the relative stoichiometry of these two factors. Secondly, while many of the experiments use two independent Gapmers for EPB41L4A-AS1 knockdown, the RNA-sequencing experiments apparently use just one, with one negative control (?). Evidence is emerging that Gapmers produce extensive off target gene expression effects in cells, potentially exceeding the amount of on-target changes arising through the intended target gene. Therefore, it is important to estimate this through use of multiple targeting and non-targeting ASOs, if one is to get a true picture of EPB41L4A-AS1 target genes. In this Reviewer's opinion, this casts some doubt over interpretation of RNA-seq experiments until that work is done. Nonetheless, the Authors have designed thorough experiments, including overexpression rescue overexpression constructs, to quite confidently assess the role of EPB41L4A-AS1 in snoRNA expression.

It is possible that EPB41L4A-AS1 plays roles in cancer, either as oncogene or tumour suppressor. However it will in future be important to extend these observations to a greater variety of cell contexts.

This work is valuable in providing an extensive and thorough analysis of the global mechanisms of an important regulatory lncRNA, and highlights the complexity of such mechanisms via cis and trans regulation and extensive protein interactions.

---

## [Referee Report · Reviewer #2 (Public review)]

Summary:

In this manuscript, Monziani et al. identified long noncoding RNAs (lncRNAs) that act in cis and are coregulated with their target genes located in close genomic proximity. The authors mined the GeneHancer database, and this analysis led to the identification of four lncRNA-target pairs. The authors decided to focus on lncRNA EPB41L4A-AS1.

They thoroughly characterised this lncRNA, demonstrating that it is located in the cytoplasm and the nuclei, and that its expression is altered in response to different stimuli. Furthermore, the authors showed that EPB41L4A-AS1 regulates EPB41L4A transcription, leading to a mild reduction in EPB41L4A protein levels. This was not recapitulated with sirna-mediated depletion of EPB41L4AAS1. RNA-seq in EPB41L4A-AS1 depleted cells with single LNA revealed 2364 DEGs linked to pathways including the cell cycle, cell adhesion, and inflammatory response. To understand the mechanism of action of EPB41L4A-AS1, the authors mined the ENCODE eCLIP data and identified SUB1 as an lncRNA interactor. The authors also found that the loss of EPB41L4A-AS1 and SUB1 leads to the accumulation of snoRNAs, and that SUB1 localisation changes upon the loss of EPB41L4A-AS1. Finally, the authors showed that EPB41L4A-AS1 deficiency did not change the steady-state levels of SNORA13 nor RNA modification driven by this RNA. The phenotype associated with the loss of EPB41L4A-AS1 is linked to increased invasion and EMT gene signature.

Overall, this is an interesting and nicely done study on the versatile role of EPB41L4A-AS1 and the multifaceted interplay between SUB1 and this lncRNA, but some conclusions and claims need to be supported with additional experiments before publication. My primary concerns are using a single LNA gapmer for critical experiments, increased invasion and nucleolar distribution of SUB1- in EPB41L4A-AS1-depleted cells.

Strengths:

The authors used complementary tools to dissect the complex role of lncRNA EPB41L4A-AS1 in regulating EPB41L4A, which is highly commendable. There are few papers in the literature on lncRNAs at this standard. They employed LNA gapmers, siRNAs, CRISPRi/a, and exogenous overexpression of EPB41L4A-AS1 to demonstrate that the transcription of EPB41L4A-AS1 acts in cis to promote the expression of EPB41L4A by ensuring spatial proximity between the TAD boundary and the EPB41L4A promoter. At the same time, this lncRNA binds to SUB1 and regulates snoRNA expression and nucleolar biology. Overall, the manuscript is easy to read, and the figures are well presented. The methods are sound, and the expected standards are met.

Weaknesses:

The authors should clarify how many lncRNA-target pairs were included in the initial computational screen for cis-acting lncRNAs and why MCF7 was chosen as the cell line of choice. Most of the data uses a single LNA gapmer targeting EPB41L4A-AS1 lncrna (eg, Fig. 2c, 3B and RNA-seq), and the critical experiments should be using at least 2 LNA gapmers. The specificity of SUB1 CUT&RUN is lacking, as well as direct binding of SUB1 to lncRNA EPB41L4A-AS1, which should be confirmed by CLIP qPCR in MCF7 cells. Finally, the role of EPB41L4A-AS1 in SUB1 distribution (Fig. 5) and cell invasion (Fig. 8) needs to be complemented with additional experiments, which should finally demonstrate the role of this lncRNA in nucleolus and cancer-associated pathways. The use of MCF7 as a single cancer cell line is not ideal.

Revised version of the manuscript:

The authors have addressed many of my concerns in their revised manuscript:

The use of single gapmers has been adequately addressed in the revised version of the manuscript, as well as CUT RUN for SUb1.

Future studies will address the role of this lncRNA in invasion and migration using more relevant and appropriate cellular assays. In addition, nucleolar fractionation and analysis of rRNA synthesis are recommended in the follow-up studies for EPB41L4A-AS1.

---

## [Referee Report · Reviewer #3 (Public review)]

Summary:

In Monziani et al. paper entitled: "EPB41L4A-AS1 long noncoding RNA acts in both cis- and trans-acting transcriptional regulation and controls nucleolar biology", the authors made some interesting observations that EPB41L4A-AS1 lncRNA can regulate the transcription of both the nearby coding gene and genes on other chromosomes. They started by computationally examining lncRNA-gene pairs by analyzing co-expression, chromatin features of enhancers, TF binding, HiC connectome and eQTLs. They then zoomed in on four pairs of lncRNA-gene pairs and used LNA antisense oligonucleotides to knock down these lncRNAs. This revealed EPB41L4A-AS1 as the only one that can regulate the expression of its cis-gene target EPB41L4A. By RNA-FISH, the authors found this lncRNA to be located in all three parts of a cell: chromatin, nucleoplasm and cytoplasm. RNA-seq after LNA knockdown of EPB41L4A-AS1 showed that this increased >1100 genes and decreased >1250 genes, including both nearby genes and genes on other chromosomes. They later found that EPB41L4A-AS1 may interact with SUB1 protein (an RNA binding protein) to impact the target genes of SUB1. EPB41L4A-AS1 knockdown reduced the mRNA level of SUB1 and altered the nuclear location of SUB1. Later, the authors observed that EPB41L4A-AS1 knockdown caused increase of snRNAs and snoRNAs, likely via disrupted SUB1 function. In the last part of the paper, the authors conducted rescue experiments that suggested that the full-length, intron- and SNORA13-containing EPB41L4A-AS1 is required to partially rescue snoRNA expression. They also conducted SLAM-Seq and showed that the increased abundance of snoRNAs is primarily due to their hosts' increased transcription and stability. They end with data showing that EPB41L4A-AS1 knockdown reduced MCF7 cell proliferation but increased its migration, suggesting a link to breast cancer progression and/or metastasis.

Strengths:

The strength of the paper includes: it is overall well-written; the results are overall presented with good technical rigor and appropriate interpretation. The observation that a complex lncRNA EPB41L4A-AS1 regulates both cis and trans target genes, if fully proven, is interesting and important.

Weaknesses:

The weakness includes: the paper is a bit disjointed as it started from cis and trans gene regulation, but later it switched to a partially relevant topic of snoRNA metabolism via SUB1; the paper was limited in the mechanisms as to how these trans genes (including SUB1 or NPM1 genes themselves) are affected by EPB41L4A-AS1 knockdown; there are discrepancy of results upon EPB41L4A-AS1 knockdown by LNA versus by CRISPR activation, or by plasmid overexpression of this lncRNA.

Overall, the data is supportive of a role of this lncRNA in regulating cis and trans target genes, and thereby impacting cellular phenotypes.

---

## [Author Response]

The following is the authors’ response to the original reviews.

**Public Reviews:**

**Reviewer #1 (Public review):**
Monziani and Ulitsky present a large and exhaustive study on the lncRNA EPB41L4A-AS1 using a variety of genomic methods. They uncover a rather complex picture of an RNA transcript that appears to act via diverse pathways to regulate the expression of large numbers of genes, including many snoRNAs. The activity of EPB41L4A-AS1 seems to be intimately linked with the protein SUB1, via both direct physical interactions and direct/indirect of SUB1 mRNA expression.The study is characterised by thoughtful, innovative, integrative genomic analysis. It is shown that EPB41L4A-AS1 interacts with SUB1 protein and that this may lead to extensive changes in SUB1's other RNA partners. Disruption of EPB41L4A-AS1 leads to widespread changes in non-polyA RNA expression, as well as local cis changes. At the clinical level, it is possible that EPB41L4A-AS1 plays disease-relevant roles, although these seem to be somewhat contradictory with evidence supporting both oncogenic and tumour suppressive activities.A couple of issues could be better addressed here. Firstly, the copy number of EPB41L4A-AS1 is an important missing piece of the puzzle. It is apparently highly expressed in the FISH experiments. To get an understanding of how EPB41L4A-AS1 regulates SUB1, an abundant protein, we need to know the relative stoichiometry of these two factors. Secondly, while many of the experiments use two independent Gapmers for EPB41L4A-AS1 knockdown, the RNA-sequencing experiments apparently use just one, with one negative control (?). Evidence is emerging that Gapmers produce extensive off-target gene expression effects in cells, potentially exceeding the amount of on-target changes arising through the intended target gene. Therefore, it is important to estimate this through the use of multiple targeting and non-targeting ASOs, if one is to get a true picture of EPB41L4A-AS1 target genes. In this Reviewer's opinion, this casts some doubt over the interpretation of RNA-seq experiments until that work is done. Nonetheless, the Authors have designed thorough experiments, including overexpression rescue constructs, to quite confidently assess the role of EPB41L4A-AS1 in snoRNA expression.It is possible that EPB41L4A-AS1 plays roles in cancer, either as an oncogene or a tumour suppressor. However, it will in the future be important to extend these observations to a greater variety of cell contexts.This work is valuable in providing an extensive and thorough analysis of the global mechanisms of an important regulatory lncRNA and highlights the complexity of such mechanisms via cis and trans regulation and extensive protein interactions.
**Reviewer #2 (Public review):**
Summary:In this manuscript, Monziani et al. identified long noncoding RNAs (lncRNAs) that act in cis and are coregulated with their target genes located in close genomic proximity. The authors mined the GeneHancer database, and this analysis led to the identification of four lncRNA-target pairs. The authors decided to focus on lncRNA EPB41L4A-AS1.They thoroughly characterised this lncRNA, demonstrating that it is located in the cytoplasm and the nuclei, and that its expression is altered in response to different stimuli. Furthermore, the authors showed that EPB41L4A-AS1 regulates EPB41L4A transcription, leading to a mild reduction in EPB41L4A protein levels. This was not recapitulated with siRNA-mediated depletion of EPB41L4AAS1. RNA-seq in EPB41L4A-AS1-depleted cells with single LNA revealed 2364 DEGs linked to pathways including the cell cycle, cell adhesion, and inflammatory response. To understand the mechanism of action of EPB41L4A-AS1, the authors mined the ENCODE eCLIP data and identified SUB1 as an lncRNA interactor. The authors also found that the loss of EPB41L4A-AS1 and SUB1 leads to the accumulation of snoRNAs, and that SUB1 localisation changes upon the loss of EPB41L4A-AS1. Finally, the authors showed that EPB41L4A-AS1 deficiency did not change the steady-state levels of SNORA13 nor RNA modification driven by this RNA. The phenotype associated with the loss of EPB41L4A-AS1 is linked to increased invasion and EMT gene signature.Overall, this is an interesting and nicely done study on the versatile role of EPB41L4A-AS1 and the multifaceted interplay between SUB1 and this lncRNA, but some conclusions and claims need to be supported with additional experiments. My primary concerns are using a single LNA gapmer for critical experiments, increased invasion, and nucleolar distribution of SUB1- in EPB41L4A-AS1-depleted cells. These experiments need to be validated with orthogonal methods.Strengths:The authors used complementary tools to dissect the complex role of lncRNA EPB41L4A-AS1 in regulating EPB41L4A, which is highly commendable. There are few papers in the literature on lncRNAs at this standard. They employed LNA gapmers, siRNAs, CRISPRi/a, and exogenous overexpression of EPB41L4A-AS1 to demonstrate that the transcription of EPB41L4A-AS1 acts in cis to promote the expression of EPB41L4A by ensuring spatial proximity between the TAD boundary and the EPB41L4A promoter. At the same time, this lncRNA binds to SUB1 and regulates snoRNA expression and nucleolar biology. Overall, the manuscript is easy to read, and the figures are well presented. The methods are sound, and the expected standards are met.Weaknesses:The authors should clarify how many lncRNA-target pairs were included in the initial computational screen for cis-acting lncRNAs and why MCF7 was chosen as the cell line of choice. Most of the data uses a single LNA gapmer targeting EPB41L4A-AS1 lncRNA (eg, Fig. 2c, 3B, and RNA-seq), and the critical experiments should be using at least 2 LNA gapmers. The specificity of SUB1 CUT&RUN is lacking, as well as direct binding of SUB1 to lncRNA EPB41L4A-AS1, which should be confirmed by CLIP qPCR in MCF7 cells. Finally, the role of EPB41L4A-AS1 in SUB1 distribution (Figure 5) and cell invasion (Figure 8) needs to be complemented with additional experiments, which should finally demonstrate the role of this lncRNA in nucleolus and cancer-associated pathways. The use of MCF7 as a single cancer cell line is not ideal.
**Reviewer #3 (Public review):**
Summary:In this paper, the authors made some interesting observations that EPB41L4A-AS1 lncRNA can regulate the transcription of both the nearby coding gene and genes on other chromosomes. They started by computationally examining lncRNA-gene pairs by analyzing co-expression, chromatin features of enhancers, TF binding, HiC connectome, and eQTLs. They then zoomed in on four pairs of lncRNA-gene pairs and used LNA antisense oligonucleotides to knock down these lncRNAs. This revealed EPB41L4A-AS1 as the only one that can regulate the expression of its cis-gene target EPB41L4A. By RNA-FISH, the authors found this lncRNA to be located in all three parts of a cell: chromatin, nucleoplasm, and cytoplasm. RNA-seq after LNA knockdown of EPB41L4A-AS1 showed that this increased >1100 genes and decreased >1250 genes, including both nearby genes and genes on other chromosomes. They later found that EPB41L4A-AS1 may interact with SUB1 protein (an RNA-binding protein) to impact the target genes of SUB1. EPB41L4A-AS1 knockdown reduced the mRNA level of SUB1 and altered the nuclear location of SUB1. Later, the authors observed that EPB41L4A-AS1 knockdown caused an increase of snRNAs and snoRNAs, likely via disrupted SUB1 function. In the last part of the paper, the authors conducted rescue experiments that suggested that the full-length, intron- and SNORA13-containing EPB41L4A-AS1 is required to partially rescue snoRNA expression. They also conducted SLAM-Seq and showed that the increased abundance of snoRNAs is primarily due to their hosts' increased transcription and stability. They end with data showing that EPB41L4A-AS1 knockdown reduced MCF7 cell proliferation but increased its migration, suggesting a link to breast cancer progression and/or metastasis.Strengths:Overall, the paper is well-written, and the results are presented with good technical rigor and appropriate interpretation. The observation that a complex lncRNA EPB41L4A-AS1 regulates both cis and trans target genes, if fully proven, is interesting and important.Weaknesses:The paper is a bit disjointed as it started from cis and trans gene regulation, but later it switched to a partially relevant topic of snoRNA metabolism via SUB1. The paper did not follow up on the interesting observation that there are many potential trans target genes affected by EPB41L4A-AS1 knockdown and there was limited study of the mechanisms as to how these trans genes (including SUB1 or NPM1 genes themselves) are affected by EPB41L4A-AS1 knockdown. There are discrepancies in the results upon EPB41L4A-AS1 knockdown by LNA versus by CRISPR activation, or by plasmid overexpression of this lncRNA.
**Recommendations for the authors:**

**Reviewer #1 (Recommendations for the authors):**
(1) Copy number:Perhaps I missed it, but it seems that no attempt is made to estimate the number of copies of EPB41L4A-AS1 transcripts per cell. This should be possible given RNAseq and FISH. At least an order of magnitude estimate. This is important for shedding light on the later observations that EPB41L4A-AS1 may interact with SUB1 protein and regulate the expression of thousands of mRNAs.

We thank the reviewer for the insightful suggestion. We agree that an estimate of EPB41L4A-AS1 copy number might further strengthen the hypotheses presented in the manuscript. Therefore, we analyzed the smFISH images and calculated the copy number per cell of this lncRNA, as well as that of GAPDH as a comparison.

Because segmenting MCF-7 cells proved to be difficult due to the extent of the cell-cell contacts they establish, we imaged multiple (n = 14) fields of view, extracted the number of EPB41L4A-AS1/GAPDH molecules in each field and divided them by the number of cells (as assessed by DAPI staining, 589 cells in total). We detected an average of 33.37 ± 3.95 EPB41L4A-AS1 molecules per cell, in contrast to 418.27 ± 61.79 GAPDH molecules. As a comparison, within the same qPCR experiment the average of the Ct values of these two RNAs is about 22.3 and 17.5, the FPKMs in the polyA+ RNA-seq are ~ 2479.4 and 35.6, and the FPKMs in the rRNA-depleted RNA-seq are ~ 3549.9 and 19.3, respectively. Thus, our estimates of the EPB41L4A-AS1 copy number in MCF-7 cells fits well into these observations.

The question whether an average of ~35 molecules per cell is sufficient to affect the expression of thousands of genes is somewhat more difficult to ascertain. As discussed below, it is unlikely that all the genes dysregulated following the KD of EPB41L4A-AS1 are all direct targets of this lncRNA, and indeed SUB1 depletion affects an order of magnitude fewer genes. It has been shown that lncRNAs can affect the behavior of interacting RNAs and proteins in a substoichiometric fashion (Unfried & Ulitsky, 2022), but whether this applies to EPB41L4A-AS1 remains to be addressed in future studies. Nonetheless, this copy number appears to be sufficient for a trans-acting functions for this lncRNA, on top of its cis-regulatory role in regulating EPB41L4A. We added this information in the text as follows:

“Using single-molecule fluorescence in-situ hybridization (smFISH) and subcellular fractionation we found that EPB41L4A-AS1 is expressed at an average of 33.37 ± 3.95 molecule per cell, and displays both nuclear and cytoplasmic localization in MCF-7 cells (Fig. 1D), with a minor fraction associated with chromatin as well (Fig. 1E).”

We have updated the methods section as well:

“To visualize the subcellular localization of EPB41L4A-AS1 in vivo, we performed single-molecule fluorescence in situ hybridization (smFISH) using HCR amplifiers. Probe sets (n = 30 unique probes) targeting EPB41L4A-AS1 and GAPDH (positive control) were designed and ordered from Molecular Instruments. We followed the Multiplexed HCR v3.0 protocol with minor modifications. MCF-7 cells were plated in 8-well chambers (Ibidi) and cultured O/N as described above. The next day, cells were fixed with cold 4% PFA in 1X PBS for 10 minutes at RT and then permeabilized O/N in 70% ethanol at -20°C. Following permeabilization, cells were washed twice with 2X SSC buffer and incubated at 37°C for 30 minutes in hybridization buffer (HB). The HB was then replaced with a probe solution containing 1.2 pmol of EPB41L4A-AS1 probes and 0.6 pmol of GAPDH probes in HB. The slides were incubated O/N at 37°C. To remove excess probes, the slides were washed four times with probe wash buffer at 37°C for 5 minutes each, followed by two washes with 5X SSCT at RT for 5 minutes. The samples were then pre-amplified in amplification buffer for 30 minutes at RT and subsequently incubated O/N in the dark at RT in amplification buffer supplemented with 18 pmol of the appropriate hairpins. Finally, excess hairpins were removed by washing the slides five times in 5X SSCT at RT. The slides were mounted with ProLong Glass Antifade Mountant (Invitrogen), cured O/N in the dark at RT, and imaged using a Nikon CSU-W1 spinning disk confocal microscope. In order to estimate the RNA copy number, we imaged multiple distinct fields, extracted the number of EPB41L4A-AS1/GAPDH molecules in each field using the “Find Maxima” tool in ImageJ/Fiji, and divided them by the number of cells (as assessed by DAPI staining).”

(2) Gapmer results:Again, it is quite unclear how many and which Gapmer is used in the genomics experiments, particularly the RNA-seq. In our recent experiments, we find very extensive off-target mRNA changes arising from Gapmer treatment. For this reason, it is advisable to use both multiple control and multiple targeting Gapmers, so as to identify truly target-dependent expression changes. While I acknowledge and commend the latter rescue experiments, and experiments using multiple Gapmers, I'd like to get clarification about how many and which Gapmers were used for RNAseq, and the authors' opinion on the need for additional work here.

We agree with the Reviewer that GapmeRs are prone to off-target and unwanted effects (Lai et al., 2020; Lee & Mendell, 2020; Maranon & Wilusz, 2020). Early in our experiments, we found out that LNA1 triggers a non-specific CDKN1A/p21 activation (Fig. S5A-C), and thus, we have initially performed some experiments such as RNA-seq with only LNA2.

Nonetheless, other experiments were performed using both GapmeRs, such as multiple RT-qPCRs, UMI-4C, SUB1 and NPM1 imaging, and the in vitro assays, among others, and consistent results were obtained with both LNAs.

To accommodate the request by this and the other reviewers, we have now performed another round of polyA+ RNA-seq following EPB41L4A-AS1 knockdown using LNA1 or LNA2, as well as the previously used and an additional control GapmeR. The FPKMs of the control samples are highly-correlated both within replicates and between GapmeRs (Fig. S6A). More importantly, the fold-changes to control are highly correlated between the two on-target GapmeRs LNA1 and LNA2, regardless of the GapmeR used for normalization (Fig. S6B), thus showing that the bulk of the response is shared and likely the direct result of the reduction in the levels of EPB41L4A-AS1. Notably, key targets NPM1 and MTREX (see discussion, Fig. S12A-C and comments to Reviewer 3) were found to be downregulated by both LNAs (Fig. S6C).

However, we acknowledge that some of the dysregulated genes are observed only when using one GapmeR and not the other, likely due to a combination of indirect, secondary and non-specific effects, and as such it is difficult to infer the direct response. Supporting this, LNA2 yielded a total of 1,069 DEGs (617 up and 452 down) and LNA1 2,493 DEGs (1,328 up and 1,287 down), with the latter triggering a stronger response most likely as a result of the previously mentioned CDKN1A/p21 induction. Overall, 45.1% of the upregulated genes following LNA2 transfection were shared with LNA1, in contrast to only the 24.3% of the downregulated ones.

We have now included these results in the Results section (see below) and in Supplementary Figure (Fig. S6).

“Most of the consequences of the depletion of EPB41L4A-AS1 are thus not directly explained by changes in EPB41L4A levels. An additional trans-acting function for EPB41L4A-AS1 would therefore be consistent with its high expression levels compared to most lncRNAs detected in MCF-7 (Fig. S5G). To strengthen these findings, we have transfected MCF-7 cells with LNA1 and a second control GapmeR (NT2), as well as the previous one (NT1) and LNA2, and sequenced the polyadenylated RNA fraction as before. Notably, the expression levels (in FPKMs) of the replicates of both control samples are highly correlated with each other (Fig. S6A), and the global transcriptomic changes triggered by the two EPB41L4A-AS1-targeting LNAs are largely concordant (Fig. S6B and S6C). Because of this concordance and the cleaner (i.e., no CDKN1A upregulation) readout in LNA2-transfected cells, we focused mainly on these cells for subsequent analyses.”

(3) Figure 1E:Can the authors comment on the unusual (for a protein-coding mRNA) localisation of EPB41L4A, with a high degree of chromatin enrichment?

We acknowledge that mRNAs from protein-coding genes displaying nuclear and chromatin localizations are quite unusual. The nuclear and chromatin localization of some mRNAs are often due to their low expression, length, time that it takes to be transcribed, repetitive elements and strong secondary structures (Bahar Halpern et al., 2015; Didiot et al., 2018; Lubelsky & Ulitsky, 2018; Ly et al., 2022).

We now briefly mention this in the text:

“In contrast, both EPB41L4A and SNORA13 were mostly found in the chromatin fraction (Fig. 1E), the former possibly due to the length of its pre-mRNA (>250 kb), which would require substantial time to transcribe (Bahar Halpern et al., 2015; Didiot et al., 2018; Lubelsky & Ulitsky, 2018; Ly et al., 2022).”

Supporting our results, analysis of the ENCODE MCF-7 RNA-seq data of the cytoplasmic, nuclear and total cell fractions indeed shows a nuclear enrichment of the EPB41L4A mRNA (Author response image 1), in line with what we observed in Fig. 1E by RT-qPCR.

**Author response image 1. sa4fig1:** The EPB41L4A transcript is nuclear-enriched in the MCF-7 ENCODE subcellular RNA-seq dataset. Scatterplot of gene length versus cytoplasm/nucleus ratio (as computed by DESeq2) in MCF-7 cells. Each dot represents an unique gene, color-coded reflecting if their DESeq2 adjusted p-value < 0.05 and absolute log_2_FC > .41 (33% enrichment or depletion).GAPDH and MALAT1 are shown as representative cytoplasmic and nuclear transcripts, respectively. Data from ENCODE.

(4) Annotation and termini of EPB41L4A-AS1:The latest Gencode v47 annotations imply an overlap of the sense and antisense, different from that shown in Figure 1C. The 3' UTR of EPB41L4A is shown to extensively overlap EPB41L4A-AS1. This could shed light on the apparent regulation of the former by the latter that is relevant for this paper. I'd suggest that the authors update their figure of the EPB41L4A-AS1 locus organisation with much more detail, particularly evidence for the true polyA site of both genes. What is more, the authors might consider performing RACE experiments for both RNAs in their cells to definitely establish whether these transcripts contain complementary sequence that could cause their Watson-Crick hybridisation, or whether their two genes might interfere with each other via some kind of polymerase collision.

We thank the reviewer for pointing this out. Also in previous GENCODE annotations, multiple isoforms were reported with some overlapping the 3’ UTR of EPB41L4A. In the EPB41L4A-AS1 locus image (Fig. 1C), we report at the bottom the different transcripts isoforms currently annotated, and a schematics of the one that is clearly the most abundant in MCF-7 cells based on RNA-seq read coverage. This is supported by both the polyA(+) and ribo(-) RNA-seq data, which are strand-specific, as shown in the figure.

We now also examined the ENCODE/CSHL MCF-7 RNA-seq data from whole cell, cytoplasm and nucleus fractions, as well as 3P-seq data (Jan et al., 2011) (unpublished data from human cell lines), reported in Author response image 2. All these data support the predominant use of the proximal polyA site in human cell lines. This shorter isoform does not overlap EPB41L4A.

**Author response image 2. sa4fig2:** Most EPB41L4A-AS1 transcripts end before the 3’ end of EPB41L4A. UCSC genome browser view showing tracks from 3P-seq data in different cell lines and neural crest (top, with numbers representing the read counts, i.e. how many times that 3’ end has been detected), and stranded ENCODE subcellular RNA-seq (bottom).

Based on these data, the large majority of cellular transcripts of EPB41L4A-AS1 terminate at the earlier polyA site and don’t overlap with EPB41L4A. There is a small fraction that appears to be restricted to the nucleus that terminates later at the annotated isoform. 3' RACE experiments are not expected to provide substantially different information beyond what is already available.

(5) Figure 3C:There is an apparent correlation between log2FC upon EPB41L4A-AS1 knockdown, and the number of clip sites for SUB1. However, I expect that the clip signal correlates strongly with the mRNA expression level, and that log2FC may also correlate with the same. Therefore, the authors would be advised to more exhaustively check that there really is a genuine relationship between log2FC and clip sites, after removing any possible confounders of overall expression level.

As the reviewer suggested, there is a correlation between the baseline expression level and the strength of SUB1 binding in the eCLIP data. To address this issue, we built expression-matched controls for each group of SUB1 interactors and checked the fold-changes following EPB41L4A-AS1 KD, similarly to what we have done in Fig. 3C. The results are presented, and are now part of Supplementary Figure 7 (Fig. S7C).

Based on this analysis, while there is a tendency of increased expression with increased SUB1 binding, when controlling for expression levels the effect of down-regulation of SUB1-bound RNAs upon lncRNA knockdown remains, suggesting that it is not merely a confounding effect. We have updated the text as follows:

“We hypothesized that loss of EPB41L4A-AS1 might affect SUB1, either via the reduction in its expression or by affecting its functions. We stratified SUB1 eCLIP targets into confidence intervals, based on the number, strength and confidence of the reported binding sites. Indeed, eCLIP targets of SUB1 (from HepG2 cells profiled by ENCODE) were significantly downregulated following EPB41L4A-AS1 KD in MCF-7, with more confident targets experiencing stronger downregulation (Fig. 3C). Importantly, this still holds true when controlling for gene expression levels (Fig. S7C), suggesting that this negative trend is not due to differences in their baseline expression.”

(6) The relation to cancer seems somewhat contradictory, maybe I'm missing something. Could the authors more clearly state which evidence is consistent with either an Oncogene or a Tumour Suppressive function, and discuss this briefly in the Discussion? It is not a problem if the data are contradictory, however, it should be discussed more clearly.

We acknowledge this apparent contradiction. Cancer cells are characterized by a multitude of hallmarks depending on the cancer type and stage, including high proliferation rates and enhanced invasive capabilities. The notion that cells with reduced EPB41L4A-AS1 levels exhibit lower proliferation, yet increased invasion is compatible with a function as an oncogene. Cells undergoing EMT may reduce or even completely halt proliferation/cell division, until they revert back to an epithelial state (Brabletz et al., 2018; Dongre & Weinberg, 2019). Notably, downregulated genes following EPB41L4A-AS1 KD are enriched in GO terms related to cell proliferation and cell cycle progression (Fig. 2I), whereas those upregulated are enriched for terms linked to EMT processes. Thus, while we cannot rule out a potential function as tumor suppressor gene, our data fit better the notion that EPB41L4A-AS1 promotes invasion, and thus, primarily functions as an oncogene. We now address this in point in the discussion:

“The notion that cells with reduced EPB41L4A-AS1 levels exhibit lower proliferation (Fig. 8C), yet increased invasion (Fig. 8A and 8B) is compatible with a function as an oncogene by promoting EMT (Fig. 8D and 8E). Cells undergoing this process may reduce or even completely halt proliferation/cell division, until they revert back to an epithelial state (Brabletz et al., 2018; Dongre & Weinberg, 2019). Notably, downregulated genes following EPB41L4A-AS1 KD are enriched in GO terms related to cell proliferation and cell cycle progression (Fig. 2I), whereas those upregulated for terms linked to EMT processes. Thus, while we cannot rule out a potential function as tumor suppressor gene, our data better fits the idea that this lncRNA promotes invasion, and thus, primarily functions as an oncogene.”

**Reviewer #2 (Recommendations for the authors):**
Below are major and minor points to be addressed. We hope the authors find them useful.(1) Figure 1:

Where are LNA gapmers located within the EPB41L4A-AS1 gene? Are they targeting exons or introns of the EPB41L4A-AS1? Please clarify or include in the figure.

We now report the location of the two GapmeRs in Fig. 1C. LNA1 targets the intronic region between SNORA13 and exon 2, and LNA2 the terminal part of exon 1.

(2) Figure 2B:Why is a single LNA gapmer used for EPB41L4A Western? In addition, are the qPCR data in Figure 2B the same as in Figure 1B? Please clarify.

The Western Blot was performed after transfecting the cells with either LNA1 or LNA2. We now have replaced Fig. 2C with the full Western Blot image, in order to show both LNAs. With respect to the qPCRs in Fig. 1B and 2B, they represent the results from two independent experiments.

(3) Figure 2F:2364 DEGs for a single LNA is a lot of deregulated genes in RNA-seq data. How do the authors explain such a big number in DEGs? Is that because this LNA was intronic? Additional LNA gapmer would minimise the "real" lncRNA target and any potential off-target effect.

We agree with the Reviewer that GapmeRs are prone to off-target and unwanted effects (Lai et al.,2020; Lee & Mendell, 2020; Maranon & Wilusz, 2020). Early in our experiments, we found out that LNA1 triggers a non-specific CDKN1A/p21 activation (Fig. S5A-C), and thus, we have initially performed some experiments such as RNA-seq with only LNA2.

Nonetheless, other experiments were performed using both GapmeRs, such as multiple RT-qPCRs, UMI-4C, SUB1 and NPM1 imaging, and the in vitro assays, among others, and consistent results were obtained with both LNAs.

To accommodate the request by this and the other reviewers, we have now performed another round of polyA+ RNA-seq following EPB41L4A-AS1 knockdown using LNA1 or LNA2, as well as the previously used and an additional control GapmeR. The FPKMs of the control samples are highly-correlated both within replicates and between GapmeRs (Fig. S6A). More importantly, the fold-changes to control are highly correlated between the two on-target GapmeRs LNA1 and LNA2, regardless of the GapmeR used for normalization (Fig. S6B), thus showing that despite significant GapmeR-specific effects, the bulk of the response is shared and likely the direct result of the reduction in the levels of EPB41L4A-AS1. Notably, key targets NPM1 and MTREX (see discussion, Fig. S12A-C and comments to Reviewer 3) were found to be downregulated by both LNAs (Fig. S6C).

However, we acknowledge that some of the dysregulated genes are observed only when using one GapmeR and not the other, likely due to a combination of indirect, secondary and non-specific effects, and as such it is difficult to infer the direct response. Supporting this, LNA2 yielded a total of 1,069 DEGs (617 up and 452 down) and LNA1 2,493 DEGs (1,328 up and 1,287 down), with the latter triggering a stronger response most likely as a result of the previously mentioned CDKN1A/p21 induction. Overall, 45.1% of the upregulated genes following LNA2 transfection were shared with LNA1, in contrast to only the 24.3% of the downregulated ones.

We have now included these results in the Results section (see below) and in Supplementary Figure (Fig. S6).

“Most of the consequences of the depletion of EPB41L4A-AS1 are thus not directly explained by changes in EPB41L4A levels. An additional trans-acting function for EPB41L4A-AS1 would therefore be consistent with its high expression levels compared to most lncRNAs detected in MCF-7 (Fig. S5G). To strengthen these findings, we have transfected MCF-7 cells with LNA1 and a second control GapmeR (NT2), as well as the previous one (NT1) and LNA2, and sequenced the polyadenylated RNA fraction as before. Notably, the expression levels (in FPKMs) of the replicates of both control samples are highly correlated with each other (Fig. S6A), and the global transcriptomic changes triggered by the two EPB41L4A-AS1-targeting LNAs are largely concordant (Fig. S6B and S6C). Because of this concordance and the cleaner (i.e., no CDKN1A upregulation) readout in LNA2-transfected cells, we focused mainly on these cells for subsequent analyses.”

(4) Figure 3B: Does downregulation of SUB1 and NPM1 reflect at the protein level with both LNA gapmers? The authors should show a heatmap and metagene profile for SUB1 CUT & RUN. How did the author know that SUB1 binding is specific, since CUT & RUN was not performed in SUB1-depleted cells?

As requested by both Reviewer #2 and #3, we have performed WB for SUB1, NPM1 and FBL following EPB41L4A-AS1 KD with two targeting (LNA1 and LNA2) and the previous control GapmeRs. Interestingly, we did not detect any significant downregulation of either proteins (Author response image 3), although this might be the result of the high variability observed in the control samples. Moreover, the short timeframe in which the experiments have been conducted━that is, transient transfections for 3 days━might not be sufficient time for the existing proteins to be degraded, and thus, the downregulation is more evident at the RNA (Fig. 3B and Supplementary Figure 6C) rather than protein level.

**Author response image 3. sa4fig3:** EPB41L4A-AS1 KD has only marginal effects on the levels of nucleolar proteins. (A) Western Blots for the indicated proteins after the transfection for 3 days of the control and targeting GapmeRs. (B) Quantification of the protein levels from (A). All experiments were performed in n=3 biological replicates, with the error bars in the barplots representing the standard deviation. ns - P>0.05; * - P<0.05; ** - P<0.01; *** - P<0.001 (two-sided Student’s t-test).

Following the suggestion by the Reviewer, we now show both the SUB1 CUT&RUN metagene profile (previously available as Fig. 3F) and the heatmap (now Fig. 3G) around the TSS of all genes, stratified by their expression level. Both graphs are reported.

We show that the antibody signal is responsive to SUB1 depletion via siRNAs in both WB (Fig. S8F) and IF (Fig. 5E) experiments. As mentioned below, this and the absence of non-specific signals makes us confident in the CUT&RUN data. Performing CUT&RUN in SUB1 depleted cells would be difficult to interpret as perturbations are typically not complete, and so the remaining protein can still bind the same regions. Since there isn’t a clear way to add spike-ins to CUT&RUN experiments, it is very difficult to show specificity of binding by CUT&RUN in siRNA-knockdown cells.

(5) Figure 3D: The MW for the depicted proteins are lacking. Why is there no SUB1 protein in the input? Please clarify. Since the authors used siRNA to deplete SUB1, it would be good to know if the antibody is specific in their CUT & RUN (see above)

We apologize for the lack of the MW in Fig. 3D. As shown in Fig. S8F, SUB1 is ~18 kDa and the antibody signal is responsive to SUB1 depletion via siRNAs in both WB (Fig. S8F) and IF (Fig. 5E) experiments. Thus, given its (1) established specificity in those two settings and (2) the lack of generalized signal at most open chromatin regions, which is typical of nonspecific CUT&RUN experiments, we are confident in the specificity of the CUT&RUN results.

We now mention the MW of SUB1 in Fig. 3D as well and we provide in Author response image 4 the full SUB1 WB picture, enhancing the contrast to highlight the bands. We agree that the SUB1 band in the input is weak, likely reflecting the low abundance in that fraction and the detection difficulty due to its low MW (see Fig. S8F).

**Author response image 4. sa4fig4:** Western blot for SUB1 following RIP using either a SUB1 or IgG antibody. IN - input, SN - supernatant/unbound, B - bound.

(6) Supplementary Figure 6C:The validation of lncRNA EPB41L4A-AS1 binding to SUB1 should be confirmed by CLIP qPCR, since native RIP can lead to reassociation of RNA-protein interactions (PMID: 15388877). Additionally, the eclip data presented in Figure 3a were from a different cell line and not MCF7.

We acknowledge that the SUB1 eCLIP data was generated in a different cell line, as we mentioned in the text:

“Indeed, eCLIP targets of SUB1 (from HepG2 cells profiled by ENCODE) were significantly downregulated following EPB41L4A-AS1 KD in MCF-7, with more confident targets experiencing stronger downregulation (Fig. 3C). Importantly, this still holds true when controlling for gene expression levels (Fig. S7C), suggesting that this negative trend is not due to differences in their baseline expression. To obtain SUB1-associated transcripts in MCF-7 cells; we performed a native RNA immunoprecipitation followed by sequencing of polyA+ RNAs (RIP-seq) (Fig. 3D, S7D and S7E).”

Because of this, we resorted to native RIP, in order to get binding information in our experimental system. As we show independent evidence for binding using both eCLIP and RIP, and the substantial challenge in establishing the CLIP method, which has not been successfully used in our group, we respectfully argue that further validations are out of scope of this study. We nonetheless agree that several genes which are nominally significantly enriched in our RIP data are likely not direct targets of SUB1, especially given that it is difficult to assign the perfect threshold that discriminates between bound and unbound RNAs.

We now additionally mention this at the beginning of the paragraph as well:

“In order to identify potential factors that might be associated with EPB41L4A-AS1, we inspected protein-RNA binding data from the ENCODE eCLIP dataset(Van Nostrand et al., 2020). The exons of the EPB41L4A-AS1 lncRNA were densely and strongly bound by SUB1 (also known as PC4) in both HepG2 and K562 cells (Fig. 3A).”

(7) Figure 3G:Can the authors distinguish whether loss of EPB41L4A-AS1 affects SUB1 chromatin binding or its activity as RBP? Please discuss.

Distinguishing between altered SUB1 chromatin and RNA binding is challenging, as this protein likely does not interact directly with chromatin and exhibits rather promiscuous RNA binding properties (Ray et al., 2023). In particular, SUB1 (also known as PC4) interacts with and regulates the activity of all three RNA polymerases, and was reported to be involved in transcription initiation and elongation, response to DNA damage, chromatin condensation (Conesa & Acker, 2010; Das et al., 2006; Garavís & Calvo, 2017; Hou et al., 2022) and telomere maintenance (Dubois et al., 2025; Salgado et al., 2024).

Based on our data, genes whose promoters are occupied by SUB1 display marginal, yet highly significant changes in their steady-state expression levels upon lncRNA perturbations. We also show that upon EPB41L4A-AS1 KD, SUB1 acquires a stronger nucleolar localization (Fig. 5A), which likely affects its RNA interactome as well. However, further elucidating these activities would require performing RIP-seq and CUT&RUN in lncRNA-depleted cells, which we argue is out of the scope of the current study. We note that KD of SUB1 with siRNAs have milder effects than that of EPB41L4A-AS1 (Fig. S8G), suggesting that additional players and effects shape the observed changes. Therefore, it is highly likely that the loss of this lncRNA affects both SUB1 chromatin binding profile and RNA binding activity, with the latter likely resulting in the increased snoRNAs abundance.

(8) Figure 4: Can the authors show that a specific class of snorna is affected upon depletion of SUB1 and EPB41L4A-AS1? Can they further classify the effect of their depletion on H/ACA box snoRNAs, C/D box snoRNAs, and scaRNAs?

Such potential distinct effect on the different classes of snoRNAs was considered, and the results are available in Fig. S8B and S8H (boxplots, after EPB41L4A-AS1 and SUB1 depletion), as well as Fig. 4F and S9F (scatterplots between EPB41L4A-AS1 and SUB1 depletion, and EPB41L4A-AS1 and GAS5 depletion, respectively). We see no preferential effect on one group of snoRNAs or the other.

(9) Figure 5: From the representative images, it looks to me that LNA 2 targeting EPB41L4A-AS1 has a bigger effect on nucleolar staining of SUB1. To claim that EPB41L4A-AS1 depletion "shifts SUB1 to a stronger nucleolar distribution", the authors need to perform IF staining for SUB1 and Fibrillarin, a known nucleolar marker. Also, how does this data fit with their qPCR data shown in Figure 3B? It is instrumental for the authors to demonstrate by IF or Western blotting that SUB1 levels decrease in one fraction and increase specifically in the nucleolus. They could perform Western blot for SUB1 and Fibrillarin in EPB41L4A-AS1-depleted cells and isolate cytoplasmic, nuclear, and nucleolar fractions.This experiment will strengthen their finding. The scale bar is missing for all the images in Figure 5. The authors should also show magnified images of a single representative cell at 100x.

We apologize for the confusion regarding the scale bars. As mentioned here and elsewhere, the scale bars are present in the top-left image of each panel only, in order to avoid overcrowding the panel. All the images are already at 100X, with the exception of Fig. 5E (IF for SUB1 upon siSUB1 transfection) which is 60X in order to better show the lack of signal. We however acknowledge that the images are sometimes confusing, due to the PNG features once imported into the document. In any case, in the submission we have also provided the original images in high-quality PDF and .ai formats. The suggested experiment would require establishing a nucleolar fractionation protocol which we currently don’t have available and we argue that it is out of scope of the current study.

(10) Additionally, is rRNA synthesis affected in SUB1- and EPB41L4A-AS1-depleted cells? The authors could quantify newly synthesised rRNA levels in the nucleoli, which would also strengthen their findings about the role of this lncRNA in nucleolar biology.

We acknowledge that there are many aspects of the role of EPB41L4A-AS1 in nucleolar biology that remain to be explored, as well as in nucleolar biology itself, but given the extensive experimental data we already provide in this and other subjects, we respectfully suggest that this experiment is out of scope of the current work. We note that a recent study has shown that SUB1 is required for Pol I-mediated rDNA transcription in the nucleolus (Kaypee et al., 2025). In the presence of nucleolar SUB1, rDNA transcription proceeds as expected, but when SUB1 is depleted or its nucleolar localization is affected—by either sodium butyrate treatment or inhibition of KAT5-mediated phosphorylation at its lysine 35 (K35)—the levels of the 47S pre-rRNA are significantly reduced. In our settings, SUB1 enriches into the nucleolus following EPB41L4A-AS1 KD; thus, we might expect to see a slightly increased rDNA transcription or no effect at all, given that SUB1 localizes in the nucleolus in baseline conditions as well. We now mention this novel role of SUB1 both in the results and discussion.

“SUB1 interacts with all three RNA polymerases and was reported to be involved in transcription initiation and elongation, response to DNA damage, chromatin condensation(Conesa & Acker, 2010; Das et al., 2006; Garavís & Calvo, 2017; Hou et al., 2022), telomere maintenance(Dubois et al., 2025; Salgado et al., 2024) and rDNA transcription(Kaypee et al., 2025). SUB1 normally localizes throughout the nucleus in various cell lines, yet staining experiments show a moderate enrichment for the nucleolus (source: Human Protein Atlas; here; Kaypee et al., 2025).”

“Several features of the response to EPB41L4A-AS1 resemble nucleolar stress, including altered distribution of NPM1(Potapova et al., 2023; Yang et al., 2016). SUB1 was shown to be involved in many nuclear processes, including transcription(Conesa & Acker, 2010), DNA damage response(Mortusewicz et al., 2008; Yu et al., 2016), telomere maintenance(Dubois et al., 2025), and nucleolar processes including rRNA biogenesis(Kaypee et al., 2025; Tafforeau et al., 2013). Our results suggest a complex and multi-faceted relationship between EPB41L4A-AS1 and SUB1, as SUB1 mRNA levels are reduced by the transient (72 hours) KD of the lncRNA (Fig. 3B), the distribution of the protein in the nucleus is altered (Fig. 5A and 5C), while the protein itself is the most prominent binder of the mature EPB41L4A-AS1 in ENCODE eCLIP data (Fig. 3A). The most striking connection between EPB41L4A-AS1 and SUB1 is the similar phenotype triggered by their loss (Fig. 4). We note that a recent study has shown that SUB1 is required for Pol I-mediated rDNA transcription in the nucleolus(Kaypee et al., 2025). In the presence of nucleolar SUB1, rDNA transcription proceeds as expected, but when SUB1 is depleted or its nucleolar localization is affected—by either sodium butyrate treatment or inhibition of KAT5-mediated phosphorylation at its lysine 35 (K35)—the levels of the 47S pre-rRNA are significantly reduced. In our settings, SUB1 enriches into the nucleolus following EPB41L4A-AS1 KD; thus, we might expect to see a slightly increased rDNA transcription or no effect at all, given that SUB1 localizes in the nucleolus in baseline conditions as well. It is however difficult to determine which of the connections between these two genes is the most functionally relevant and which may be indirect and/or feedback interactions. For example, it is possible that EPB41L4A-AS1 primarily acts as a transcriptional regulator of SUB1 mRNA, or that its RNA product is required for proper stability and/or localization of the SUB1 protein, or that EPB41L4A-AS1 acts as a scaffold for the formation of protein-protein interactions of SUB1.”

(11) Figure 8: The scratch assay alone cannot be used as a measure of increased invasion, and this phenotype must be confirmed with a transwell invasion or migration assay. Thus, I highly recommend that the authors conduct this experiment using the Boyden chamber. Do the authors see upregulation of N-cadherin, Vimentin, and downregulation of E-cadherin in their RNA-seq?

We agree with the reviewer that those phenotypes are complex and normally require multiple in vitro, as well as in vivo assays to be thoroughly characterized. However, we respectfully consider those as out of scope of the current work, which is more focused on RNA biology and the molecular characterization and functions of EPB41L4A-AS1.

Nevertheless, in Fig. 8D we show that the canonical EMT signature (taken from MSigDB) is upregulated in cells with reduced expression of EPB41L4A-AS1. Notably, EMT has been found to not possess an unique gene expression program, but it rather involves distinct and partially overlapping gene signatures (Youssef et al., 2024). In Fig. 8D, the most upregulated gene is TIMP3, a matrix metallopeptidase inhibitor linked to a particular EMT signature that is less invasive and more profibrotic (EMT-T2, (Youssef et al., 2024)). Interestingly, we observed a strong upregulation of other genes linked to EMT-T2, such as TIMP1, FOSB, SOX9, JUNB, JUN and KLF4, whereas MPP genes (linked to EMT-T1, which is highly proteolytic and invasive) are generally downregulated or not expressed. With regards to N- and E-cadherin, the first does not pass our cutoff to be considered expressed, and the latter is not significantly changing. Vimentin is also not significantly dysregulated. All these examples are reported, which were added as Fig. 8E:

The text has also been updated accordingly:

“These findings suggest that proper EPB41L4A-AS1 expression is required for cellular proliferation, whereas its deficiency results in the onset of more aggressive and migratory behavior, likely linked to the increase of the gene signature of epithelial to mesenchymal transition (EMT) (Fig. 8D). Because EMT is not characterized by a unique gene expression program and rather involves distinct and partially overlapping gene signatures (Youssef et al., 2024), we checked the expression level of marker genes linked to different types of EMTs (Fig. 8E). The most upregulated gene in Fig. 8D is TIMP3, a matrix metallopeptidase inhibitor linked to a particular EMT signature that is less invasive and more profibrotic (EMT-T2) (Youssef et al., 2024). Interestingly, we observed a stark upregulation of other genes linked to EMT-T2, such as TIMP1, FOSB, SOX9, JUNB, JUN and KLF4, whereas MPP genes (linked to EMT-T1, which is highly proteolytic and invasive) are generally downregulated or not expressed. This suggests that the downregulation of EPB41L4A-AS1 is primarily linked to a specific EMT program (EMT-T2), and future studies aimed at uncovering the exact mechanisms and relevance will shed light upon a possible therapeutic potential of this lncRNA.”

(12) Minor points:(a) What could be the explanation for why only the EPB41L4A-AS1 locus has an effect on the neighbouring gene?

There might be multiple reasons why EPB41L4A-AS1 is able to modulate the expression of the neighboring genes. First, it is expressed from a TAD boundary exhibiting physical contacts with several genes in the two flanking TADs (Fig. 1F and 2A), placing it in the right spot to regulate their expression. Second, it is highly expressed when compared to most of the genes nearby, with transcription having been linked to the establishment and maintenance of TAD boundaries (Costea et al., 2023). Accordingly, the (partial) depletion of EPB41L4A-AS1 via GapmeRs transfection slightly reduces the contacts between the lncRNA and EPB41L4A loci (Fig. 2E and S4J), although this effect could also be determined by a premature transcription termination triggered by the GapmeRs.

There are a multitude of mechanisms by which lncRNAs with regulatory functions modulate the expression of one or more target genes in cis (Gil & Ulitsky, 2020), and our data do not unequivocally point to one of them. Distinguishing between these possibilities is a major challenge in the field and would be difficult to address in the context of this one study. It could be that the processive RNA polymerases at the EPB41L4A-AS1 locus are recruited to the neighboring loci, facilitated by the close proximity in the 3D space. It could also be possible that chromatin remodeling factors are recruited by the nascent RNA, and then promote and/or sustain the opening of chromatin at the target site. The latter possibility is intriguing, as this mechanism is proposed to be widespread among lncRNAs (Gil & Ulitsky, 2020; Oo et al., 2025) and we observed a significant reduction of H3K27ac levels at the EPB41L4A promoter region (Fig. 2D). Future studies combining chromatin profiling (e.g., CUT&RUN and ATAC-seq) and RNA pulldown experiments will shed light upon the exact mechanisms by which this lncRNA regulates the expression of target genes in cis and its interacting partners.

(b) The scale bar is missing on all the images in the Supplementary Figures as well.

The scale bars are present in the top-left figure of each panel. We acknowledge that due to the export as PNG, some figures (including those with microscopy images) display abnormal font sizes and aspect ratio. All images were created using consistent fonts, sizes and ratio, and are provided as high-quality PDF in the current submission.

(13) Methods:The authors should double-check if they used sirn and LNA gapmers at 25 and 50um concentrations, as that is a huge dose. Most papers used these reagents in the range of 5-50nM maximum.

We apologize for the typo, the text has been fixed. We performed the experiments at 25 and 50nM, respectively, as suggested by the manufacturer’s protocol.

(14) Discussion:Which cell lines were used in reference 27 (Cheng et al., 2024 Cell) to study the role of SNORA13? It may be useful to include this in the discussion.

We already mentioned the cell system in the discussion, and now we edited to include the specific cell line that was used:

“A recent study found that SNORA13 negatively regulates ribosome biogenesis in TERT-immortalized human fibroblasts (BJ-HRAS^G12V^), by decreasing the incorporation of RPL23 into the maturing 60S ribosomal subunits, eventually triggering p53-mediated cellular senescence(Cheng et al., 2024).”

**Reviewer #3 (Recommendations for the authors):**
Major comments on weaknesses:(1) The paper is quite disjointed:(a) Figures1/2 studied the cis- and potential trans target genes altered by EPB41L4A-AS1 knockdown. They also showed some data about EPB41L4A-AS1 overlaps a strong chromatin boundary.(b) Figures3/4/5 studied the role of SUB1 - as it is altered by EPB41L4A-AS1 knockdown - in affecting genes and snoRNAs, which may partially underlie the gene/snoRNA changes after EPB41L4A-AS1 knockdown.(c) Figure 6 showed that EPB41L4A-AS1 knockdown did not directly affect SNORA13, the snoRNA located in the intron of EPB41L4A-AS1. Thus, the upregulation of many snoRNAs is not due to SNORA13.(d) Figure 7 studied whether the changes of cis genes or snoRNAs are due to transcriptional stability.(e) Figure 8 studied cellular phenotypes after EPB41L4A-AS1 knockdown.These points are overly spread out and this dilutes the central theme of these results, which this Reviewer considered to be on cis or trans gene regulation by this lncRNA.The title of the paper implies EPB41L4A-AS1 knockdown affected trans target genes, but the paper did not focus on studying cis or trans effects, except briefly mentioning that many genes were changed in Figure 2. The many changes of snoRNAs are suggested to be partially explained by SUB1, but SUB1 itself is affected (>50%, Figure 3B) by EPB41L4A-AS1 knockdown, so it is unclear if these are mostly secondary changes due to SUB1 reduction. Given the current content of the paper, the authors do not have sufficient evidence to support that the changes of trans genes are due to direct effects or indirect effects. And so they are encouraged to revise their title to be more on snoRNA regulation, as this area took the majority of the efforts in this paper.

We respectfully disagree with the reviewer. We show that the effect on the proximal genes are cis-acting, as they are not rescued by exogenous expression, whereas the majority of the changes observed in the RNA-seq datasets appear to be indirect, and the snoRNA changes, that indeed might be indirect and not necessarily involve direct interaction partners of the lncRNA, such as SUB1, appear to be trans-regulated, as they can be rescued partially by exogenous expression of the lncRNA. We also show that KD of the main cis-regulated gene, EPB41L4A, results in a much milder transcriptional response, further solidifying the contribution of trans-acting effects. While we agree that the snoRNA effects are interesting, we do not consider them to be the main result, as they are accompanied by many additional changes in gene expression, and changes in the subnuclear distribution of the key nucleolar proteins, so it is difficult for us to claim that EPB41L4A-AS1 is specifically relevant to the snoRNAs rather than to the more broad nucleolar biology. Therefore, we prefer not to mention snoRNAs specifically in the title.

(2) EPB41L4A-AS1 knockdown caused ~2,364 gene changes. This is a very large amount of change on par with some transcriptional factors. It thus needs more scrutiny. First, on Page 9, second paragraph, the authors used|log2Fold-change| >0.41 to select differential genes, which is an unusual cutoff. What is the rationale? Often |log2Fold-change| >1 is more common. How many replicates are used? To examine how many gene changes are likely direct target genes, can the authors show how many of the cist-genes that are changed by EPB41L4A-AS1 knockdown have direct chromatin contacts with EPB41L4A-AS1 in HiC data? Is there any correlation between HiC contact with their fold changes? Without a clear explanation of cis target genes as direct target genes, it is more difficult to establish whether any trans target genes are directly affected by EPB41L4A-AS1 knockdown.

A |log_2_Fold-change| >0.41 equals a change of 33% or more, which together with an adjusted P < 0.05 is a threshold that has been used in the past. All RNA-seq experiments have been performed in triplicates, in line with the standards in the field. While it is possible that the EPB41L4A-AS1 establishes multiple contacts in trans—a process that has been observed in at least another lncRNA, namely Firre but involving its mature RNA product—we do believe this to be less likely that the alternative, namely that the > 2,000 DEGs are predominantly result from secondary changes rather than genes directly regulated by EPB41L4A-AS1 contacts.

In any case, we have inspected our UMI-4C data to identify other genes exhibiting higher contact frequencies than background levels, and thus, potentially regulated in cis. To this end, we calculated the UMI-4C coverage in a 10kb window centered around the TSS of the genes located on chromosome 5, which we subsequently normalized based on the distance from EPB41L4A-AS1, in order to account for the intrinsic higher DNA recovery the closer to the target DNA sequence. However, in our UMI-4C experiment we have employed baits targeting three different genes—EPB41L4A-AS1, EPB41L4A and STARD4—and therefore such approach assumes that the lncRNA locus has the most regulatory features in this region. As expected, we detected a strong negative correlation between the normalized coverage and the distance from the EPB41L4A-AS1 locus (⍴ = -0.51, p-value < 2.2e-16), and the genes in the two neighboring TADs exhibited the strongest association with the bait region (Author response image 5). The genes that we see are down-regulated in the adjacent TADs, namely NREP, MCC and MAN2A1 (Fig. 2F) show substantially higher contacts than background with the EPB41L4A-AS1 gene, thus potentially constituting additional cis-regulated targets of this lncRNA. We note that both SUB1 and NPM1 are located on chromosome 5 as well, albeit at distances exceeding 75 and 50 Mb, respectively, and they do not exhibit any striking association with the lncRNA locus.

**Author response image 5. sa4fig5:** UMI-4C coverage over the TSS of the genes located on chromosome 5. (A) Correlation between the normalized UMI-4C coverage over the TSS (± 5kb) of chromosome 5 genes and the absolute distance (in megabases, Mb) from EPB41L4A-AS1. (B) Same as in (A), but with the x axis showing the relative distance from EPB41L4A-AS1. In both cases, the genes in the two flanking TADs are colored in red and their names are reported.

To increase the confidence in our RNA-seq data, we have now performed another round of polyA+ RNA-seq following EPB41L4A-AS1 knockdown using LNA1 or LNA2, as well as the previously used and an additional control GapmeR. The FPKMs of the control samples are highly-correlated both within replicates and between GapmeRs (Fig. S6A). More importantly, the fold-changes to control are highly correlated between the two on-target GapmeRs LNA1 and LNA2, regardless of the GapmeR used for normalization (Fig. S6B), thus showing that despite significant GapmeR-specific effects, the bulk of the response is shared and likely the direct result of the reduction in the levels of EPB41L4A-AS1. Notably, key targets NPM1 and MTREX (see discussion, Fig. S12A-C and comments to Reviewer 3) were found to be downregulated by both LNAs (Fig. S6C).

However, we acknowledge that some of the dysregulated genes are observed only when using one GapmeR and not the other, likely due to a combination of indirect, secondary and non-specific effects, and as such it is difficult without short time-course experiments (Much et al., 2024) to infer the direct response. Supporting this, LNA2 yielded a total of 1,069 DEGs (617 up and 452 down) and LNA1 2,493 DEGs (1,328 up and 1,287 down), with the latter triggering a stronger response most likely as a result of the previously mentioned CDKN1A/p21 induction. Overall, 45.1% of the upregulated genes following LNA2 transfection were shared with LNA1, in contrast to only the 24.3% of the downregulated ones.

We have now included these results in the Results section (see below) and in Supplementary Figure (Fig. S6).

“Most of the consequences of the depletion of EPB41L4A-AS1 are thus not directly explained by changes in EPB41L4A levels. An additional trans-acting function for EPB41L4A-AS1 would therefore be consistent with its high expression levels compared to most lncRNAs detected in MCF-7 (Fig. S5G). To strengthen these findings, we have transfected MCF-7 cells with LNA1 and a second control GapmeR (NT2), as well as the previous one (NT1) and LNA2, and sequenced the polyadenylated RNA fraction as before. Notably, the expression levels (in FPKMs) of the replicates of both control samples are highly correlated with each other (Fig. S6A), and the global transcriptomic changes triggered by the two EPB41L4A-AS1-targeting LNAs are largely concordant (Fig. S6B and S6C). Because of this concordance and the cleaner (i.e., no CDKN1A upregulation) readout in LNA2-transfected cells, we focused mainly on these cells for subsequent analyses.”

Figure 3B, SUB1 mRNA is reduced >half by EPB41L4A-AS1 KD. How much did SUB1 protein reduce after EPB41L4A-AS1 KD? Similarly, how much is the NPM1 protein reduced? If these two important proteins were affected by EPB41L4A-AS1 KD simultaneously, it is important to exclude how many of the 2,364 genes that changed after EPB41L4A-AS1 KD are due to the protein changes of these two key proteins. For SUB1, Figures S7E,F,G provided some answers. But NPM1 KD is also needed to fully understand such. Related to this, there are many other proteins perhaps changed in addition to SUB1 and NPM1, this renders it concerning how many of the EPB41L4A-AS1 KD-induced changes are directly caused by this RNA. In addition to the suggested study of cist targets, the alternative mechanism needs to be fully discussed in the paper as it remains difficult to fully conclude direct versus indirect effect due to such changes of key proteins or ncRNAs (such as snoRNAs or histone mRNAs).

As requested by both Reviewer #2 and #3, we have performed WB for SUB1, NPM1 and FBL following EPB41L4A-AS1 KD with two targeting (LNA1 and LNA2) and the previous control GapmeRs. Interestingly, we did not detect any significant downregulation of either proteins (Author response image 3), although this might be the result of the high variability observed in the control samples. Moreover, the short timeframe in which the experiments have been conducted━that is, transient transfections for 3 days━might not be sufficient time for the existing proteins to be degraded, and thus, the downregulation is more evident at the RNA (Fig. 3B and Supplementary Figure 6C) rather than protein level.

We acknowledge that many proteins might change simultaneously, and to pinpoint which ones act upstream of the plethora of indirect changes is extremely challenging when considering such large-scale changes in gene expression. In the case of SUB1 and NPM1━which were prioritized for their predicted binding to the lncRNA (Fig. 3A)━we show that the depletion of the former affects the latter in a similar way than that of the lncRNA (Fig. 5F). Moreover, snoRNAs changes are also similarly affected (as the reviewer pointed out, Fig. 4F), suggesting that at least this phenomenon is predominantly mediated by SUB1. Other effects might also be indirect consequences of cellular responses, such as the decrease in histone mRNAs (Fig. 4A) that might reflect the decrease in cellular replication (Fig. 8C) and cell cycle genes (Fig. 2I) (although a link between SUB1 and histone mRNA expression has been described (Brzek et al., 2018)).

Supporting the notion that additional proteins might be involved in driving the observed phenotypes, one of the genes that most consistently was affected by EPB41L4A-AS1 KD with GapmeRs is MTREX (also known as MTR4), that becomes downregulated at both the RNA and protein levels (now presented in the main text as Supplementary Figure 12). MTREX it’s part of the NEXT and PAXT complexes (Contreras et al., 2023), that target several short-lived RNAs for degradation, and the depletion of either MTREX or other complex members leads to the upregulation of such RNAs, that include PROMPTs, uaRNAs and eRNAs, among others. Given the lack in our understanding in snoRNA biogenesis from introns in mammalian systems(Monziani & Ulitsky, 2023), it is tempting to hypothesize a role for MTREX-containing complexes in trimming and degrading those introns and release the mature snoRNAs.

We updated the discussion section to include these observations:

“Beyond its site of transcription, EPB41L4A-AS1 associates with SUB1, an abundant protein linked to various functions, and these two players are required for proper distribution of various nuclear proteins. Their dysregulation results in large-scale changes in gene expression, including up-regulation of snoRNA expression, mostly through increased transcription of their hosts, and possibly through a somewhat impaired snoRNA processing and/or stability. To further hinder our efforts in discerning between these two possibilities, the exact molecular pathways involved in snoRNAs biogenesis, maturation and decay are still not completely understood. One of the genes that most consistently was affected by EPB41L4A-AS1 KD with GapmeRs is MTREX (also known as MTR4), that becomes downregulated at both the RNA and protein levels (Fig. S12A-C). Interestingly, MTREX it is part of the NEXT and PAXT complexes(Contreras et al., 2023), that target several short-lived RNAs for degradation, and the depletion of either MTREX or other complex members leads to the upregulation of such RNAs, that include PROMPTs, uaRNAs and eRNAs, among others. It is therefore tempting to hypothesize a role for MTREX-containing complexes in trimming and degrading those introns, and releasing the mature snoRNAs. Future studies specifically aimed at uncovering novel players in mammalian snoRNA biology will both conclusively elucidate whether MTREX is indeed involved in these processes.”

With regards to the changes in gene expression between the two LNAs, we provide a more detailed answer above and to the other reviewers as well.

(3) A Strong discrepancy of results by different approaches of knockdown or overexpression:(a) CRISPRa versus LNA knockdown: Figure S4 - CRISPRa of EPB41L4A-AS1 did not affect EPB41L4A expression (Figure S4B). The authors should discuss how to interpret this result. Did CRISPRa not work to increase the nuclear/chromatin portion of EPB41L4A-AS1? Did CRISPRa of EPB41L4A-AS1 affect the gene in the upstream, the STARD4? Did CRISPRa of EPB41L4A-AS1 also affect chromatin interactions between EPB41L4A-AS1 and the EPB41L4A gene? If so, this may argue that chromatin interaction is not necessary for cis-gene regulation.

There are indeed several possible explanations, the most parsimonious is that since the lncRNA is already very highly transcribed, the relatively modest effect of additional transcription mediated by CRISPRa is not sufficient to elicit a measurable effect. For this reason, we did not check by UMI-4C the contact frequency between the lncRNA and EPB41L4A upon CRISPRa.

CRISPRa augments transcription at target loci, and thus, the nuclear and chromatin retention of EPB41L4A-AS1 are not expected to be affected. We did not check the expression of STARD4, because we focused on EPB41L4A which appears to be the main target locus according to Hi-C (Fig. 2A), UMI-4C (Fig. 2E and S4J) and GeneHancer (Fig. S1).

We already provide extensive evidence of a cis-regulation of EPB41L4A-AS1 over EPB41L4A, and show that EPB41L4A is lowly-expressed and likely has a limited role in our experimental settings. Thus, we respectfully propose that an in-deep exploration of the mechanism of action of this regulatory axis is out of scope of the current study, that instead focused more on the global effects of EPB41L4A-AS1 perturbation.

(b) Related to this, while CRISPRa alone did not show an effect, upon LNA knockdown of EPB41L4A-AS1, CRISPRa of EPB41L4A-AS1 can increase EPB41L4A expression. It is perplexing as to why, upon LNA treatment, CRISPRa will show an effect (Figure S4H)? Actually, Figures S4H and I are very confusing in the way they are currently presented. They will benefit from being separated into two panels (H into 2 and I into two). And for Ectopic expression, please show controls by empty vector versus EPB41L4A-AS1, and for CRISPRa, please show sgRNA pool versus sgRNA control.

The results are consistent with the parsimonious assumption mentioned above that the high transcription of the lncRNA at baseline is sufficient for maximal positive regulation of EPB41L4A, and that upon KD, the reduced transcription and/or RNA levels are no longer at saturating levels, and so CRISPRa can have an effect. We now mention this interpretation in the text:

“Levels of EPB41L4A were not affected by increased expression of EPB41L4A-AS1 from the endogenous locus by CRISPR activation (CRISPRa), nor by its exogenous expression from a plasmid (Fig. S4B and S4C). The former suggests that endogenous levels of EPB41L4A-AS1—that are far greater than those of EPB41L4A—are sufficient to sustain the maximal expression of this target gene in MCF7 cells.”

We apologize for the confusion regarding the control used in the rescue experiments in Fig. S4H and S4I. The “-” in the Ectopic overexpression and CRISPRa correspond to the Empty Vector and sgControl, respectively, and not the absence of any vector. We changed the text in the figure legends:

“(H) Changes in EPB41L4A-AS1 expression after rescuing EPB41L4A-AS1 with an ectopic plasmid or CRISPRa following its KD with GapmeRs. In both panels (Ectopic OE and CRISPRa) the “-” samples represent those transfected with the Empty Vector or sgControl. Asterisks indicate significance relative to the –/– control (transfected with both the control GapmeR and vector). (I) Same as in (H), but for changes in EPB41L4A expression.”

(c) siRNA versus LNA knockdown: Figure S3A showed that siRNA KD of EPB41L4A-AS1 does not affect EPB41L4A expression. How to understand this data versus LNA?

As explained in the text, siRNA-mediated KD presumably affects mostly the cytoplasmic pool of EPB41L4A-AS1 and not the nuclear one, which we assume explains the different effects of the two perturbations, as observed for other lncRNAs (e.g., (Ntini et al., 2018)). However, we acknowledge that we do not know what aspect of the nuclear RNA biology is relevant, let it be the nascent EPB41L4A-AS1 transcription, premature transcriptional termination or even the nuclear pool of this lncRNA, and this can be elucidated further in future studies.

(d) EPB41L4A-AS1 OE versus LNA knockdown: Figure 6F showed that EPB41L4A-AS1 OE caused reduction of EPB41L4A mRNA, particularly at 24hr. How to interpret that both LNA KD and OE of EPB41L4A-AS1 reduce the expression of EPB41L4A mRNA?

We do not believe that the OE of EPB41L4A-AS1, and in particular the one elicited by an ectopic plasmid affects EPB41L4A RNA levels. In the experiment in Fig. 6F, EPB41L4A relative expression at 24h is ~0.65 (please note the log_2_ scale in the graph), which is significant as reported. However, throughout this study (and as shown in Fig. S4C for the ectopic and Fig. S4B for the CRISPRa overexpression, respectively), we observed no such behavior, suggesting that the effect reported in Fig. 6F is the result of either that particular setting, and unlikely to reflect a general phenomenon.

(e) Did any of the effects on snoRNAs or trans target genes after EPB41L4A-AS1 knockdown still appear by CRISPRa?

As mentioned above, we did a limited number of experiments after CRISPRa, prompted by the fact that endogenous levels of EPB41L4A-AS1 are already high enough to sustain its functions. Pushing the expression even higher will likely result in no or artifactual effects, which is why we respectfully propose such experiments are not essential in this current work, which instead mostly relies on loss-of-function experiments.

For issue 3, extensive data repetition using all these methods may be unrealistic, but key data discrepancy needs to be fully discussed and interpreted.Other comments on weakness:(1) This manuscript will benefit from having line numbers so comments from Reviewers can be made more specifically.

We added line numbers as suggested by the reviewer.

(2) Figure 2G, to distinguish if any effects of EPB41L4A-AS1 come from the cytoplasmic or nuclear portion of EPB41L4A-AS1, an siRNA KD RNA-seq will help to filter out the genes affected by EPB41L4A-AS1 in the cytoplasm, as siRNA likely mainly acts in the cytoplasm.

This experiment would be difficult to interpret as while the siRNAs mostly deplete the cytoplasmic pool of their target, they can have some effects in the nucleus as well (e.g., (Sarshad et al., 2018)) and so siRNAs knockdown will not necessarily report strictly on the cytoplasmic functions.

(3) Figure 2H, LNA knockdown of EPB41L4A should check the protein level reduction, is it similar to the change caused by knockdown of EPB41L4A-AS1?

As suggested by reviewer #2, we have now replaced the EPB41L4A Western Blot that now shows the results with both LNA1 and LNA2. Please note that the previous Fig. 2C was a subset of this, i.e., we have previously cropped the results obtained with LNA1. Unfortunately, we did not have sufficient antibody to check for EPB41L4A protein reduction following LNA KD of EPB41L4A in a timely manner.

(4) There are two LNA Gapmers used by the paper to knock down EPB41L4A-AS1, but some figures used LNA1, some used LNA2, preventing a consistent interpretation of the results. For example, in Figures 2A-D, LNA2 was used. But in Figures 2E-H, LNA1 was used. How consistent are the two in changing histone H3K27ac (like in Figure 2D) versus gene expression in RNA-seq? The changes in chromatin interaction appear to be weaker by LNA2 (Figure S4J) versus LNA1 (Figure 2E).

As explained above and in response to Reviewer #1, we now provide more RNA-seq data for LNA1 and LNA2. We note that besides the unwanted and/or off-target effects, these two GapmeRs might be not equally effective in knocking down EPB41L4A-AS1, which could explain why LNA1 seems to have a stronger effect on chromatin than LNA2. Nonetheless, when we have employed both we have obtained similar and consistent results (e.g., Fig. 5A-D and 8A-C), suggesting that these and the other effects are indeed on target effects due to EPB41L4A-AS1 depletion.

(5) It will be helpful if the authors provide information on how long they conducted EPB41L4A-AS1 knockdown for most experiments to help discern direct or indirect effects.

The length of all perturbations was indicated in the Methods section, and we now mention them also in the Results. Unless specified otherwise, they were carried out for 72 hours. We agree with the reviewer that having time course experiments can have added value, but due to the extensive effort that these will require, we suggest that they are out of scope of the current study.

(6) In Figures 1C and F, the authors showed results about EPB41L4A-AS1 overlapping a strong chromatin boundary. But these are not mentioned anymore in the later part of the paper. Does this imply any mechanism? Does EPB41L4A-AS1 knockdown or OE, or CRISPRa affect the expression of genes near the other interacting site, STARD4? Do genes located in the two adjacent TADs change more strongly as compared to other genes far away?

We discuss this point in the Discussion section:

“At the site of its own transcription, which overlaps a strong TAD boundary, EPB41L4A-AS1 is required to maintain expression of several adjacent genes, regulated at the level of transcription. Strikingly, the promoter of EPB41L4A-AS1 ranks in the 99.8th percentile of the strongest TAD boundaries in human H1 embryonic stem cells(Open2C et al., 2024; Salnikov et al., 2024). It features several CTCF binding sites (Fig. 2A), and in MCF-7 cells, we demonstrate that it blocks the propagation of the 4C signal between the two flanking TADSs (Fig. 1F). Future studies will help elucidate how EPB41L4A-AS1 transcription and/or the RNA product regulate this boundary. So far, we found that EPB41L4A-AS1 did not affect CTCF binding to the boundary, and while some peaks in the vicinity of EPB41L4A-AS1 were significantly affected by its loss, they did not appear to be found near genes that were dysregulated by its KD (Fig. S11C). We also found that KD of EPB41L4A-AS1—which depletes the RNA product, but may also affect the nascent RNA transcription(Lai et al., 2020; Lee & Mendell, 2020)—reduces the spatial contacts between the TAD boundary and the EPB41L4A promoter (Fig. 2E). Further elucidation of the exact functional entity needed for the cis-acting regulation will require detailed genetic perturbations of the locus, that are difficult to carry out in the polypoid MCF-7 cells, without affecting other functional elements of this locus or cell survival as we were unable to generate deletion clones despite several attempts.”

As mentioned in the text (pasted below) and in Fig. 2F, most genes in the two flanking TADs become downregulated following EPB41L4A-AS1 KD. While STARD4 – which was chosen because it had spatial contacts above background with EPB41L4A-AS1 – did not reach statistical significance, others did and are highlighted. Those included NREP, which we also discuss:

“Consistently with the RT-qPCR data, KD of EPB41L4A-AS1 reduced EPB41L4A expression, and also reduced expression of several, but not all other genes in the TADs flanking the lncRNA (Fig. 2F).Based on these data, EPB41L4A-AS1 is a significant cis-acting activator according to TransCistor (Dhaka et al., 2024) (P=0.005 using the digital mode). The cis-regulated genes reduced by EPB41L4A-AS1 KD included NREP, a gene important for brain development, whose homolog was downregulated by genetic manipulations of regions homologous to the lncRNA locus in mice(Salnikov et al., 2024). Depletion of EPB41L4A-AS1 thus affects several genes in its vicinity.”

(7) Related to the description of SUB1 regulation of genes are DNA and RNA levels: "Of these genes, transcripts of only 56 genes were also bound by SUB1 at the RNA level, suggesting largely distinct sets of genes targeted by SUB1 at both the DNA and the RNA levels." SUB1 binding to chromatin by Cut&Run only indicates that it is close to DNA/chromatin, and this interaction with chromatin may still likely be mediated by RNAs. The authors used SUB1 binding sites in eCLIP-seq to suggest whether it acts via RNAs, but these binding sites are often from highly expressed gene mRNAs/exons. Standard analysis may not have examined low-abundance RNAs close to the gene promoters, such as promoter antisense RNAs. The authors can examine whether, for the promoters with cut&run peaks of SUB1, SUB1 eCLIP-seq shows binding to the low-abundance nascent RNAs near these promoters.

In response to a related comment by Reviewer 1, we now show that when considering expression level–matched control genes, knockdown of EPB41L4A-AS1 still significantly affects expression of SUB1 targets over controls. The results are presented in Supplementary Figure 7 (Fig. S7C).

Based on this analysis, while there is a tendency of increased expression with increased SUB1 binding, when controlling for expression levels the effect of down-regulation of SUB1-bound RNAs upon lncRNA knockdown remains, suggesting that it is not merely a confounding effect. We have updated the text as follows:

“We hypothesized that loss of EPB41L4A-AS1 might affect SUB1, either via the reduction in its expression or by affecting its functions. We stratified SUB1 eCLIP targets into confidence intervals, based on the number, strength and confidence of the reported binding sites. Indeed, eCLIP targets of SUB1 (from HepG2 cells profiled by ENCODE) were significantly downregulated following. EPB41L4A-AS1 KD in MCF-7, with more confident targets experiencing stronger downregulation (Fig. 3C). Importantly, this still holds true when controlling for gene expression levels (Fig. S7C), suggesting that this negative trend is not due to differences in their baseline expression.”

(8) Figure 8, the cellular phenotype is interesting. As EPB41L4A-AS1 is quite widely expressed, did it affect the phenotypes similarly in other breast cancer cells? MCF7 is not a particularly relevant metastasis model. Can a similar phenotype be seen in commonly used metastatic cell models such as MDA-MB-231?

We agree that further expanding the models in which EPB41L4A-AS1 affects cellular proliferation, migration and any other relevant phenotype is of potential interest before considering targeting this lncRNA as a therapeutic approach. However, given that (1) others have already identified similar phenotypes upon the modulation of EPB41L4A-AS1 in a variety of different systems (see Results and Discussion), and (2) we were most interested in the molecular consequences following the loss of this lncRNA, we respectfully suggest that these experiments are out of scope of the current study.

References

Bahar Halpern, K., Caspi, I., Lemze, D., Levy, M., Landen, S., Elinav, E., Ulitsky, I., & Itzkovitz, S. (2015). Nuclear Retention of mRNA in Mammalian Tissues. Cell Reports, 13(12), 2653–2662.

Brabletz, T., Kalluri, R., Nieto, M. A., & Weinberg, R. A. (2018). EMT in cancer. Nature Reviews. Cancer, 18(2), 128–134.

Brzek, A., Cichocka, M., Dolata, J., Juzwa, W., Schümperli, D., & Raczynska, K. D. (2018). Positive cofactor 4 (PC4) contributes to the regulation of replication-dependent canonical histone gene expression. BMC Molecular Biology, 19(1), 9.

Cheng, Y., Wang, S., Zhang, H., Lee, J.-S., Ni, C., Guo, J., Chen, E., Wang, S., Acharya, A., Chang, T.-C., Buszczak, M., Zhu, H., & Mendell, J. T. (2024). A non-canonical role for a small nucleolar RNA in ribosome biogenesis and senescence. Cell, 187(17), 4770–4789.e23.

Conesa, C., & Acker, J. (2010). Sub1/PC4 a chromatin associated protein with multiple functions in transcription. RNA Biology, 7(3), 287–290.

Contreras, X., Depierre, D., Akkawi, C., Srbic, M., Helsmoortel, M., Nogaret, M., LeHars, M., Salifou, K., Heurteau, A., Cuvier, O., & Kiernan, R. (2023). PAPγ associates with PAXT nuclear exosome to control the abundance of PROMPT ncRNAs. Nature Communications, 14(1), 6745.

Costea, J., Schoeberl, U. E., Malzl, D., von der Linde, M., Fitz, J., Gupta, A., Makharova, M., Goloborodko, A., & Pavri, R. (2023). A de novo transcription-dependent TAD boundary underpins critical multiway interactions during antibody class switch recombination. Molecular Cell, 83(5), 681–697.e7.

Das, C., Hizume, K., Batta, K., Kumar, B. R. P., Gadad, S. S., Ganguly, S., Lorain, S., Verreault, A., Sadhale, P. P., Takeyasu, K., & Kundu, T. K. (2006). Transcriptional coactivator PC4, a chromatin-associated protein, induces chromatin condensation. Molecular and Cellular Biology, 26(22), 8303–8315.

Dhaka, B., Zimmerli, M., Hanhart, D., Moser, M. B., Guillen-Ramirez, H., Mishra, S., Esposito, R., Polidori, T., Widmer, M., García-Pérez, R., Julio, M. K., Pervouchine, D., Melé, M., Chouvardas, P., & Johnson, R. (2024). Functional identification of cis-regulatory long noncoding RNAs at controlled false discovery rates. Nucleic Acids Research, 52(6), 2821–2835.

Didiot, M.-C., Ferguson, C. M., Ly, S., Coles, A. H., Smith, A. O., Bicknell, A. A., Hall, L. M., Sapp, E., Echeverria, D., Pai, A. A., DiFiglia, M., Moore, M. J., Hayward, L. J., Aronin, N., & Khvorova, A. (2018). Nuclear Localization of Huntingtin mRNA Is Specific to Cells of Neuronal Origin. Cell Reports, 24(10), 2553–2560.e5.

Dongre, A., & Weinberg, R. A. (2019). New insights into the mechanisms of epithelial-mesenchymal transition and implications for cancer. Nature Reviews. Molecular Cell Biology, 20(2), 69–84.

Dubois, J.-C., Bonnell, E., Filion, A., Frion, J., Zimmer, S., Riaz Khan, M., Teplitz, G. M., Casimir, L., Méthot, É., Marois, I., Idrissou, M., Jacques, P.-É., Wellinger, R. J., & Maréchal, A. (2025). The single-stranded DNA-binding factor SUB1/PC4 alleviates replication stress at telomeres and is a vulnerability of ALT cancer cells. Proceedings of the National Academy of Sciences of the United States of America, 122(2), e2419712122.

Garavís, M., & Calvo, O. (2017). Sub1/PC4, a multifaceted factor: from transcription to genome stability. Current Genetics, 63(6), 1023–1035.

Gil, N., & Ulitsky, I. (2020). Regulation of gene expression by cis-acting long non-coding RNAs. Nature Reviews. Genetics, 21(2), 102–117.

Hou, Y., Gan, T., Fang, T., Zhao, Y., Luo, Q., Liu, X., Qi, L., Zhang, Y., Jia, F., Han, J., Li, S., Wang, S., & Wang, F. (2022). G-quadruplex inducer/stabilizer pyridostatin targets SUB1 to promote cytotoxicity of a transplatinum complex. Nucleic Acids Research, 50(6), 3070–3082.

Jan, C. H., Friedman, R. C., Ruby, J. G., & Bartel, D. P. (2011). Formation, regulation and evolution of *Caenorhabditis elegans* 3’UTRs. Nature, 469(7328), 97–101.

Kaypee, S., Ochiai, K., Shima, H., Matsumoto, M., Alam, M., Ikura, T., Kundu, T. K., & Igarashi, K. (2025). Positive coactivator PC4 shows dynamic nucleolar distribution required for rDNA transcription and protein synthesis. Cell Communication and Signaling : CCS, 23(1), 283.

Lai, F., Damle, S. S., Ling, K. K., & Rigo, F. (2020). Directed RNase H Cleavage of Nascent Transcripts Causes Transcription Termination. Molecular Cell, 77(5), 1032–1043.e4.

Lee, J.-S., & Mendell, J. T. (2020). Antisense-Mediated Transcript Knockdown Triggers Premature Transcription Termination. Molecular Cell, 77(5), 1044–1054.e3.

Lubelsky, Y., & Ulitsky, I. (2018). Sequences enriched in Alu repeats drive nuclear localization of long RNAs in human cells. Nature, 555(7694), 107–111.

Ly, S., Didiot, M.-C., Ferguson, C. M., Coles, A. H., Miller, R., Chase, K., Echeverria, D., Wang, F., Sadri-Vakili, G., Aronin, N., & Khvorova, A. (2022). Mutant huntingtin messenger RNA forms neuronal nuclear clusters in rodent and human brains. Brain Communications, 4(6), fcac248.

Maranon, D. G., & Wilusz, J. (2020). Mind the Gapmer: Implications of Co-transcriptional Cleavage by Antisense Oligonucleotides. Molecular Cell, 77(5), 932–933.

Monziani, A., & Ulitsky, I. (2023). Noncoding snoRNA host genes are a distinct subclass of long noncoding RNAs. Trends in Genetics : TIG, 39(12), 908–923.

Mortusewicz, O., Roth, W., Li, N., Cardoso, M. C., Meisterernst, M., & Leonhardt, H. (2008). Recruitment of RNA polymerase II cofactor PC4 to DNA damage sites. The Journal of Cell Biology, 183(5), 769–776.

Much, C., Lasda, E. L., Pereira, I. T., Vallery, T. K., Ramirez, D., Lewandowski, J. P., Dowell, R. D., Smallegan, M. J., & Rinn, J. L. (2024). The temporal dynamics of lncRNA Firre-mediated epigenetic and transcriptional regulation. Nature Communications, 15(1), 6821.

Ntini, E., Louloupi, A., Liz, J., Muino, J. M., Marsico, A., & Ørom, U. A. V. (2018). Long ncRNA A-ROD activates its target gene DKK1 at its release from chromatin. Nature Communications, 9(1), 1636.

Oo, J. A., Warwick, T., Pálfi, K., Lam, F., McNicoll, F., Prieto-Garcia, C., Günther, S., Cao, C., Zhou, YGavrilov, A. A., Razin, S. V., Cabrera-Orefice, A., Wittig, I., Pullamsetti, S. S., Kurian, L., Gilsbach, R., Schulz, M. H., Dikic, I., Müller-McNicoll, M., … Leisegang, M. S. (2025). Long non-coding RNAs direct the SWI/SNF complex to cell type-specific enhancers. Nature Communications, 16(1), 131.

Open2C, Abdennur, N., Abraham, S., Fudenberg, G., Flyamer, I. M., Galitsyna, A. A., Goloborodko, A., Imakaev, M., Oksuz, B. A., Venev, S. V., & Xiao, Y. (2024). Cooltools: Enabling high-resolution Hi-C analysis in Python. PLoS Computational Biology, 20(5), e1012067.

Potapova, T. A., Unruh, J. R., Conkright-Fincham, J., Banks, C. A. S., Florens, L., Schneider, D. A., & Gerton, J. L. (2023). Distinct states of nucleolar stress induced by anticancer drugs. https://doi.org/10.7554/eLife.88799.

Ray, D., Laverty, K. U., Jolma, A., Nie, K., Samson, R., Pour, S. E., Tam, C. L., von Krosigk, N., Nabeel-Shah, S., Albu, M., Zheng, H., Perron, G., Lee, H., Najafabadi, H., Blencowe, B., Greenblatt, J., Morris, Q., & Hughes, T. R. (2023). RNA-binding proteins that lack canonical RNA-binding domains are rarely sequence-specific. Scientific Reports, 13(1), 5238.

Salgado, S., Abreu, P. L., Moleirinho, B., Guedes, D. S., Larcombe, L., & Azzalin, C. M. (2024). Human PC4 supports telomere stability and viability in cells utilizing the alternative lengthening of telomeres mechanism. EMBO Reports, 25(12), 5294–5315.

Salnikov, P., Korablev, A., Serova, I., Belokopytova, P., Yan, A., Stepanchuk, Y., Tikhomirov, S., & Fishman, V. (2024). Structural variants in the Epb41l4a locus: TAD disruption and Nrep gene misregulation as hypothetical drivers of neurodevelopmental outcomes. Scientific Reports, 14(1), 5288.

Sarshad, A. A., Juan, A. H., Muler, A. I. C., Anastasakis, D. G., Wang, X., Genzor, P., Feng, X., Tsai, P.-F., Sun, H.-W., Haase, A. D., Sartorelli, V., & Hafner, M. (2018). Argonaute-miRNA Complexes Silence Target mRNAs in the Nucleus of Mammalian Stem Cells. Molecular Cell, 71(6), 1040–1050.e8.

Tafforeau, L., Zorbas, C., Langhendries, J.-L., Mullineux, S.-T., Stamatopoulou, V., Mullier, R., Wacheul, L., & Lafontaine, D. L. J. (2013). The complexity of human ribosome biogenesis revealed by systematic nucleolar screening of Pre-rRNA processing factors. Molecular Cell, 51(4), 539–551.

Unfried, J. P., & Ulitsky, I. (2022). Substoichiometric action of long noncoding RNAs. Nature Cell Biology, 24(5), 608–615.

Van Nostrand, E. L., Freese, P., Pratt, G. A., Wang, X., Wei, X., Xiao, R., Blue, S. M., Chen, J.-Y.,Cody, N. A. L., Dominguez, D., Olson, S., Sundararaman, B., Zhan, L., Bazile, C., Bouvrette, L. P. B., Bergalet, J., Duff, M. O., Garcia, K. E., Gelboin-Burkhart, C., … Yeo, G. W. (2020). A large-scale binding and functional map of human RNA-binding proteins. Nature, 583(7818), 711–719.

Yang, K., Wang, M., Zhao, Y., Sun, X., Yang, Y., Li, X., Zhou, A., Chu, H., Zhou, H., Xu, J., Wu, M., Yang, J., & Yi, J. (2016). A redox mechanism underlying nucleolar stress sensing by nucleophosmin. Nature Communications, 7, 13599.

Youssef, K. K., Narwade, N., Arcas, A., Marquez-Galera, A., Jiménez-Castaño, R., Lopez-Blau, C., Fazilaty, H., García-Gutierrez, D., Cano, A., Galcerán, J., Moreno-Bueno, G., Lopez-Atalaya, J. P., & Nieto, M. A. (2024). Two distinct epithelial-to-mesenchymal transition programs control invasion and inflammation in segregated tumor cell populations. Nature Cancer, 5(11), 1660–1680.

Yu, L., Ma, H., Ji, X., & Volkert, M. R. (2016). The Sub1 nuclear protein protects DNA from oxidative damage. Molecular and Cellular Biochemistry, 412(1-2), 165–171.